# Dynamin 1xA interacts with Endophilin A1 via its spliced long C-terminus for ultrafast endocytosis

Yuuta Imoto[1,2,12], Jing Xue [3,12], Lin Luo[4], Sumana Raychaudhuri[1], Kie Itoh[1,2], Ye Ma [5], George E Craft [3], Ann H Kwan[6], Tyler H Ogunmowo[1], Annie Ho [1], Joel P Mackay[7], Taekjip Ha[5,8,9], Shigeki Watanabe [1,10,11 ✉] & Phillip J Robinson[3 ✉]

## Abstract

**Dynamin 1 mediates fission of endocytic synaptic vesicles in the brain and has two major splice variants, Dyn1xA and Dyn1xB, which are nearly identical apart from the extended C-terminal region of Dyn1xA. Despite a similar set of binding partners, only Dyn1xA is enriched at endocytic zones and accelerates vesicle fission during ultrafast endocytosis. Here, we report that Dyn1xA achieves this localization by preferentially binding to Endophilin A1 through a newly defined binding site within its long C-terminal tail extension. Endophilin A1 binds this site at higher affinity than the previously reported site, and the affinity is determined by amino acids within the Dyn1xA tail but outside the binding site. This interaction is regulated by the phosphorylation state of two serine residues specific to the Dyn1xA variant. Dyn1xA and Endophilin A1 colocalize in patches near the active zone, and mutations disrupting Endophilin A binding to the long tail cause Dyn1xA mislocalization and stalled endocytic pits on the plasma membrane during ultrafast endocytosis. Together, these data suggest that the specificity for ultrafast endocytosis is defined by the phosphorylation-regulated interaction of Endophilin A1 with the C-terminal extension of Dyn1xA.**

**Keywords** Dynamin Splice Variants; Endophilin; Amphiphysin; Ultrafast Endocytosis; Flash-and-freeze
**Subject Categories** Membranes & Trafficking; Neuroscience; Organelles

## Introduction

Dynamin and its binding partners generate vesicles from the plasma membrane during different modes of endocytosis. They assemble a contractile collar around the neck of vesicles and sever the vesicles upon GTP hydrolysis by dynamin GTPase. This macromolecular complex forms via the specific interaction of Dynamin's multiple proline-rich motifs (PRM) within a large intrinsically disordered region called the proline-rich region (PRR) of the tail with the Src-homology 3 (SH3) domain of different endocytic proteins like Endophilin A (Ringstad et al, 1997), Syndapin 1 (Qualmann et al, 1999), and Amphiphysin 1 (David et al, 1996). In addition to the C-terminal SH3 domain, these partners contain an N-terminal Bin/Amphiphysin/RVS (BAR) domain, which binds phospholipids and senses membrane curvature and is able to tubulate membranes (Frost et al, 2008). These BAR proteins typically accumulate on the plasma membrane sequentially and form a rigid scaffolding structure to generate and stabilize the neck of endocytic vesicles (Daumke et al, 2014; Taylor et al, 2011). Dynamin is then recruited to the sites of endocytosis for the fission reaction (Taylor et al, 2011, 2012). Thus, the specific interaction of these proteins is essential for endocytosis. However, it is unclear which partner is involved in which endocytic mode and at what step in the process.

SH3 domains are capable of binding to PxxP-containing sequences (where P is proline and x is any amino acid) but also require non-PxxP sequences. Essential to most motifs are basic residues R or K within 2–4 amino acids flanking the C-terminal or N-terminal side of the PxxP (Li, 2005), defined as Class I ([R/K] xxPxxP) or II (PxxPx[R/K]), respectively (Kaneko et al, 2008). These flanking basic residues control the binding orientation and also determine specificity for SH3/PxxP interactions by forming salt bridges. While this interaction is typically with micromolar affinity, the full-length SH3 domain of Syndapin I achieves nanomolar affinity by engaging further distal 'elements', amino acids located short or long distances (SDE or LDEs) from the core PxxP motif, which are still within intrinsically disordered regions (Luo et al, 2016).

The neuronally enriched isoform of dynamin, Dynamin 1 (Dyn1), mediates fission of endocytic vesicles (Ferguson and De Camilli, 2012; Haucke et al, 2011; Saheki and De Camilli, 2012).

[1]Department of Cell Biology, Johns Hopkins University School of Medicine, Baltimore, MD, USA. [2]Department of Developmental Neurobiology, St. Jude Children's Research Hospital, 262 Danny Thomas Place, Memphis, TN 38105, USA. [3]Cell Signalling Unit, Children's Medical Research Institute, The University of Sydney, Locked Bag 23, Wentworthville 2145 NSW, Australia. [4]Institute for Molecular Bioscience, Institute for Molecular Bioscience Centre for Inflammation and Disease Research, and Australian Infectious Diseases Research Centre, The University of Queensland, Brisbane, QLD 4072, Australia. [5]Department of Biomedical Engineering, Johns Hopkins University, Baltimore, MD, USA. [6]School of Life and Environmental Sciences and Sydney Nano Institute, University of Sydney, Camperdown, NSW, Australia. [7]School of Life and Environmental Sciences, University of Sydney, Camperdown, NSW 2006, Australia. [8]Department of Biophysics and Biophysical Chemistry, Johns Hopkins University, Baltimore, MD, USA. [9]Howard Hughes Medical Institute, Baltimore, MD, USA. [10]The Center for Cell Dynamics, Johns Hopkins University, Baltimore, MD, USA. [11]Solomon H. Snyder Department of Neuroscience, Johns Hopkins University, School of Medicine, Baltimore, MD, USA. [12]These authors contributed equally: Yuuta Imoto, Jing Xue. ✉E-mail: shigeki.watanabe@jhmi.edu; probinson@cmri.org.au

**Glossary**

| | | | |
|---|---|---|---|
| LDE | long distance element | PRR | proline rich region |
| PRM | proline rich motif | SH3 | src-3 homology |

During neuronal activity, synaptic vesicles containing neurotransmitter undergo exocytosis at synapses, and subsequently, these vesicles are retrieved by endocytosis. Dyn1 mediates fission of endocytic vesicles as evidenced by stalled endocytic pits on the plasma membrane in neurons lacking Dyn1 (Ferguson et al, 2007; Kittelmann et al, 2013; Raimondi et al, 2011). This process also requires a coordinated action of several endocytic proteins through the PRM and SH3-domain containing protein interaction (Anggono et al, 2006; Daumke et al, 2014; Ferguson and De Camilli, 2012; Haucke et al, 2011).

Dyn1 has two major splice variants differing at its C-terminus, Dyn1xA and Dyn1xB; both containing 10 shared PRMs (Robinson et al, 1993). The two alternative C-terminal tails begin after residue P844 in its large intrinsically disordered region, and Dyn1xA and xB contain 20 and 7 unique amino acids at their C-terminal extensions, respectively. The long tail of Dyn1xA provides 3 extra PxxP motifs, while the short tail of Dyn1xB contains no additional PxxP motifs but constitutes a conserved calcineurin-binding motif (PxIxI[T/S]) (Xue et al, 2011). Binding of endocytic proteins to both variants is regulated by the phosphorylation status of the shared residues at S774 and S778 (in a region collectively termed phosphobox-1) (Clayton et al, 2010; Xue et al, 2011). There are also two additional phosphorylated serines (S851 and S857) unique to the extension of Dyn1xA splice variants (collectively termed phosphobox-2) with no known function (Graham et al, 2007). Although the degree of phosphorylation at these sites is variable across these two splice variants (Chan et al, 2010), neuronal activity induces calcineurin-mediated dephosphorylation at both phosphoboxes. Dephosphorylation of phosphobox-1 allows Dyn1 interaction with Syndapin 1, but was reported to not regulate Amphiphysin 1 or Endophilin A interactions (Anggono et al, 2006; Luo et al, 2016). However, the function of the long tail extension and the role of phosphobox-2 in Dyn1xA are unclear.

Although Dyn1xA and xB appear to bind the same set of proteins overall (with calcineurin binding selectively to the shorter xB), these two variants display different localization patterns and function on two distinct endocytic pathways at synapses. Dyn1xA forms liquid-like condensates through its interaction with Syndapin 1 near the shared phosphobox-1 and is highly enriched at the endocytic zone within presynaptic boutons (Imoto et al, 2022). Through this interaction, Dyn1xA mediates ultrafast endocytosis, a clathrin-independent form of endocytosis that retrieves synaptic vesicle membranes on a millisecond time scale at the physiological range of stimuli (Imoto et al, 2022; Watanabe et al, 2013a, 2013b). Similarly, Dyn1xB interacts with Syndapin 1 at phosphobox-1. However, Dyn1xB is predominantly cytoplasmic (Imoto et al, 2022) and mediates bulk membrane retrieval after intense neuronal activity (activity-dependent bulk endocytosis) (Xue et al, 2011; Cheung and Cousin, 2019). The marked difference between these two variants clearly arises from their respective C-terminal tail extensions. By its interaction with calcineurin, the phosphobox-1 of Dyn1xB is dephosphorylated during intense neuronal activity, allowing Syndapin 1 binding. However, regardless of the

phosphorylation status, Dyn1xB does not mediate ultrafast endocytosis (Imoto et al, 2022). Thus, the C-terminal tail extension of Dyn1xA likely specifies its localization and function.

To understand what makes Dyn1xA so unique, we aimed to uncover the binding partners of Dyn1xA involved in its long tail extension. We performed mass spectrometry analysis, site-directed mutagenesis, and nuclear magnetic resonance (NMR) chemical shift perturbation (CSP) experiments. We showed that a 40 kDa protein, identified as Endophilin A1, was selectively associated with a newly identified binding site restricted to Dyn1xA near phosphobox-2, with higher affinity than its previously known binding site near phosphobox-1 (Anggono et al, 2006), resulting in this variant uniquely containing two endophilin interaction sites. Endophilin A1 bound to a specific PRM near phosphobox-2, and binding was enhanced by several amino acid residues near the splice boundary including R846. Mutating R846 to alanine or double mutation of S851/857 to phosphomimetics in Dyn1xA abolished this interaction, revealing a Class II binding mode and the presence of SDEs and LDEs in the extension that contribute to the observed high affinity binding to this site (Kaneko et al, 2008). Consistent with this, STED imaging showed that Endophilin A1 and wild-type Dyn1xA colocalize at the endocytic zone in mouse hippocampal synapses, but Dyn1xA-R846A and Dyn1xA-S851/857D are diffuse along axons. Likewise, flash-and-freeze electron microscopy shows that wild-type Dyn1xA rescues the defect in ultrafast endocytosis in Dyn1 knock out (KO) neurons, but neither Dyn1xA-S851/857D nor Dyn1xA-R846A rescues the phenotype. The collective data indicate that phospho-regulated Endophilin A binding to the 20 amino acid extension of Dyn1xA determines its specificity for ultrafast endocytosis at synapses.

## Results

### Endophilin A binds the C-terminal extension of Dyn1xA

The initial hypothesis was that additional SH3 containing proteins bind to one or more of the three additional PRMs exclusive to the C-terminal PRR extension at the tail of Dyn1xA (Fig. 1A). To identify the potential isoform-selective binding partners, the full-length PRRs of Dyn1xA-746 to 864 and xB-746 to 851 (hereafter, Dyn1xA-PRR and Dyn1xB-PRR, respectively) were expressed as GST-fusion proteins bound to GSH-sepharose beads and incubated with rat brain synaptosomal lysate for pull-down experiments. A protein at ~40 kDa was selectively enriched in its association with Dyn1xA-PRR (Fig. 1B,C). LC–MS/MS analysis revealed that this is Endophilin A1 (Fig. EV1A). To quantify the relative abundance of the three known Endophilin A isoforms a quantitative selective reaction monitoring (SRM) mass spectrometry-based assay was performed on the excised 40 kDa band, revealing the predominant species as Endophilin A1, 11-fold lower A3 and over 250-fold lower A2 (Fig. EV1). However, A2 has an insert sequence increasing its mass on SDS gels (Ringstad et al, 2001) such that some of it may

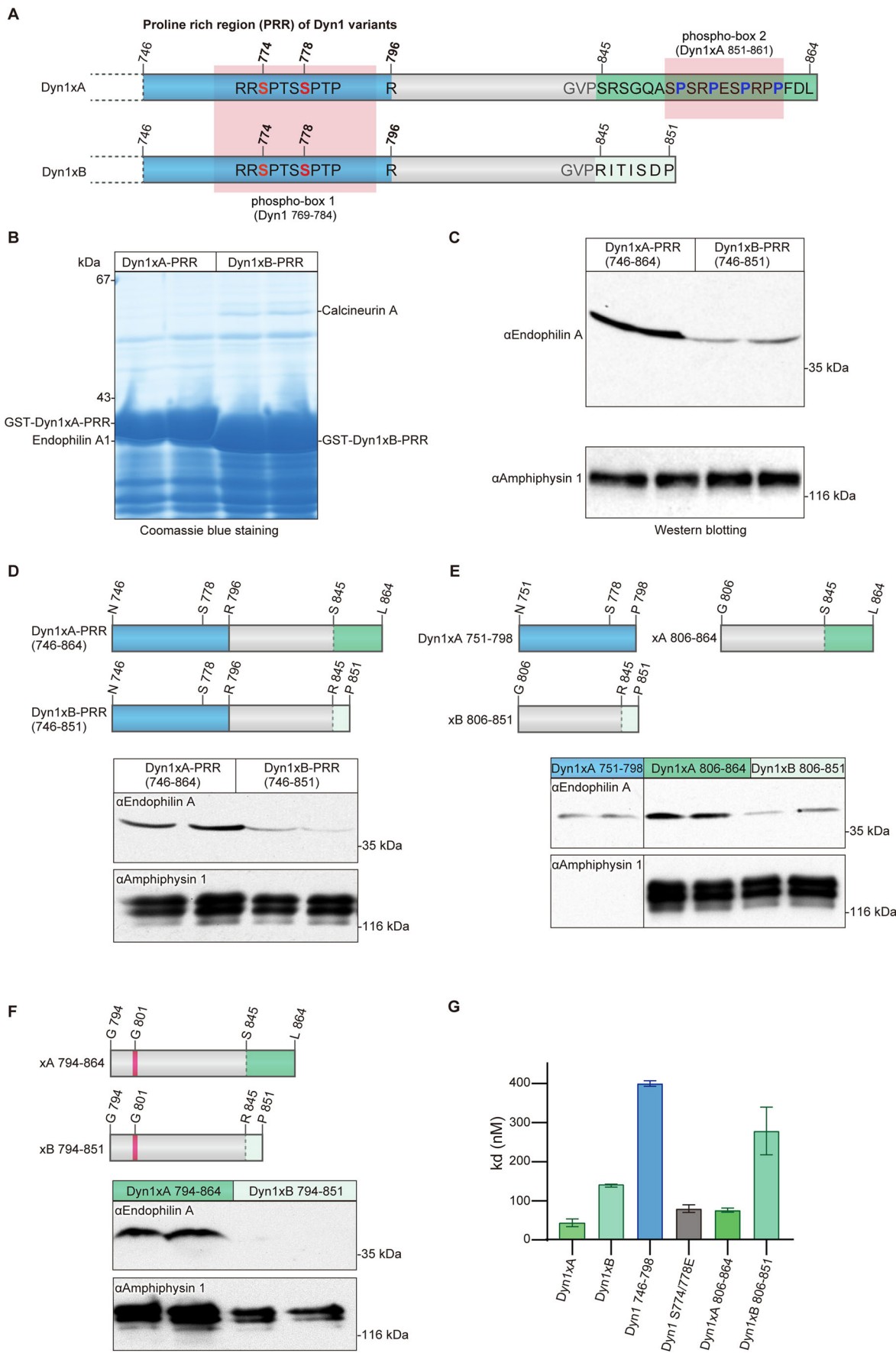

**Figure 1.  Endophilin A specifically associates with long-tail Dyn1xA.**

(A) Schematic diagram represents the protein structural elements in proline rich region (PRR) of the Dyn1 splice variants. Dyn1xA and xB share identical amino acid sequence up to Pro844. Blue color shows previously reported Endophilin A binding region. Green color shows 20 amino acid extension of Dyn1xA. Light green color shows calcineurin binding sites specific to Dyn1xB. Ser845 is the splice site boundary. (B) Synaptosomal lysates were incubated with GST-Dyn1-PRR (either xA or xB) coupled to GSH-sepharose beads. Bound proteins were separated by SDS-PAGE and stained with Coomassie blue. The bands marked as Endophilin A were subjected to LC–MS/MS analysis. Samples are from duplicate experiments. Note that the two GST-tagged PRR baits are overloaded and are of different sizes (left vs right panels). (C) Bound proteins from pull-down experiments with GST-Dyn1-PRR (either xA or xB) were subjected to Western blot analysis with antibodies against endophilin A, amphiphysin 1, each being run in duplicate experiments. Results are representative of at least two independent experiments. (D–F) Schematic diagram represents truncated PRR used in pull-down experiments. Dyn1-751-798 contains the previously reported Endophilin A binding region (blue). Dyn1xA-794-864 and Dyn1xB-794-851 contain Glycine 801 (magenta) which has inhibitory role for Endophilin A binding. Corresponding synaptosomal lysates were incubated with truncation constructs (D) coupled to GSH-sepharose beads. Bound proteins were subjected to Western blot analysis with antibodies against Endophilin A and Amphiphysin 1. All conditions are performed on the same blot. Results shown are in duplicate samples from one of at least 3 independent experiments. (G) ELISA assays used to determine the binding affinity, demonstrated that the C-terminus Dyn1xA-806-864 is has the major binding site for Endophilin A. His-tagged mouse Endophilin-SH3 domain was coated onto a 96-well plate and examined for the ability to bind to increasing concentrations of a variety of GST-tagged Dyn1-PRR peptides by an ELISA assay. The binding affinity (Kd) for GST-tagged each dynI-PRD construct was calculated based on two experiments, each containing four replicates. Mean ± SD is shown. Source data are available online for this figure.

have been missed in our analysis of only the 40 kDa band. The identity of Endophilin A1 was independently confirmed by Western blotting with polyclonal antibodies (Fig. 1C). Another Dyn1 binding partner Amphiphysin 1 binds about equally to Dyn1xA-PRR and Dyn1xB-PRR in pull-down experiments (Fig. 1C,E). These results suggest that Endophilin A1 binding may define specificity of the Dyn1xA function.

Previous studies suggest that Endophilin A1 can bind tandem PRMs around phosphobox-1 (Fig. 1A,D), which are present in both Dyn1xA and xB (Anggono and Robinson, 2007) but binds poorly to the region including the sequence specific to xB (G806 to P851; $xB_{806-851}$; Fig. 1D) (Solomaha et al, 2005). Thus, a second binding site for Endophilin A1 likely lies within the C-terminal extension unique to Dyn1xA. To refine the Endophilin binding sites in the Dyn1xA-PRR, we generated the following GST-fusion peptides to separate the common from unique parts of the tails of each splice variant as N- or C-terminal regions of the two PRRs (Fig. 1D,E): Dyn1xA-PRR peptide separating the PRR into two halves: N751 to P798 ($xA_{751-798}$, common to both splice variants), G806 to L864 ($xA_{806-864}$, the unique extended terminus of xA), and Dyn1xB-PRR peptide spanning G806 to P851 ($xB_{806-851}$. the unique xB calcineurin binding terminus). $xA_{751-798}$ contains known binding sites for both Syndapin 1 and Endophilin A1. $xA_{806-864}$ bound Endophilin A at the similar level to Dyn1xA-PRR, but greater than $xA_{751-798}$ and $xB_{806-851}$ (Fig. 1D,E), suggesting that Endophilin A1 preferentially binds to the extended C-terminal tail of Dyn1xA. We generated two additional GST-fusion constructs: Dyn1xA-PRR peptide spanning G794 to L864 ($xA_{794-864}$) and Dyn1xB-PRR peptide spanning G794 to P851 ($xB_{794-851}$) (Fig. 1F). They both contain G801, which has an inhibitory role in Endophilin binding (Luo et al, 2016). The $xA_{794-864}$ construct bound Endophilin A to the same level as the full-length xA-PRR and slightly less than $xA_{806-864}$ (Fig. 1G). By contrast, Endophilin A1 binding to $xB_{794-851}$ was further reduced compared to $xB_{806-851}$ (Fig. 1E,G). Since Dyn1xA-PRR and xB-PRR contain the well-characterized Amphiphysin 1 binding site ($_{833}$PSRPNR$_{838}$) (Grabs et al, 1997), we also tested Amphiphysin 1 binding. Dyn1x 751-798 does not bind Amphiphysin 1 SH3 domain (Fig. 1G), confirming the specificity of binding to the 833-838 motif as reported in previous studies (Solomaha et al, 2005; Grabs et al, 1997) (Fig. 1D–F). Together, these data suggest that Endophilin A1 has a second and unexpected binding site spanning the spliced C-terminal extension of Dyn1xA-PRR, $xA_{806-864}$ region.

Pull-down experiments provide only a crude affinity index since they are equilibrium experiments and are performed in whole tissue lysates in the presence of competing binders. To quantitatively determine the binding affinity, we performed enzyme-linked immunoassay (ELISA) to measure the direct binding affinity of these peptides to the Endophilin A1 SH3 domain (Fig. 1G). The His-tagged mouse Endophilin A1 SH3 domain was coated onto an ELISA microtiter plate and overlaid with various concentrations of GST-tagged $xA_{751-798}$, $xA_{806-864}$, $xA_{746-864}$, and $xB_{746-851}$. The $xA_{746-864}$ recombinant peptide had a nanomolar binding affinity for Endophilin A1 SH3 (Kd = 44 ± 10 nM)—3-fold higher than for $xB_{746-851}$ (Fig. 1G). The $xA_{751-798}$ construct exhibited 9-fold weaker affinity than $xA_{746-864}$. $xA_{806-864}$ yielded less than 2-fold difference from $xA_{746-864}$. Thus, the newly identified Endophilin A1 SH3-binding site in $xA_{806-864}$ has a 5-fold higher affinity than the binding site in $xA_{751-798}$. The results reveal there are two independent Endophilin A binding sites in the Dyn1xA-PRR with differential affinities for Endophilin, with the second novel site being specific to xA and having greater affinity.

## Endophilin A1 interacts with Dyn1xA at multiple sites

The PRR-SH3 interaction can be enhanced by several amino acid 'elements' surrounding the core binding site, including residues that are near or distantly located in the protein sequence (short or long distance elements (SDE or LDE)) (Luo et al, 2016). To characterize how Dyn1xA and Endophilin A interact, we first identified the binding interfaces between these proteins by NMR spectroscopy, employing $^1$H-$^{15}$N Heteronuclear Single Quantum Coherence (HSQC)-CSP experiments. We first probed the Dyn1xA-PRR – Endophilin A1 SH3 interaction interface using the $xA_{751-798}$ peptide to validate our approach, since the binding sites have been mapped previously by mutagenesis studies (Anggono and Robinson, 2007) and truncation experiments (Fig. 1). The sequence in this region is common to both xA and xB variants. Triple resonance spectra were obtained from the $xA_{751-798}$ peptide labeled with $^{13}$C and $^{15}$N isotopes. HSQC and resonance assignments were made for backbone H, N, Cα, and Cβ nuclei in $xA_{751-798}$ from standard triple resonance data (Fig. EV2). Analysis of the chemical shifts showed that $xA_{751-798}$ is disordered in solution. $^{15}$N-labeled $xA_{751-798}$ was incubated with increasing concentrations of unlabeled endophilin SH3 domain (residues D291-H352). CSPs for backbone amides of $xA_{751-798}$ were plotted

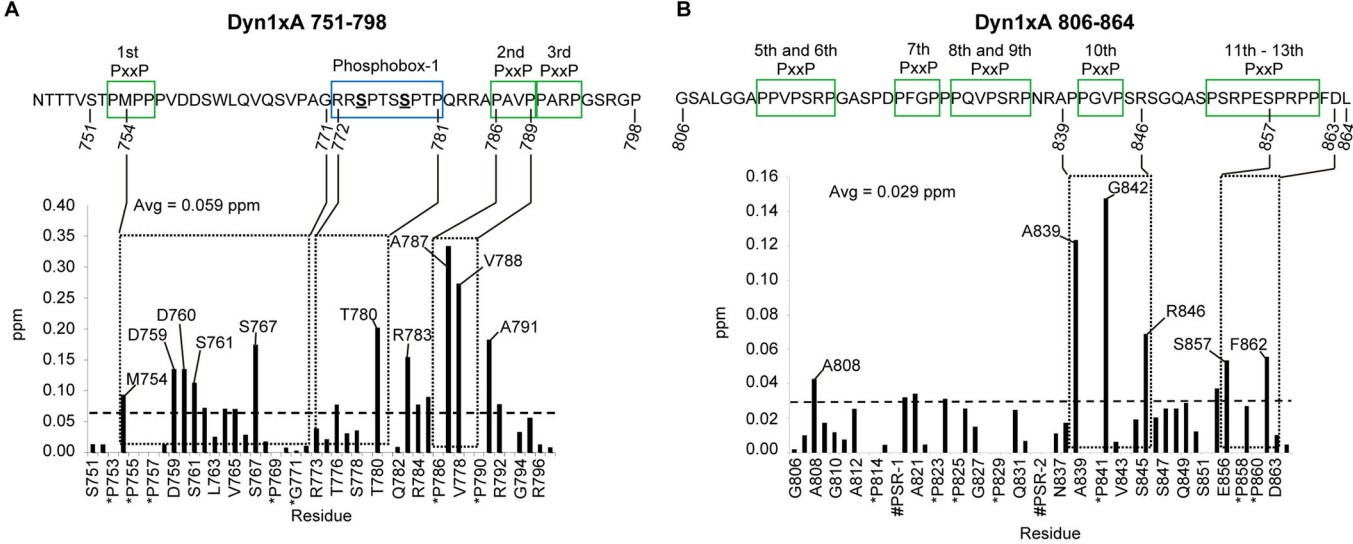

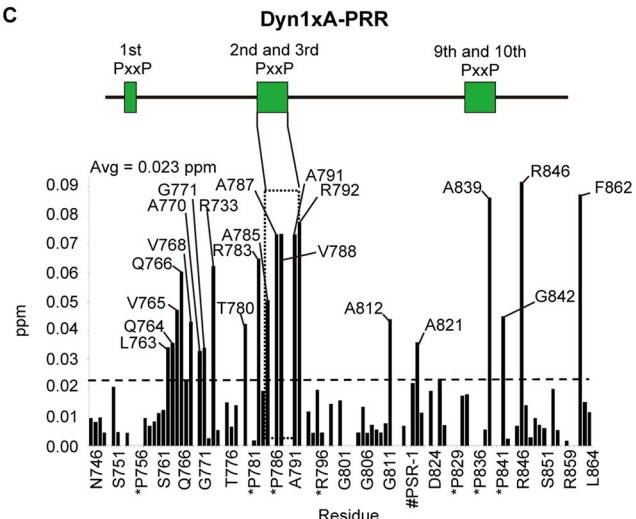

**Figure 2. NMR titrations identify two distinct binding sites for Endophilin A1 SH3 on Dynamin.**

(A, B) Summary of backbone amide chemical shift perturbations (Δδ in ppm) following titration of Endophilin A1 SH3 into a solution of [15]N-labeled Dyn1xA 751-798 (A) or Dyn1xA 806-864 (in B). (C) Summary of the backbone amide chemical shift perturbations (Δδ in ppm) following titration of Endophilin A1 SH3 into a solution of [15]N-labeled GST-tagged Dyn1xA-PRR. PxxP motifs are outlined as green boxes. Phosphobox-1 is outlined as a blue box. Source data are available online for this figure.

against residue number (Fig. 2A shows example for Dyn1xA 751-798 only). Several regions within $xA_{751-798}$ underwent a large CSPs, with the largest perturbations observed between residues 780 and 791. In line with our previous studies, these data suggest that $_{780}$TPQRRAPAVPPARP$_{793}$ is the primary binding site (first binding site, hereafter) for Endophilin in the $xA_{751-798}$ (underlined residues showed higher than average chemical shifts; note that proline cannot be detected with this method). This region is common in xA and xB variants and encompasses half of phosphobox-1, as our previous studies proposed (Anggono and Robinson, 2007).

We then tested how Endophilin A1 SH3 domain is engaged by the second binding site in the C-terminal half of the PRR that includes the xA spliced tail extension using the $xA_{806-864}$ peptide

labeled with [13]C and [15]N isotopes. Backbone assignments were made for a $xA_{806-864}$ peptide using standard triple resonance experiments. Similar to the $xA_{751-798}$ peptide, chemical shift analysis showed that the $xA_{806-864}$ peptide alone exhibited very little secondary structure in solution. Upon titration of [15]N-$xA_{806-864}$ with Endophilin A1 SH3, significant CSPs were observed (Fig. 2B). The greatest CSPs occurred between residues in the A839 to R846 region (within $_{833}$PSRPNRAPPGVPSR$_{846}$; second binding site, hereafter), which contain two PxxP motifs and overlaps the splice site boundary (Fig. 2B). The first half of this site, $_{833}$PSRPNR$_{838}$, is also known to bind Amphiphysin 1 (Solomaha et al, 2005; Grabs et al, 1997). In addition to this region, S857 (in phosphobox-2) and F862 also displayed larger than average CSPs. The detection of CSPs for the latter two residues suggests a form of

long distance element (LDE) (Luo et al, 2016), which contributes to the binding affinity and is located entirely within the sequence unique to the xA splice variant. Therefore, the C-terminal Endophilin binding site for xA$_{806-864}$ appears to adopt a relatively classic SH3 domain binding motif for a Class II binding mode and includes an additional LDE to the binding site, located at the end of the alternatively spliced xA tail extension.

To fully explore the Endophilin A1 SH3-binding interface within Dyn1xA-PRR, we repeated the NMR analysis with full-length Dyn1xA-PRR$_{746-864}$. Assignments of this protein were made from a combination of standard triple resonance NMR experiments and by comparison with the assignments made for xA$_{806-864}$ and xA$_{751-798}$. Figure EV2A shows the extent of assignments that were able to be made. Stepwise titration with endophilin SH3 gave rise to additional signals and increases in signal intensity. These observations suggest that (i) the unbound PRR most likely undergoes conformational dynamics on the microsecond to millisecond timescale and displays a considerable degree of disorder, and (ii) that titration with Endophilin SH3 results in a reduction in conformational dynamics—consistent with the formation of an ordered complex. CSPs were observed broadly across the PRR in several clusters (Fig. 2C). Essentially, these CSPs match the sum of the individual spectra previously obtained by the N- or C-terminal parts of the PRR, xA$_{751-798}$ and xA$_{806-864}$ (Fig. 2A,B), suggesting that the two Endophilin binding sites identified earlier were both being engaged in the context of the full-length Dyn1xA-PRR. The large CSP for A812 and A821 was observed only in xA$_{746-864}$, but not in xA$_{806-864}$, suggesting that the region $_{812}$APPVPSRPGA$_{821}$ may not be an additional binding interface for the second binding site, but rather for the first binding site within Dyn1$_{751-798}$. The large CSP displayed by R846 and F862 from the xA tail sequence appears to fully account for the difference in binding affinity between xA and xB variants. Thus, these data suggest that the second Endophilin A1 SH3-binding site is located near the splice junction that discriminates xA from xB, and that binding affinity is determined by several interfaces along unique sequences of the xA tail extension.

## Dyn1xA R846 specifies Endophilin binding to the second binding site

Despite that the Dyn1xA extension contains three new PRMs, the NMR analysis indicates that $_{833}$PSRPNRAPPGVPSR$_{846}$ region in Dyn1xA may be the second high affinity binding site for Endophilin. Amphiphysin binding is known to overlap this site, which is common to both splice variants (Solomaha et al, 2005; Grabs et al, 1997). To test this possibility, a site-directed mutagenesis approach was used for the underlined amino acids in the xA$_{806-864}$ construct (Fig. 3A). Underlined prolines were mutated into alanine (Fig. 3B,C), expressed as GST fusion proteins and used for pull-down experiments of rat endogenous full-length endophilin from nerve terminals. Double mutations on P813/816A and P816/819A did not affect the binding of either Endophilin or Amphiphysin to the xA$_{806-864}$ peptide (Fig. 3B), suggesting that this site does not act as an additional interface for the second binding site, as predicted. By contrast, P830/833A greatly reduced binding of both Amphiphysin and Endophilin (Fig. 3B). P833/836A abolished Amphiphysin 1 binding, confirming $_{833}$PSRPNR$_{838}$ as the amphiphysin binding site (Solomaha et al, 2005; Grabs et al,

1997). Surprisingly, it also completely abolished Endophilin SH3 binding (Fig. 3B). These results suggest that P833xxP836xR838 motifs are the binding sites shared between Endophilin A and Amphiphysin 1.

We next examined the binding in the full-length Dyn1xA-PRR. Like the xA$_{806-864}$ construct, both P830/833A and P833/836A strongly inhibited the Endophilin binding and completely blocked Amphiphysin binding (Fig. 3C). Both mutants P813/816A and P816/819A had no effect on Amphiphysin binding but reduced Endophilin binding to xA-PRR (Fig. 3C), consistent with the NMR data (Fig. 2B,C). Unlike in xA$_{806-864}$, P841/844A in full-length xA-PRR did not reduce Endophilin binding, although P844A single mutation in xA-PRR tended to reduce both Endophilin and Amphiphysin binding (Fig. 3E). This is similar to our previous study on Syndapin 1 SH3, showing that P844 is an LDE that regulates the affinity of the Syndapin 1-Dyn1 interaction (Luo et al, 2016). To further test that $_{830}$PQVPSRPNR$_{838}$ is the binding core motif, we also mutated R838 to Ala. Western blotting of the bound proteins from the R838A pull-down experiment showed that R838A almost abolished both Endophilin and Amphiphysin binding in xA$_{806-864}$ (Fig. 3D), and reduced Endophilin binding to xA-PRR (Fig. 3E). The latter is presumably because Endophilin has an alternate binding site only in the full-length tail, while Amphiphysin does not. This strongly supports that Endophilin binds the $_{830}$PQVPSRPNR$_{838}$ core motif (its second binding site) which spans either side of the spliced extension of Dyn1xA via the Class II binding mode PxxPxR.

The NMR data displayed strong CSP for A839, G842, R846, and F862 in both constructs (xA$_{806-864}$ and Dyn1xA-PRR, Fig. 2B,C), suggesting that these amino acids comprise elements that increase the Endophilin binding to the core motif. To test this possibility, we designed mutations in both xA$_{806-864}$ and Dyn1xA-PRR: R838A, A839G, G842A, R846A, and F862A. Pull-down experiments revealed that both A839G and R846A almost abolishes Endophilin binding in xA$_{806-864}$, and greatly reduced Amphiphysin binding (Fig. 3D). In the same construct, a single mutation of F862A reduced Endophilin binding by 29% ($p < 0.01$, $n = 4$) (see also Fig. 3D). However, Endophilin binding tended to increase with G842A (Fig. 3D). This confirms the NMR data that A839, G842 and R846 are SDEs, and F862 is a LDE for the high-affinity binding site. However, in the presence of two Endophilin binding sites in Dyn1xA-PRR, the CSPs showed a considerably more complex extent (Fig. 2C), presumably due to competition between two binding sites. Thus mutations R838A and R846A caused smaller reductions in Endophilin binding compared to wild-type Dyn1xA-PRR (Fig. 3E,F, R838A, median 68.2%; Fig. 3G, R846A, median 59.9: R838A reduced the Dyn1/Amphiphysin interaction (Fig. 3E,F, median 14.2% binding compared to wild-type Dyn1xA-PRR), whereas R846A does not significantly change the binding of Syndapin to Dyn1xA-PRR (Fig. 3E,G). Therefore, R846, being part of an SDE, is the only residue we found to specifically regulate the Dyn1 interaction with Endophilin in the context of the full-length tail (Dyn1xA-PRR).

## The Dyn1xA-PRR/Endophilin SH3 binding is one to one

The biochemical experiments revealed that the full-length Dyn1xA-PRR binds endophilin SH3 via two independent binding sites located at either ends of the PRR. In potential conflict with this, the

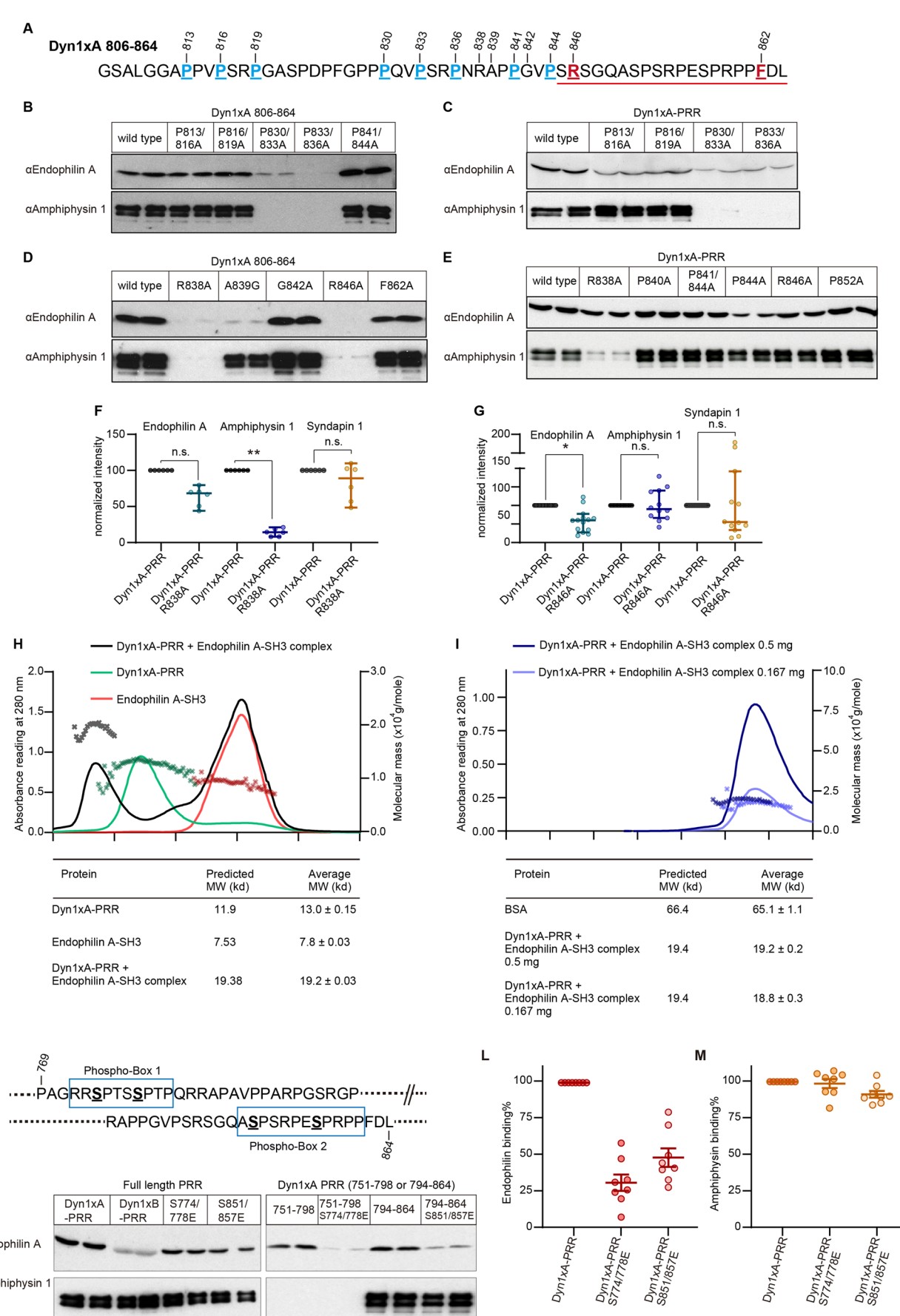

**Figure 3.  The interaction of Endophilin with both binding sites in Dyn1xA-PRR is phosphorylation-dependent in vitro.**

(A) Amino acid sequence of Dyn1xA$_{806-864}$. The amino acids for site-mutagenesis were underlined, and their positions marked with the number. The splice site after Ser845 in Dyn1xA-PRR is indicated with a red line. (B–E) Samples from pull-down experiments from brain lysates with the indicated GST-tagged Dyn1xA$_{806-864}$ or Dyn1xA-PRR and their specific point mutants, were run on gels, blotted and probed with antibodies to Endophilin A or Amphiphysin 1. Results shown are in duplicate samples from one of at least 3 independent experiments. (F) The binding of Endophilin A, Amphiphysin 1 and Syndapin 1 to Dyn1xA-PRR (wild type) or R838A mutant quantified from Western blots in (E). $n = 14$ (6 experiments with 2–4 replicates in each). Median and 95% confidential intervals are shown. Kruskal–Wallis with Dunn's multiple comparisons test was applied (**$p < 0.001$). (G) The binding of Endophilin A, Amphiphysin 1 and Syndapin 1 to Dyn1xA-PRR (wild type) or R846A mutant quantified from Western blots in (E). $n = 14$ (6 experiments with 2–4 replicates in each). Median and 95% confidential intervals are shown. Kruskal–Wallis with Dunn's multiple comparisons test was applied (*$p < 0.05$). (H) SEC-MALS profiles for Dyn1xA alone (in green), Endophilin A SH3 alone (in red) and the complex of the two (in black) are plotted. The x-axis shows retention time. The left axis is the corresponding UV absorbance (280 nm) signals in solid lines, and the right axis shows the molar mass of each peak in crosses. The molecular weight of the complex was determined and tabulated in comparison with the predicted molecular weight. x represent individual data points. (I) SEC-MALS profiles for a high concentration of Dyn1xA-PRR/Endophilin A SH3 complex (0.5 mg) (in dark blue) and a low concentration of Dyn1xA-PRR/endophilin A SH3 complex (0.167 mg) (in blue). The x-axis shows retention time. The left axis is the corresponding UV absorbance (280 nm) signals in solid lines, and the right axis shows the molar mass of each peak in crosses. The molecular weight of the complex was determined and tabulated in the table. x represent individual data points. (J) A schematic diagram of phosphobox-1 and -2 in Dyn1xA-PRR each containing dual phosphorylated serines. (K) Phosphomimetic mutants were made in the phosphoboxes. Samples from pull-down experiments with GST tagged PRD and both of its phosphomimetic mutants, S774/778E and S851/857E, GST-Dyn1xA 751-798 and Dyn1xA 751-798 S774/778E, GST- Dyn1xA 794-864 and Dyn1xA 794-864 S774/778E, were run on gels, blotted, and probed with antibodies to Endophilin A and Amphiphysin 1. Results are shown from one of three independent experiments. (L, M) The amount of Endophilin A and Amphiphysin 1 bound to full-length Dyn1xA-PRR and its phosphomimetic mutants were quantified by densitometric analysis of the Western blots such as in (K). $n = 8$. All data were expressed as a percent of Dyn1xA-PRR Mean ± SEM. Source data are available online for this figure.

NMR studies with the full-length Dyn1xA-PRR showed a composite pattern of binding, raising the possibility that both endophilin binding sites in dynamin may be simultaneously occupied. The NMR studies suggest at least two possible scenarios: a) the two endophilin SH3 domains might simultaneously bind to a single dynamin PRR, or b) the NMR data may represent a composite of two separate binding modes where endophilin is bound to either one end or the other, and the NMR may represent a mixture of the two. To discriminate between them, the mass of the PRD/endophilin SH3 complex in solution was measured by size exclusion chromatography—multi-angle laser light scattering (SEC-MALS) experiments. The complex had an average molecular weight of $19.2 \pm 0.3$ kDa (Fig. 3H), in close agreement with the predicted molecular weight of 19.4 kDa for a 1:1 complex. To eliminate the possibility of complex clustering in a concentrated sample, the complex was diluted to one-third of its concentration and re-analyzed (Fig. 3I). SEC-MALS on the diluted sample showed $18.8 \pm 0.3$ kDa, the same value as the first result, demonstrating that Dyn1xA-PRR and endophilin SH3 bind 1:1 in solution. We concluded that the composite pattern obtained by NMR reflects two alternative complexes mixed together: i.e., that PRR binds to endophilin SH3 via either binding site, but not to both sites simultaneously. This suggests that endophilin has a 'choice' concerning which binding site to occupy on dynamin.

## Phosphorylation of Dyn1xA-PRR regulates its interaction with Endophilin on both binding sites

We asked whether the binding site selection might be determined by Dyn1 phosphorylation. Dyn1xA-PRR has multiple phosphorylation sites on its PRR (Chan et al, 2010), shown as phosphobox-1 (containing S774 and S778), and phosphobox-2 (with S851 and S857) in Fig. 3J. Phosphorylation of these sites regulates the interaction of Dyn1 with other proteins (Anggono et al, 2006; Tomizawa et al, 2003; Huang et al, 2004; Xie et al, 2012; Slepnev et al, 1998), and both of these sites flank either of the two Endophilin sites identified above. We previously showed that the phosphobox-1 (Fig. 3J) is the SDE for Syndapin 1 binding (Luo

et al, 2016) while phosphobox-2 at the S851 and S857 regulates Dyn1-amphiphysin interaction (Xie et al, 2012). To determine whether endophilin binding is also regulated by Dyn1 phosphorylation, we performed pull-down experiments with full-length GST-Dyn1xA-PRR containing double point mutations, rendering these sites phosphomimetic: S774/778E or S851/857E. Either mutant pair within the full-length Dyn1xA-PRR reduced endophilin binding (Fig. 3K, left gel) by about half (Fig. 3L), without significant effect on amphiphysin binding (Fig. 3M). However, when the full-length Dyn1xA-PRR is cut in half to separate regions containing phosphobox-1 (xA$_{751-798}$) from -2 (xA$_{794-864}$), the two peptides independently bound Endophilin (Fig. 3K, right gel). As an expected control, only the phosphobox-1 (xA$_{751-798}$) peptide lost detectable Amphiphysin binding, as its binding motif is absent, while the phosphobox-2 (xA$_{794-864}$) peptide retained amphiphysin binding (Fig. 3K). By contrast, Amphiphysin binding was not significantly reduced by the phosphomimetic of xA$_{794-864}$ (Fig. 3K). This confirms the presence of two independent binding sites for Endophilin A1 in Dyn1xA-PRR. The phosphomimetic mutations were then individually introduced into either sequence, and both caused a dramatic reduction in Endophilin binding (Fig. 3K). These results show that both phosphoboxes encompass independent binding sites for Endophilin and that both are independently regulated by phosphorylation. Together with the Dyn1xA-PRR/ Endophilin A binding ratio, phosphorylation of Dyn1xA at either of the two phosphoboxes may act as a switch to regulate Endophilin A1 interaction to the other site.

## Endophilin A1 and A2 are enriched at the endocytic zone along with Dyn1xA

Dyn1xA and Endophilin A are both involved in ultrafast endocytosis at synapses, while Dyn1xB is not (Imoto et al, 2022; Watanabe et al, 2018). Our previous studies suggest that Dyn1xA forms condensates and are enriched at endocytic zones, located just outside the active zone. By contrast, Dyn1xB is diffuse in the synapse and axons (Imoto et al, 2022). We propose that the C-terminal extension of Dyn1xA may also accumulate Endophilin

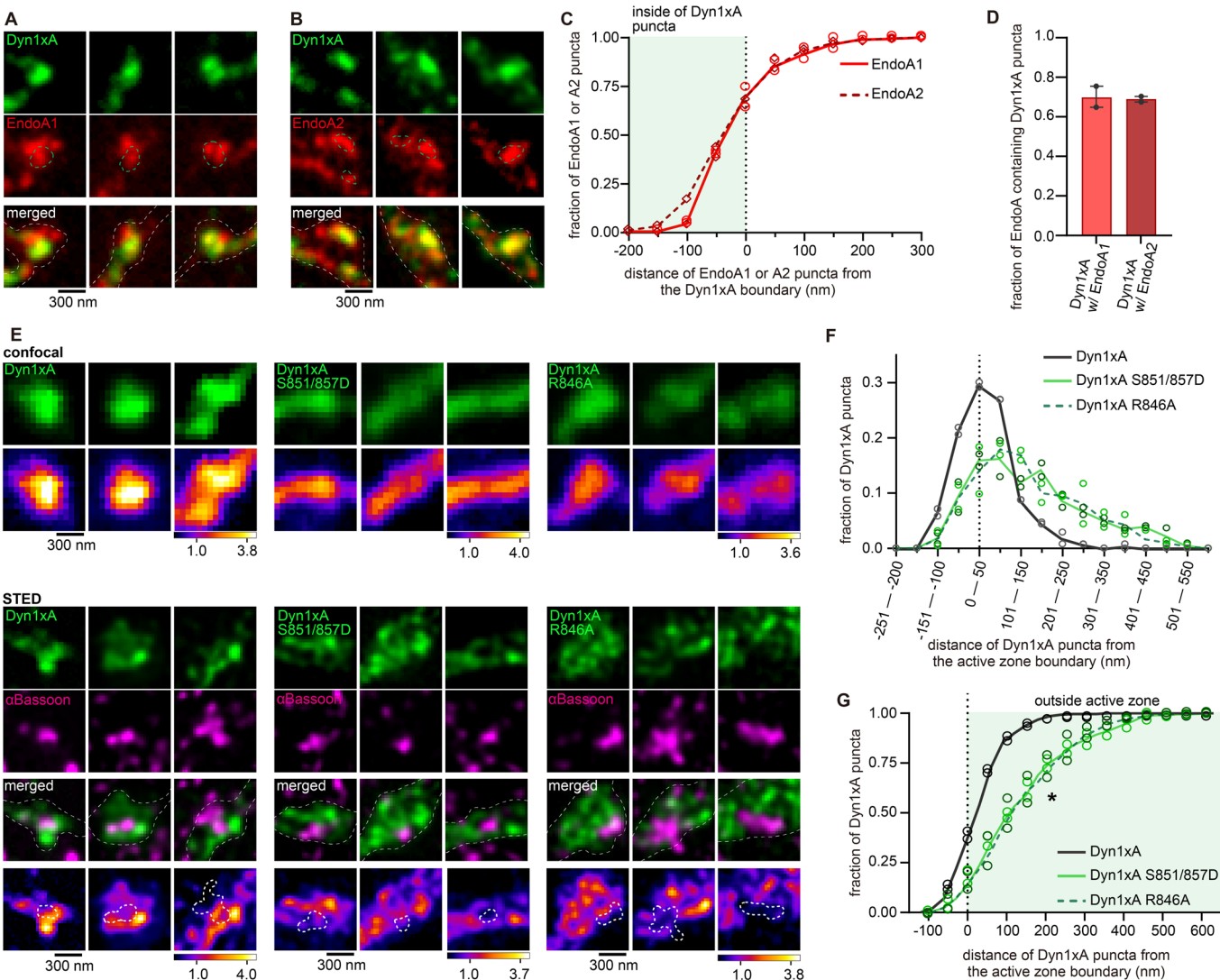

**Figure 4. The C-terminal extension of Dyn1xA is required for its colocalization with Endophilin A1 and A2 at the endocytic zone.**

(A, B) Example STED micrographs showing overexpression of GFP-tagged Dyn1xA (Dyn1xA) and mCherry-tagged Endophilin A1 (EndoA1) (A), mCherry-tagged Dyn1xA and GFP-tagged Endophilin A2 (EndoA2) (B). Green dashed line in Endophilin A1 or A2 images indicates boundary of Dyn1xA clusters defined by MATLAB script (see Methods). White dashed line in merged images indicate neuron shape based on background fluorescence. (C) Cumulative plot representing distance of Endophilin A1 or A2 puncta from the Dyn1xA boundary. Negative values indicate local maxima of Endophilin puncta are inside the boundary of Dyn1xA puncta and positive values indicate outside. (D) Fraction of Dyn1xA puncta contains Endophilin A1 or A2 within the boundary. Data information: data are presented as mean ± SEM. (E) Confocal micrographs (top panel) showing overexpression of GFP-tagged Dyn1xA, Dyn1xA S851D/857D or Dyn1xA R846A. Bottom panels show STED micrographs of the same synapses with an active zone marker Bassoon visualized by antibody. False-colored images show the relative fluorescence intensity of Dyn1xA or mutants. White thick dashed lines within false-colored STED images indicate the boundary of active zone based on the MATLAB analysis of Bassoon signals. Data information: data are presented as median with 95% confidence interval. (F) The distribution of Dyn1xA, Dyn1xA S851/857D and R846A relative to the active zone edge. Negative values indicate local maxima of Dyn1xA or mutants puncta are inside the active zone, and positive values indicate outside. Data information: *$P < 0.05$ (Kolmogorov–Smirnov test). (G) Cumulative plots of (F). More than 150 synapses are examined in each condition. $n > 4$ coverslips from 2 independent cultures. Source data are available online for this figure.

A to endocytic zones. To test this, we overexpressed in primary cultured hippocampal neurons Dyn1xA-GFP along with Endophilin A1-mCherry or A2-mCherry, two Endophilin A isoforms involved in ultrafast endocytosis (Watanabe et al, 2018). Signals of these proteins are acquired by STED microscopy and analyzed by custom-made MATLAB scripts (Appendix Figs. S1, S2). As in our previous study, only one or two puncta of Dyn1xA are present in each bouton. By contrast, Endophilin A1 or A2 formed multiple clusters (1–5 clusters) (Fig. EV3). Approximately 60% of

Endophilin A1 or A2 signals were found within the Dyn1xA puncta (Fig. 4A–C). The distribution analysis suggests that 68.4 ± 5.3% (mean ± SEM) of Dyn1xA puncta contained Endophilin A1, and 66.9 ± 1.6% (mean ± SEM) of Dyn1xA puncta contained Endophilin A2 puncta (Fig. 4D). These results suggest that most of Dyn1xA puncta are colocalized with Endophilin A1 and A2 at the presynaptic endocytic zone.

To investigate whether the C-terminal extension of Dyn1xA is necessary for this localization pattern, we overexpressed Dyn1xA-

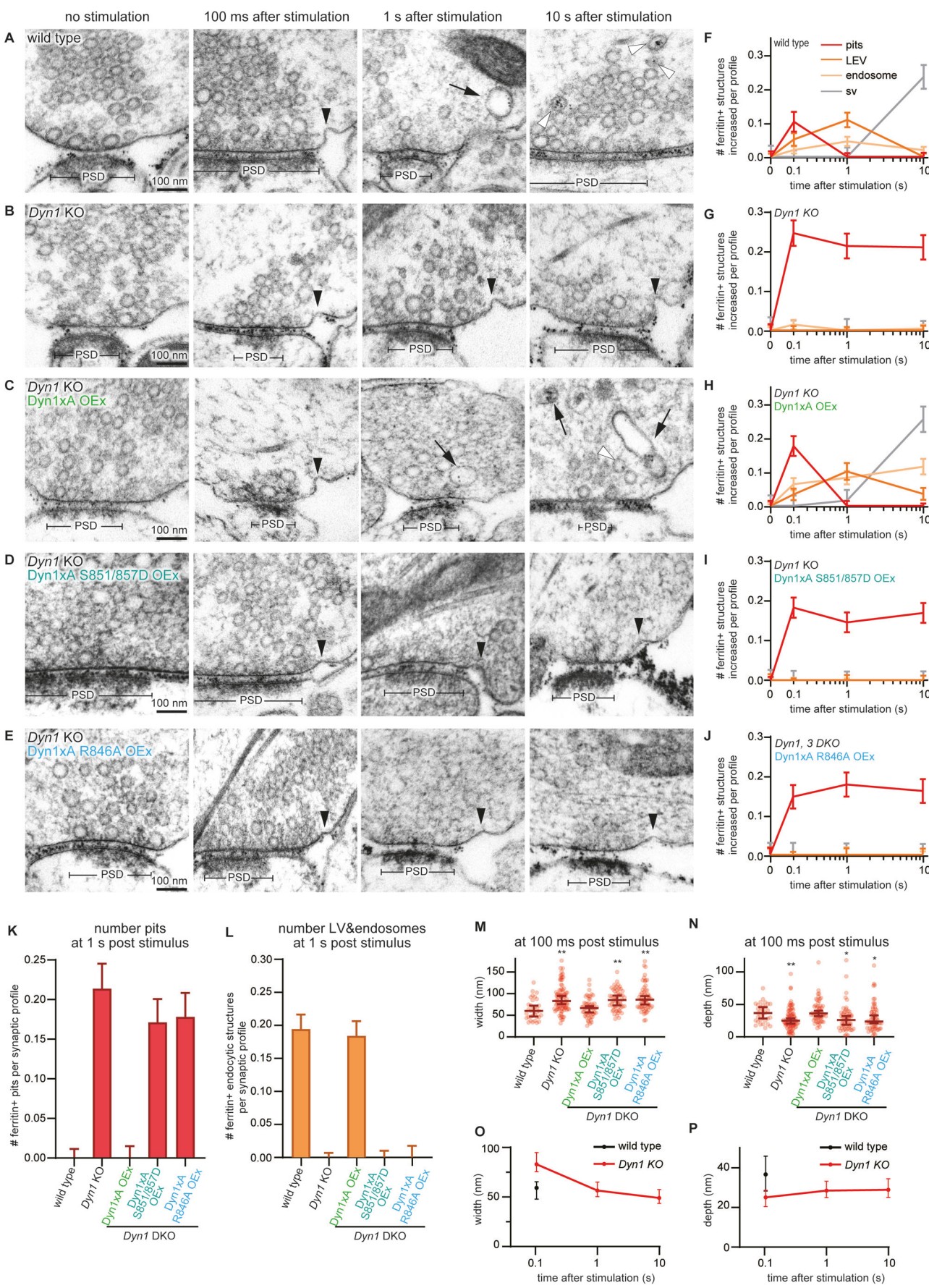

◄

**Figure 5. The Dyn1xA 20 amino acid extension is essential for ultrafast endocytosis.**

(A–E) Example micrographs showing endocytic pits and ferritin-containing endocytic structures at the indicated time points in wild-type primary cultured mice hippocampal neurons (A), *Dyn1* KO neurons (B), *Dyn1* KO neurons, overexpressing Dyn1xA (Dyn1xA OEx) (C), *Dyn1* KO neurons, overexpressing Dyn1xA S851/857D (Dyn1xA S851/857D OEx) (D) and *Dyn1* KO neurons, overexpressing Dyn1xA R846A (Dyn1xA R846A OEx) (E). Black arrowheads, endocytic pits; black arrows, ferritin-positive large endocytic vesicles (LEVs) or endosomes; white arrowheads, ferritin-positive synaptic vesicles. Scale bar: 100 nm. PSD, post-synaptic density. (F–J) Plots showing the increase in the number of each endocytic structure per synaptic profile after a single stimulus in wild-type neurons (F), *Dyn1* KO neurons (G), Dyn1xA OEx neurons (H), Dyn1xA S851/857D OEx neurons (I), Dyn1xA R846A OEx neurons (J). Data information: data are presented as mean ± SEM. (K) Number of endocytic pits at 1 s after stimulation. The numbers are re-plotted as a bar graph from the 1 s time point in (B, D, F, H and J) for easier comparison between groups. Data information: data are presented as mean ± SEM. (L) Number of ferritin-positive LEVs and endosomes at 1 s after stimulation. The numbers of LEVs and endosomes are summed from the data presented in (B, D, F, H and J), averaged, and re-plotted for easier comparison between groups. Data information: data are presented as mean ± SEM. (M, N) Plots showing the width (M) and depth (N) of endocytic pits at the 100 ms time point. The median and 95% confidence interval are shown in each graph. *n* = wild-type, 33 pits, *Dyn1* KO, 87 pits, Dyn1xA OEx, 56 pits, Dyn1xA S851/857D OEx, 55 pits, Dyn1xA R846A OEx, 60 pits. Median and 95% confidential interval is shown. Data information: data are presented as median with 95% confidence interval. *\*p < 0.05. \*\*p < 0.0001 (Kruskal–Wallis Test with Dunn's multiple comparisons tests). (O, P) Plots showing the width (O) and depth (N) changes over the time course in wild-type and *Dyn1* KO neurons. *n* = wild-type, 33 pits at 100 ms, *Dyn1* KO, 87 pits at 100 ms, 59 pits at 1 s, and 62 pits at 10 s. Median and 95% confidential interval is shown. All data are from two independent experiments from *N* = 2 mice primary cultured hippocampal neurons prepared and frozen on different days. *n* = wild-type, 849; *Dyn1* KO, 806; Dyn1xA OEx, 805; Dyn1xA S851/857D OEx, 791; Dyn1xA R846A, 801 synaptic profiles in (B, D, F, H, J, K, and L). See Quantification and Statistical Analysis for the *n* values and detailed numbers for each time point. Knock out neurons are from the mice littermates in all cases. Source data are available online for this figure.

GFP, Dyn1xA-S851/857D-GFP, or Dyn1xA-R846A-GFP in wild-type neurons and visualized the signals relative to an active zone protein, Bassoon, using confocal microscopy and Stimulated Emission Depletion (STED) microscopy (Fig. 4E). We measured distance of Dyn1xA puncta from the active zone edge defined by Bassoon signals using custom-written MATLAB codes, as in our previous study (Imoto et al, 2022). Confocal microscopy showed Dyn1xA-S851/857D-GFP and Dyn1xA-R846A-GFP signals are more diffuse along the axons than Dyn1xA-GFP (Fig. 4E). Analysis of the same synapses by STED microscopy showed that Dyn1xA-GFP signals were found at the edge of an active zone (Fig. 4E–G), a putative site for ultrafast endocytosis (Imoto et al, 2022). By contrast, Dyn1xA-S851/857D-GFP and Dyn1xA-R846A-GFP signals were distributed broadly from the active zone edge (Fig. 4G). These results suggest that the 20 amino acid spliced extension of Dyn1xA is essential for its localization at the endocytic zone along with Endophilin A1 and A2.

## Dephosphorylation and Endophilin binding to the long Dyn1xA tail are essential for ultrafast endocytosis

To explore the functional importance of the new high-affinity binding site at the C-terminal extension of Dyn1xA and the roles of the phosphorylation sites S851/857 present within it, we expressed Dyn1xA-S851/857D and Dyn1-R846A in primary cultured mouse hippocampal neurons lacking Dyn1 (*Dnm1−/−*, *Dyn1* KO) and assayed for ultrafast endocytosis using the flash-and-freeze method (Fig. 5; Appendix Fig. S3) (Imoto et al, 2022; Watanabe et al, 2013a). *Dnm1+/+* littermates served as controls (referred to as wild type). As in the previous studies (Watanabe et al, 2013b, 2014, 2018), ultrafast endocytosis and intracellular trafficking occur normally in wild-type neurons: ultrafast endocytosis at 100 ms, delivery of endocytic vesicles to synaptic endosomes by 1 s, and generation of new synaptic vesicles by 10 s (Fig. 5A,F). By contrast, *Dyn1* KO neurons showed stalled endocytic pits at the plasma membrane (Fig. 5B,G) immediately next to the active zone where ultrafast endocytosis normally takes place, and no fluid phase markers (ferritin particles) were observed in endocytic vesicles and endosomes at 1 s and 10 s (Fig. 5B,G,K,L), suggesting that ultrafast endocytosis failed completely. At 100 ms, endocytic

pits were significantly wider and shallower in *Dyn1* KO than wild type (Dyn1 KO, median width 83.0 nm and depth 25.0 nm, *n* = 88 pits; wild type, median width 59.2 nm and depth 36.7 nm, *n* = 33 pits) (Fig. 5M,N). Endocytic pits in *Dyn1* KO neurons seemed to mature slowly from 100 ms to 1 s—pits became taller and narrower at the base by 1 s (Fig. 5B,O,P), indicating that Dyn1xA may be involved in early pit maturation. However, the neck of endocytic pits did not get constricted further from 1 s onward, suggesting that endocytosis is completely blocked in the *Dyn1* KO neurons.

To test whether these phenotypes are due to the lack of Dyn1xA, we overexpressed wild-type Dyn1xA in *Dyn1* KO neurons. Almost all phenotypes were rescued in these neurons (Fig. 5C,H,K–N). Interestingly, the number of ferritin-positive endosomes did not return to the baseline (Fig. 5E,F), suggesting that other splice variants of Dyn1 may participate in endosomal resolution or overexpression of Dyn1xA causes abnormal endosomal morphology. Critically, when we expressed Dyn1xA-S851/857D or Dyn1xA-R846A, the endocytic defect of *Dyn1* KO was not rescued (Fig. 5D,I,E,J,K–N). Together, these results suggest that the C-terminal extension of Dyn1xA is essential for ultrafast endocytosis by recruiting Endophilin in a dephosphorylation-regulated manner.

## Amphiphysin is not required for ultrafast endocytosis

The C-terminal extension of Dyn1xA provides a higher affinity binding site for Endophilin A. However, this binding site is largely shared with Amphiphysin. Although Dyn1xA-R846 specifically regulates Endophilin A binding (Fig. 3D,E), a potential contribution of Amphiphysin in ultrafast endocytosis cannot be excluded. To test this, we generated shRNA against Amphiphysin 1 (Amphiphysin knock-down or KD, hereafter). Scrambled shRNA was used as a control. Knock-down efficiency was 70–80% (Appendix Fig. S4). The flash-and-freeze experiments suggest that ultrafast endocytosis occurs normally in Amphiphysin KD primary cultured hippocampal neurons (Fig. EV4, Appendix Fig. S5). Ferritin-containing large endocytic vesicles and endosomes accumulated in synaptic terminals by 1 s in Amphiphysin KD neurons, similarly to scramble shRNA neurons. However, ferritin particles were not transferred into synaptic vesicles by 10 s, suggesting a

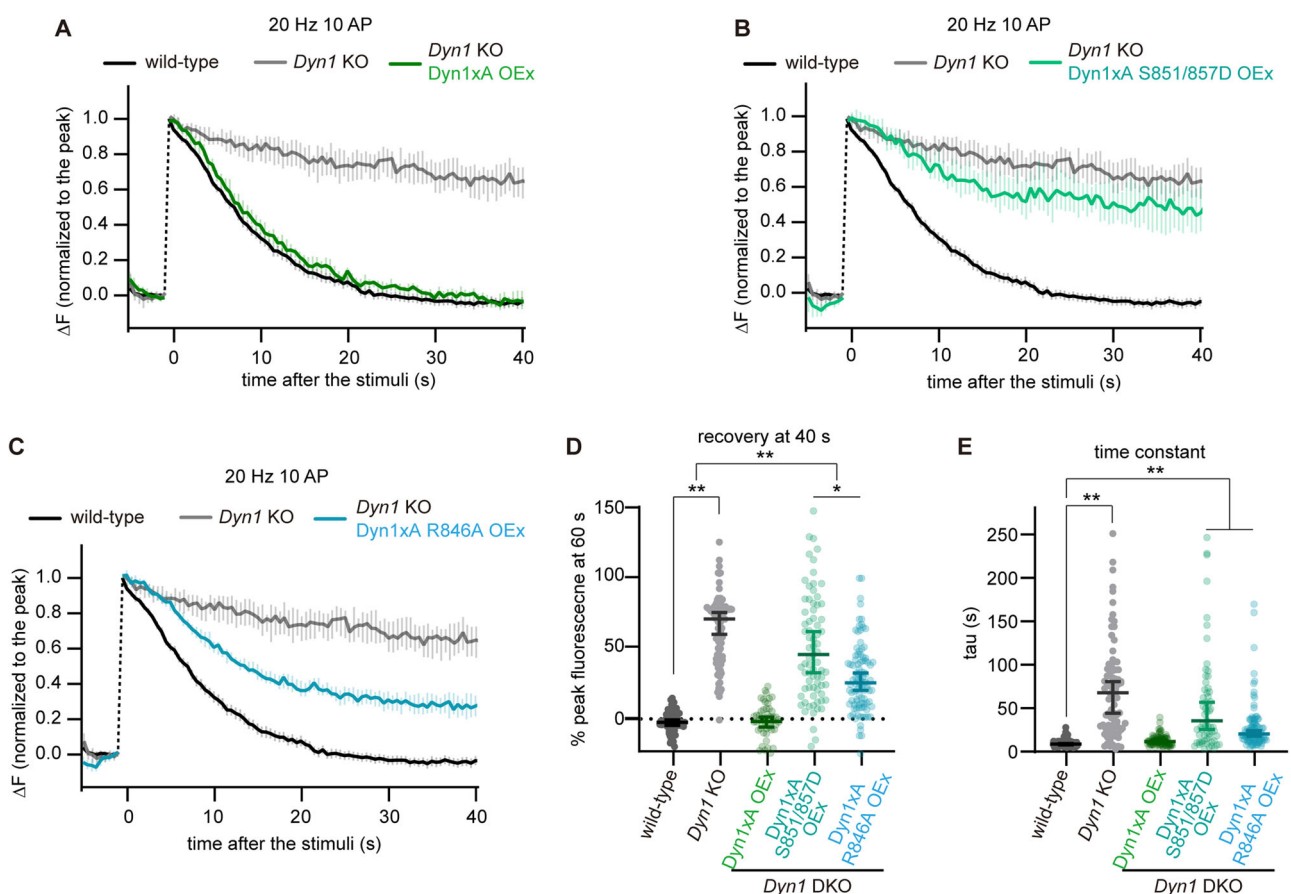

**Figure 6. Dyn1xA and its long tail is important for endocytosis of synaptic vesicle protein.**

(A–C) Plots showing average responses of vesicular glutamate transporter 1 (VGLUT1)-pHluorin in DNM1$^{+/+}$ (wild-type), DNM1$^{-/-}$ (*Dyn1* KO), *Dyn1* KO neurons, overexpressing Dyn1xA (*Dyn1KO* Dyn1xA OEx) (A), *Dyn1* KO neurons, overexpressing Dyn1xA S851/857D (*Dyn1KO* Dyn1xA S851/857D OEx) (B) or *Dyn1* KO neurons, overexpressing Dyn1xA R846A (*Dyn1KO* Dyn1xA R846A OEx). Mouse primary cultured hippocampal neurons were stimulated at 20 Hz, 10 action potentials (AP). The fluorescence signals are normalized to the peak for each bouton. Before stimulation, fluorescence images are acquired for 5 s followed by the stimulation and continued acquisition. Data information: data are presented as median with 95% confidence interval. (D) The percentage of peak fluorescence remaining at 40 s after the beginning of the imaging. Data information: data are presented as median with 95% confidence interval. *$P < 0.05$, *$P < 0.001$ (Kruskal–Wallis test with Dunn's multiple comparisons test). (E) The time constant for fluorescence recovery following 20 Hz, 10 AP. The time constants were obtained by fitting each pHluorin trace to a single exponential decay. The time constant is displayed as Median with 95% confidential interval. Data information: data are presented as median with 95% confidence interval. *$P < 0.05$, *$P < 0.0001$ (Kruskal–Wallis test with Dunn's multiple comparisons test). $n > 60$ presynaptic boutons from five different coverslips in each condition. $N = 2$ culture born from three different mothers at DIV14. *$p < 0.05$. **$p < 0.0001$. Knock out neurons are from the littermates in all cases. Kruskal–Wallis Test with full comparisons by post hoc Dunn's multiple comparisons tests. Source data are available online for this figure.

potential role of Amphiphysin 1 in resolution of endosomes. The results demonstrate that Amphiphysin is not required for ultrafast endocytosis and that the defect observed in the rescue experiments with Dyn1xA S851/857D or R846A (Fig. 5) is likely caused by the lack of both dephosphorylation and of Endophilin A binding.

## The long Dyn1xA tail is important for endocytosis of synaptic vesicle proteins

To test whether the interaction of Dyn1xA with Endophilin A is involved in endocytosis of synaptic vesicle proteins, not just membranes, we assayed recycling of pHluorin-tagged vesicular glutamate transporter 1 (vGlut1-pHluorin) (Balaji and Ryan, 2007; Granseth et al, 2006; Miesenböck et al, 1998; Voglmaier et al, 2006). pHluorin is a pH-sensitive fluorescent protein that becomes

fluorescent upon exocytosis and quenched after reacidification of vesicles following endocytosis. We applied 10 action potentials at 20 Hz—this stimulation protocol also induces ultrafast endocytosis (Watanabe et al, 2014, 2018). Wild-type or Dyn1 KO neurons were cultured and infected with two sets of lentivirus: one carrying vGlut1-pHluorin (Fig. 6A,D,E) and another carrying Dyn1xA, Dyn1xA-S851/857D or Dyn1xA-R846A. Neurons were perfused with extracellular solution containing 4 mM calcium and imaged at 37 °C. With 10 action potentials, fluorescence increased rapidly and decayed to the baseline in wild-type neurons with time constant of 10.5 s (Fig. 6A,D,E). In Dyn1 KO neurons, fluorescence increased but only decayed to 72.0% with a time constant of 68.7 s (Fig. 6A,D,E). These phenotypes were fully rescued when wild-type Dyn1xA is overexpressed in *Dyn1* KO primary cultured mouse hippocampal neurons (time constant = 13.6 s) (Fig. 6A,D,E).

However, overexpression of Dyn1xA-S851/857D or R846A displayed a similar defect in endocytosis albeit less severe (S851/857, time constant = 37.0 s to 46.3% above the baseline; R846A, time constant = 22.1 s to 26.1% above the baseline) (Fig. 6B–E). Similar defects were observed when the experiments were repeated with a single action potential—synaptic vesicle recycling is mediated by ultrafast endocytosis with this stimulation paradigm (Watanabe et al, 2013b) (S851/857 recovery is 73.3% above the baseline; R846A, recovery is 30.0% above the baseline) (Fig. EV5A–D). Together, these results suggest that the 20 amino acid extension of Dyn1xA is important for recycling of synaptic vesicle proteins mediated by specific phosphorylation and Endophilin binding sites within the extension.

## Discussion

Dynamin 1 is a neuron-enriched isoform of the classical dynamins (Ferguson et al, 2007). Two major splice variants, Dyn1xA and xB, are both expressed in neurons in about equal abundance (Chan et al, 2010) and both participate in synaptic vesicle endocytosis (Armbruster et al, 2013; Imoto et al, 2022; Xue et al, 2011). These two variants only differ in protein sequence at the very end of the C-terminus: xA has a 20 unique amino acid extension while xB has 7 residues. Our data identify the specific function and underlying mechanism of the splice variant of Dyn1xA in ultrafast endocytosis via Endophilin A binding. The data highlight the exquisite functional interplay between two dynamin splice variants with unique binding partners that are highly expressed at the synapse in different locations. Overall, we propose a model whereby Dyn1xA mediates formation of endocytic vesicles within 100 ms during ultrafast endocytosis and that Dyn1xB mediates endosome resolution. Although both splice variants bind the same set of proteins, except for calcineurin, only Dyn1xA forms condensates and is predeployed at endocytic sites (Imoto et al, 2022). How such specificity arises was previously unknown. With the combination of advanced biochemical and time-resolved electron microscopy approaches, we show that Endophilin A1 has a second binding site at the extended C-terminus of Dyn1xA with higher affinity than the previously known binding which is common to both Dyn1xA and xB. Mutations in Dyn1xA that specifically disrupt interaction with Endophilin A1 at the Dyn1xA 20 residue extension selectively block ultrafast endocytosis. Thus, the interaction of Endophilin A1 with a newly recognized binding site in Dyn1xA and its regulation by phosphorylation in the same extension determines a selective function for Dyn1xA synapses.

### Dyn1xA binds SH3 domain containing proteins in the region of two different phosphoboxes

We initially proposed that one or more of the three extra PxxP motifs present in Dyn1xA might serve as binding sites for SH3-containing proteins. However, our combination of NMR-CSP and site-directed mutagenesis revealed this be more complex. The Endophilin A binding site was found to overlap with the previously reported site for Amphiphysin 1 SH3 binding, in a sequence that is common to both splice variants. Yet, binding of the latter is not isoform-dependent, while that of Endophilin is. Amphiphysin SH3 was found to bind $_{833}$PSRPNR$_{838}$ in a Class II orientation with

binding not extending as far as the splice site boundary (Solomaha et al, 2005). For Endophilin A1, the Dyn1xA long extension also uses the Class II orientation mode, but in this case both SDEs (including R846) and LDEs (including F862) are located within the Dyn1xA extension, bridging phosphobox-2. This confers both higher affinity Endophilin binding and at the same time renders it sensitive to phosphorylation in the extension, in contrast to that of Amphiphysin 1 (Anggono et al, 2006). A similar binding model was previously reported for Syndapin I SH3 binding to Dyn1, whereby the extended binding elements provide for creation of a high-affinity phosphosensor, bridging phosphobox-1 (Anggono and Robinson, 2007). First, we determined $_{830}$PQVPSRPNR$_{838}$ as the new second and higher affinity binding site for Endophilin in Dyn1xA-PRR. Although this is a previously characterized binding site for Amphiphysin and is also present in Dyn1xB-PRR, the extended C-terminal tail of Dyn1xA contains short and long distance elements (SDE and LDE) essential for Endophilin binding, making it higher affinity for Endophilin. For example, within the unique Dyn1xA extension R846 had large CSP shift in both xA$_{806-864}$ and full-length xA-PRR. Our pull-down results showed that R846A abolished endophilin binding to xA$_{806-864}$ (which contains only the second and higher affinity binding site and the associated SDE (A839) and LDE (F862)) and reduced (Fig. 3G, median 59.3) endophilin binding to the Dyn1xA-PRR (which contains both binding sites) without affecting its interaction with Amphiphysin, providing important partner specificity, although we cannot exclude the possibility that avidity effect may additionally come in play in vivo (Rosendale et al, 2019). Similarly, F862A outside of the PxxP motif also slightly disrupts the endophilin binding. The NMR displayed larger than average CSPs for both S857 and F862 in the extension, illustrating how the full length of the 20 amino acid sequence is specifically used both to increase Endophilin affinity (via R846 and F862) and to introduce phospho-regulation (at least at S857, explaining the inhibition of binding by phosphorylation at 851/857). These results reveal how the Dyn1xA-PRR extension allows a higher affinity interaction with Endophilin, without effect on Amphiphysin.

Binding of SH3 containing proteins to Dyn1xA is phosphorylation-dependent. Seven phosphorylation sites in Dyn1xA-PRR have been well-characterized, with 2 major phosphoboxes including Ser774/778 (phosphobox-1) and Ser851/857 (phosphobox-2) (Graham et al, 2007; Chan et al, 2010; Tan et al, 2003; Larsen et al, 2004). Amphiphysin 1 and Endophilin A were shown to be two potential phosphorylation-dependent Dyn1 binding partners in vitro (Solomaha et al, 2005; Tomizawa et al, 2003; Slepnev et al, 1998). However, a previous study suggests that phosphorylation at phosphobox-1 inhibits the interaction of Dyn1 with Syndapin 1 but does not affect the association with Amphiphysin 1 or 2 (Anggono et al, 2006). Here, our results also showed that the phosphomimetic mutation at phosphobox-2 does not affect Amphiphysin 1 binding, but rather disrupts Endophilin binding. Therefore, the relevant phospho-regulated interaction partners of Dyn1xA are likely Syndapin 1 and Endophilin A.

What are the potential protein kinases regulating Dyn1? The phosphorylation of phosphobox-1 is mediated by glycogen synthase kinase-3 beta (GSK3ß) and cyclin-dependent kinase 5 (CDK5) (Anggono et al, 2006). On the other hand, a potential regulator of phosphobox-2 would be Trisomy 21-linked dual-specificity tyrosine phosphorylation-regulated kinase 1A (Mnb/

Dyrk1) (Fischbach and Heisenberg, 1981; Shindoh et al, 1996). Ser851 in phosphobox-2 is an in vitro substrate for Mnb/Dyrk1 (Huang et al, 2004). Overexpression of Mnb/Dyrk1 in cultured hippocampal neurons slows the retrieval of a synaptic vesicle protein vGlut1 (Kim et al, 2010). Consistently, our data showed that phosphomimetic mutations in phosphobox-2 result in disruption of Dyn1xA localization, inhibition of ultrafast endocytosis and slower kinetics of vGlut1 retrieval. Phospho-regulation of Dyn1xA, Syndapin 1 and Endophilin A1 interaction are likely important elements of the molecular basis of ultrafast endocytosis.

## Dyn1xA accumulates at endocytic zones with Endophilin A

Dyn1xA and Syndapin 1 form molecular condensates at presynaptic terminals and are pre-deployed at endocytic sites (Imoto et al, 2022). This cache of endocytic proteins generates the ultrafast kinetics of endocytosis for synaptic vesicle recycling. Our results here showed that ~80% of Dyn1xA puncta contain Endophilin A1 and A2. Given that there is typically one Dyn1xA punctum at each synapse and ~90% of the puncta contain Syndapin 1 (Imoto et al, 2022), Endophilin A is also likely coalesced at endocytic sites. Consistent with this notion, Dyn1xA forms aggregates in the absence of Syndapin 1 (Imoto et al, 2022). Likewise, when phosphobox-1 or 2 are mutated to disrupt Syndapin 1 or Endophilin A binding, respectively, Dyn1xA also forms aggregates (Imoto et al, 2022) (Appendix Fig. S6). How these proteins form condensates is not clear, since Dyn1 binds either Syndapin 1 or Endophilin A at 1:1—binding to one occludes the other (Anggono and Robinson, 2007). Moreover, the concentrations of Syndapin 1 and Endophilin A exceed the Dyn1xA concentration at synapses (Wilhelm et al, 2014; Imoto et al, 2022). However, there are other endocytic proteins present at synapses, many of which have been shown to directly interact with these proteins through the SH3-PRM interaction. For example, Synaptojanin 1, a phosphoinositide phosphatase, contains a PRR that interacts with the SH3 domain of Endophilin A and functions with Endophilin A to coordinately mediate the neck formation during endocytosis (Watanabe et al, 2018). Similarly, Amphiphysin directly binds with the SH3 domain of Endophilin (Micheva et al, 1997). Thus, only a subset of Endophilin A may have a direct interaction with Dyn1xA at endocytic sites. Like in active zone proteins during the development of *Caenorhabditis elegans* neurons (McDonald et al, 2020), the liquid property of Dyn1xA condensates may allow other endocytic proteins to co-accumulate at their functional sites.

Dynamin typically mediates fission of endocytic vesicles (Cocucci et al, 2014; Taylor et al, 2011). Our results are consistent with this notion, given that endocytic pits are stuck on the plasma membrane. However, the base of these stalled endocytic pits is wide-open, similar to those pits found in Endophilin A triple knock-out, Synaptojanin 1 knock-out, and Syndapin 1 knock-down neurons (Watanabe et al, 2018). In addition, the initial formation of endocytic pits is slowed in Dyn1 KO neurons, suggesting that Dyn1 may have a role in the maturation of endocytic pits in addition to vesicle fission. This scenario is consistent with previous studies demonstrating that Dyn1 has potent curvature generating activity in vitro (Liu et al, 2011) and is involved in the early stage of clathrin-mediated endocytosis (Bhave et al, 2020; Srinivasan et al, 2018). However, it is equally possible that in the absence of Dyn1, endocytic proteins like Endophilin A and Synaptojanin 1 may be diffusely localized and actively recruited to endocytic sites only *after* the initiation of ultrafast endocytosis, like in clathrin-mediated endocytosis. Hence, endocytic pits may mature slowly in Dyn1 KO neurons. Further studies are necessary to distinguish between these possibilities. Nonetheless, our data suggest that the long C-terminal extension of Dyn1xA provides several amino acids spread across its length that serve as a platform to produce a high affinity, phospho-regulated, interaction with Endophilin A1 that defines their relationship and function at synapses (Fig. 7).

# Methods

## Reagents and tools table

| List of plasmid, primer and shRNA constructs information newly used in this study | | | | | | |
|---|---|---|---|---|---|---|
| | Template sequence for insert | Backbone plasmid | Primers used for insert | Primers used for backbone | Restriction enzymes | Cloning strategy |
| f(syn)NLS-RFP-P2A-hsDyn1xA R846A | NA | f(syn)NLS-RFP-P2A-hsDyn1xA (Imoto et al https://doi.org/10.1016/j.neuron.2022.06.010) | NA | 5'-CCAGCCGTT CGGGTCA GGCAAGTCCA TC-3' 5'-GACCCGAA CGGCTGG GGACCCCGG-3' | NA | In-Fusion HD cloning Kit |
| f(syn)NLS-RFP-P2A-hsDyn1xA S851/857D | NA | f(syn)NLS-RFP-P2A-hsDyn1xA (Imoto et al https://doi.org/10.1016/j.neuron.2022.06.010) | NA | 5'-CCTAGCAGAC CAGAAGATCCA CGACCCCCCTTC GACCTCTAAGG-3' 5'-TTCTGGTCTGC TAGGATCTGCC TGACCCGATCG GCTG-3' | NA | In-Fusion HD cloning Kit |
| phsDyn1xA R846A- EGFP-N1 | NA | phsDyn1xA-EGFP-N1 (Kong et al https://doi.org/10.1038/s41586-018-0378-6) Addgene #120313 | NA | 5'-CCAGCCGTTCG GGTCAGGCAA GTCCATC-3' 5'-GACCCGAACG GCTGGGGACC CCGG-3' | NA | In-Fusion HD cloning Kit |
| phsDyn1xA S851/857D-EGFP-N1 | NA | f(syn)NLS-RFP-P2A | NA | 5'-CCTAGCAGACCA GAAGATCCAC GACCCCCCTTCGAC AAGCTTC-3' 5'-TTCTGGTCTGCTA GGATCTGCCTG ACCCGATCGGCTG-3' | NA | In-Fusion HD cloning Kit |

**List of plasmid, primer and shRNA constructs information newly used in this study**

| | Template sequence for insert | Backbone plasmid | Primers used for insert | Primers used for backbone | Restriction enzymes | Cloning strategy |
|---|---|---|---|---|---|---|
| Endophilin A1-eGFP-N1 | Gift from Mike Cousin's Lab | pEGFP-N1 (Ogunmowo et al https://doi.org/10.1101/2023.08.22.554276) | 5'-GACTCAGA TCCTGCCA TGTCGGTGG CAGGGCTG-3' 5'-CAGAATT CGAAGCTTA TGGGGCAGA GCAACCAG-3' | 5'-AAGCTTCGAATTC TGCAGTCG-3' 5'-GGCAGGATCTGA GTCCGG-3' | NA | In-Fusion HD cloning Kit |
| Endophilin A1-mCherry-N1 | pmCherry-CLC addgene #27680 | pEndophilin A1-eGFP-N1 | 5'-CCACCGG TCGCCACCAT GGTGAGCAA GGGCGAGG-3' 5'-GTCGCGGC CGCTTTACTT GTACAGCTCG TCCATGCC-3' | 5'-TAAAGCGGCCGC GACTCTAG-3' 5'-GGTGGCGACCGG TGGATC-3' | NA | In-Fusion HD cloning Kit |
| Endophilin A2-EGFP-N1 | Endophilin II Full Addgene #47409 | pEGFP-N1 | 5'-CCTGCCAT GTCGGTGGC GGGGC-3' 5'-GGTGCCTC TGCCTCAGA AGCTTCG-3' | 5'-GCCTCAGAAGCTT CGAATTCTGCAG-3' 5'-CGGACTCAGATCCTG CCATGTCGGTG-3' | NA | In-Fusion HD cloning Kit |
| shRNA Amphiphysin 1 #1 | (Millipore Sigma TRCN0000093273) | pLKO.1 puro | NA | NA | NA | NA |
| shRNA Amphiphysin 1 #2 | (Millipore Sigma TRCN0000093270) | pLKO.1 puro | NA | NA | NA | NA |
| shRNA Amphiphysin 1 #3 | Millipore Sigma TRCN0000380846 | pLKO.1 puro | NA | NA | NA | NA |
| Scramble or nontargeting | GATCCCTTCGCACCCTAC TTCGTGGttcaagaga CCACGAAGTAGGGTG CGAATTTTTGGAAATTAAT | pFUGW | NA | NA | NA | BamH1 and PacI sites insertion using TAKARA solution I |
| GST-dyn1xA-PRR | Rat sequence | pGEX-4T-1 | 5'-CGGCGAAT TCAACACGA CCACCGTCAG CACGCCC-3' 5'-CTGCAGAA TTGCGGCCG CTTAGAGGTCG AAGGGG-3' | 5'-GGGCTGGCAAGCC ACGTTTGGTG-3' 5'-CCGGGAGCTGCA TGTGTCAGAGG-3' | NA | NA |
| GST-dyn1xB-PRR | Rat sequence | pGEX-4T-1 | 5'-CGGCGAAT TCAACACGA CCACCGTCAG CACGCCC-3' 5'-CTGCAGAAT TGCGGCCG CTTAGGGGTCAC TGATAGTG-3' | 5'-GGGCTGGCAAGCCA CGTTTGGTG-3' 5'-CCGGGAGCTGCATGT GTCAGAGG-3' | NA | NA |
| GST-dyn1xA-PRR 751-798 | NA | pGEX-6P-1 | 5'-CGGGATCCA GCACGCCCA TGCCCCCG-3' 5'-AAAAGGAA AAGCGGCCGC TTAAGGGCCCC GCGACCCGGG-3' | 5'-GGGCTGGCAAGC CACGTTTGGTG-3' 5'-CCGGGAGCTGCAT GTGTCAGAGG-3' | Bam HI, Not I | |
| GST-dyn1xA-PRR 806-864 | NA | pGEX-6P-1 | 5'-GGAATTCGGAT CCGCCCTG-3' 5'-AAAAGGAAA AGCGGCCGCTT AGAGGTC-3' | 5'-GGGCTGGCAAGCC ACGTTTGGTG-3' 5'-CCGGGAGCTGCAT GTGTCAGAGG-3' | Eco RI, Not I | |
| GST-dyn1xB-PRR 806-851 | NA | pGEX-6P-1 | 5'-GGAATTCGGA TCCGCCCTG-3' 5'-AAAAGGAAAAGCGG CCGCTTAGGGGTCAC-3' | 5'-GGGCTGGCAAGC CACGTTTGGTG-3' 5'-CCGGGAGCTGCA TGTGTCAGAGG-3' | Eco RI, Not I | |
| GST-dyn1xA-PRR 806-864 P813/816A | NA | pGEX-6P-1 | 5'-GGGGGGCTGCCCCC GTGGCCTCCAGGCCGGG-3' 5'-CCCGGCCTGGAGGC CACGGGGGCAGCCCCC-3' | 5'-GGGCTGGCAAGC CACGTTTGGTG-3' 5'-CCGGGAGCTGCATGT GTCAGAGG-3' | NA | Site Directed Mutagenesis with Stratagene Pfu Turbo |
| GST-dyn1xA-PRR 806-864 P816/819A | NA | pGEX-6P-1 | 5'-GCTCCCCCCGTGGCCTCC AGGGCGGGGCTTCCCC-3' 5'-GGGGAAGCCCCCGCCCT GGAGGCCACGGGGGGAGC-3' | 5'-GGGCTGGCAAGC CACGTTTGGTG-3' 5'-CCGGGAGCTGCATG TGTCAGAGG-3' | NA | Site Directed Mutagenesis with Stratagene Pfu Turbo |
| GST-dyn1xA-PRR 806-864 P830/833A | NA | pGEX-6P-1 | 5'-CTTTGGCCCCCCCTGCCC AGGTGGCCTCGCGCCCCAAC-3' 5'-GTTGGGGCGCGAGGCC ACCTGGGCAGGGGGGGCCAAAG-3' | 5'-GGGCTGGCAAGCCA CGTTTGGTG-3' 5'-CCGGGAGCTGCATGT GTCAGAGG-3' | NA | Site Directed Mutagenesis with Stratagene Pfu Turbo |
| GST-dyn1xA-PRR 806-864 P833/836A | NA | pGEX-6P-1 | 5'-CTCCCCAGGTGGCCTC GCGCGCCAACCGCGCCCCG-3' 5'-CGGGGCGCGGTTGGCGC GCGAGGCCACCTGGGGAG-3' | 5'-GGGCTGGCAAGCCA CGTTTGGTG-3' 5'-CCGGGAGCTGCATG TGTCAGAGG-3' | NA | Site Directed Mutagenesis with Stratagene Pfu Turbo |

**List of plasmid, primer and shRNA constructs information newly used in this study**

| | Template sequence for insert | Backbone plasmid | Primers used for insert | Primers used for backbone | Restriction enzymes | Cloning strategy |
|---|---|---|---|---|---|---|
| GST-dyn1xA-PRR 806-864 P841/844A | GST-dyn1xA-PRR 806-864 P844A was used as a template | pGEX-6P-1 | 5'-CCCAACCGCGCCCCGGC TGGGGTCGCCAGC-3' 5'-GCTGGCGACCCCAGCC GGGGCGCGGTTGGG-3' | 5'-GGGCTGGCAAGCCA CGTTTGGTG-3' 5'-CCGGGAGCTGCATG TGTCAGAGG-3' | NA | Site Directed Mutagenesis with Stratagene Pfu Turbo |
| GST-dyn1xA-PRR P813/816A | NA | pGEX-6P-1 | 5'-GGGGGGCTGCCCCCGT GGCCTCCAGGCCGGG-3' 5'-CCCGGCCTGGAGGCCAC GGGGGCAGCCCCCC-3' | 5'-GGGCTGGCAAGCCAC GTTTGGTG-3' 5'-CCGGGAGCTGCAT GTGTCAGAGG-3' | NA | Site Directed Mutagenesis with Stratagene Pfu Turbo |
| GST-dyn1xA-PRR P816/819A | NA | pGEX-6P-1 | 5'-GCTCCCCCGTGGCCT CCAGGGCGGGGGCTTCCCC-3' 5'-GGGGAAGCCCCCGCCCT GGAGGCCACGGGGGGAGC-3' | 5-GGGCTGGCAAGCCAC GTTTGGTG-3' 5'-CCGGGAGCTGCATG TGTCAGAGG-3' | NA | Site Directed Mutagenesis with Stratagene Pfu Turbo |
| GST-dyn1xA-PRR P830/833A | NA | pGEX-6P-1 | 5'-CTTTGGCCCCCCTGCC CAGGTGGCCTCGCGCCCCAAC-3' 5'-GTTGGGGCGCGAGGCC ACCTGGGCAGGGGGGCCAAAG-3' | 5'-GGGCTGGCAAGCCA CGTTTGGTG-3' 5'-CCGGGAGCTGCATGT GTCAGAGG-3' | NA | Site Directed Mutagenesis with Stratagene Pfu Turbo |
| GST-dyn1xA-PRR P833/836A | NA | pGEX-6P-1 | 5'-CTCCCCAGGTGGCCTCGC GCGCCAACCGCGCCCCG-3' 5'-CGGGGCGCGGTTGGCG CGCGAGGCCACCTGGGGAG-3' | 5'-GGGCTGGCAAGCCA CGTTTGGTG-3' 5'-CCGGGAGCTGCATG TGTCAGAGG-3' | NA | Site Directed Mutagenesis with Stratagene Pfu Turbo |
| GST-dyn1xA-PRR 806-864 R838A | NA | pGEX-6P-1 | 5'-CCCTCGCGCCCCAACG CCGCCCCGCCTGGGGTC-3' 5'-GACCCCAGGCGGGGC GGCGTTGGGGCGCGAGGG-3' | 5'-GGGCTGGCAAGCCA CGTTTGGTG-3' 5'-CCGGGAGCTGCATG TGTCAGAGG-3' | NA | Site Directed Mutagenesis with Stratagene Pfu Turbo |
| GST-dyn1xA-PRR 806-864 A839G | NA | pGEX-6P-1 | 5'-CGCCCCAACCGCGG CCCGCCTGGGGTCCCC-3' 5'-GGGGACCCCAGGCG GGCCGCGGTTGGGGCG-3' | 5'-GGGCTGGCAAGCCA CGTTTGGTG-3' 5'-CCGGGAGCTGCATGT GTCAGAGG-3' | NA | Site Directed Mutagenesis with Stratagene Pfu Turbo |
| GST-dyn1xA-PRR 806-864 G842A | NA | pGEX-6P-1 | 5'-CAACCGCGCCCCGCC TGCGGTCCCCAGCCGATCG-3' 5'-CGATCGGCTGGGGACCG CAGGCGGGGCGCGGTTG-3' | 5'-GGGCTGGCAAGCCAC GTTTGGTG-3' 5'-CCGGGAGCTGCATG TGTCAGAGG-3' | NA | Site Directed Mutagenesis with Stratagene Pfu Turbo |
| GST-dyn1xA-PRR 806-864 R846A | NA | pGEX-6P-1 | 5'-CCTGGGGTCCCCAGC GCATCGGGTCAGGCAAGTC-3' 5'-GACTTGCCTGACCCGAT GCGCTGGGGACCCCAGG-3' | 5'-GGGCTGGCAAGCCAC GTTTGGTG-3' 5'-CCGGGAGCTGCATG TGTCAGAGG-3' | NA | Site Directed Mutagenesis with Stratagene Pfu Turbo |
| GST-dyn1xA-PRR 806-864 F862A | NA | pGEX-6P-1 | 5'-GAGCCCCAGGCCCCCCG CCGACCTCTAAGCGGCC-3' 5'-GGCCGCTTAGAGGTCGG CGGGGGGGCCTGGGGCTC-3' | 5'-GGGCTGGCAAGCCAC GTTTGGTG-3' 5'-CCGGGAGCTGCATGT GTCAGAGG-3' | NA | Site Directed Mutagenesis with Stratagene Pfu Turbo |
| GST-dyn1xA-PRR R838A | NA | pGEX-6P-1 | 5'-CCCTCGCGCCCCAACG CCGCCCCGCCTGGGGTC-3' 5'-GACCCCAGGCGGGGCG GCGTTGGGGCGCGAGGG-3' | 5'-GGGCTGGCAAGCCAC GTTTGGTG-3' 5'-CCGGGAGCTGCATG TGTCAGAGG-3' | NA | Site Directed Mutagenesis with Stratagene Pfu Turbo |
| GST-dyn1xA-PRR P840A | NA | pGEX-6P-1 | 5'-CGCCCCAACCGCGCCGC GCCTGGGGTCCCCAGC-3' 5'-GCTGGGGACCCCAGGC GCGGCGCGGTTGGGGCG-3' | 5'-GGGCTGGCAAGC CACGTTTGGTG-3' 5'-CCGGGAGCTGCA TGTGTCAGAGG-3' | NA | Site Directed Mutagenesis with Stratagene Pfu Turbo |
| GST-dyn1xA-PRR P841/844A | GST-dyn1xA-PRR P844A was used as a template | pGEX-6P-1 | 5'-CCCAACCGCGCCCCGG CTGGGGTCGCCAGC-3' 5'-GCTGGCGACCCCAGCCG GGGCGCGGTTGGG-3' | 5'-GGGCTGGCAAG CCACGTTTGGTG-3' 5'-CCGGGAGCTGCA TGTGTCAGAGG-3' | NA | Site Directed Mutagenesis with Stratagene Pfu Turbo |
| GST-dyn1xA-PRR P844A | NA | pGEX-6P-1 | 5'-GCCCCGCCTGGGGTCGC AAGCCGATCGGGTCAG-3' 5'-CTGACCCGATCGGCTTG CGACCCCAGGCGGGGC-3' | 5'-GGGCTGGCAAGCCA CGTTTGGTG-3' 5'-CCGGGAGCTGCAT GTGTCAGAGG-3' | NA | Site Directed Mutagenesis with Stratagene Pfu Turbo |
| GST-dyn1xA-PRR R846A | NA | pGEX-6P-1 | 5'-CCTGGGGTCCCCAGCG CATCGGGTCAGGCAATGC-3' 5'-GCATTGCCTGACCCGAT GCGCTGGGGACCCCAGG-3' | 5'-GGGCTGGCAAGCC ACGTTTGGTG-3' 5'-CCGGGAGCTGCATG TGTCAGAGG-3' | NA | Site Directed Mutagenesis with Stratagene Pfu Turbo |
| GST-dyn1xA-PRR P852A | NA | pGEX-6P-1 | 5'-CGGGTCAGGCAAGTGCG TCCCGTCCCGAGAGC-3' 5'-GCTCTCGGGACGGGACG CACTTGCCTGACCCG-3' | 5'-GGGCTGGCAAG CCACGTTTGGTG-3' 5'-CCGGGAGCTGCATG TGTCAGAGG-3' | NA | Site Directed Mutagenesis with Stratagene Pfu Turbo |
| GST-dyn1xA-PRR S774/778E | NA | pGEX-6P-1 | 5'-GCCGGACGCAGGGAGCCC ACGTCCGAACCCACGCCGCAG-3' 5'-CTGCGGCGTGGGTTCGG ACGTGGGCTCCCTGCGTCGGC-3' | 5'-GGGCTGGCAAGCCA CGTTTGGTG-3' 5'-CCGGGAGCTGCATG TGTCAGAGG-3' | NA | Site Directed Mutagenesis with Stratagene Pfu Turbo |
| GST-dyn1xA-PRR S851/857E | NA | pGEX-6P-1 | 5'-CGGGTCAGGCAGAACCGTCCCG TCCCGAGGAACCCAGGCCCCCC-3' 5'-GGGGGGCCTGGGTTCCTCGGGA CGGGACGGTTCTGCCTGACCCG-3' | 5'-GGGCTGGCAAGCCA CGTTTGGTG-3' 5'-CCGGGAGCTGCATGT GTCAGAGG-3' | NA | Site Directed Mutagenesis with Stratagene Pfu Turbo |
| GST-dyn1xA-PRR 746-798 S774/778E | NA | pGEX-6P-1 | 5'-GCCGGACGCAGGGAGCCCACGT CCGAACCCACGCCGCAG-3' 5'-CTGCGGCGTGGGTTCGGACGT GGGCTCCCTGCGTCCGGC-3' | 5'-GGGCTGGCAAGCCA CGTTTGGTG-3' 5'-CCGGGAGCTGCAT GTGTCAGAGG-3' | NA | Site Directed Mutagenesis with Stratagene Pfu Turbo |
| GST-dyn1xA-PRR 796-864 S851/857E | NA | pGEX-6P-1 | 5'-CGGGTCAGGCAGAACCGTCCCG TCCCGAGGAACCCAGGCCCCCC-3' 5'-GGGGGGCCTGGGT TCCTCGGGACG GGACGGTTCTGCCTGACCCG-3' | 5'-GGGCTGGCAAGCC ACGTTTGGTG-3' 5'-CCGGGAGCTGCAT GTGTCAGAGG-3' | NA | Site Directed Mutagenesis with Stratagene Pfu Turbo |

| List of plasmid, primer and shRNA constructs information newly used in this study | | | | | | |
|---|---|---|---|---|---|---|
| | Template sequence for insert | Backbone plasmid | Primers used for insert | Primers used for backbone | Restriction enzymes | Cloning strategy |
| GST-dyn1xA-PRR 746-798 S774/778E | NA | pGEX-6P-1 | 5'-GCCGGACGCAGGGAGCCCACGT CCGAACCCACGCCGCAG-3' 5'-CTGCGGCGTGGGTTCGGACGT GGGCTCCCTGCGTCCGGC-3' | 5'-GGGCTGGCAAGCC ACGTTTGGTG-3' 5'-CCGGGAGCTGCATG TGTCAGAGG-3' | NA | Site Directed Mutagenesis with Stratagene Pfu Turbo |
| GST-dyn1xA-PRR 796-864 S851/857E | NA | pGEX-6P-1 | 5'-CGGGTCAGGCAGAACCGTCCCG TCCCGAGGAACCCAGGCCCCCC-3' 5'-GGGGGGCCTGGGTTCCTCGGGA CGGGACGGTTCTGCCTGACCCG-3' | 5'-GGGCTGGCAAGCCAC GTTTGGTG-3' 5'-CCGGGAGCTGCATG TGTCAGAGG-3' | NA | Site Directed Mutagenesis with Stratagene Pfu Turbo |

## Mice and rats

All experiments with mice were performed according to the rules and regulations of the National Institute of Health, USA. Animal protocols were approved by animal care and use committee of the Johns Hopkins University (MO21M375, approved on 11/2/21). Wild-type mice (C57/BL-6N) were obtained from the Charles River. *Dnm1* KO mice were obtained from Dr. Pietro De Camilli. DNM1+/− were bred to obtain *Dnm1+/+* and *Dnm1−/−* animals. Heterozygote animals were not used in this study. The sex of pups was undetermined and therefore used indiscriminately. Mice are maintained in accordance with the Johns Hopkins Animal Care and Use Committee regulations. These include ad libitum access to food and water, sterile environment with enrichment items, weekly cage changes, temperature control at 22 °C, and a 12-h light/12-h dark cycle. Rat brains were harvested from adult male Wistar rats (6–8 weeks old, 250 gm body weight), euthanised by DecapiCone® and guillotine with approval from the Animal Care and Ethics Committee for the Children's Medical Research Institute, Sydney, Australia (project number C116).

## Materials

Leupeptin was from Merck (Darmstadt, Germany). Tissue culture plastics were from Falcon (Franklin Lakes, NJ). All other materials were from Sigma (St. Louis, MO) unless otherwise stated. Antibodies to Endophilin, Amphiphysin I, and Dynamin IIwere from Santa Cruz Biotechnology (Santa Cruz, CA), antibodies to GFP were from Life Technology and secondary antibodies were all from Dako (Denmark).

## Plasmid construction and protein expression

For the protein expression constructs, GST-tagged dyn1xA-PRR plasmid (long tail, rat sequence, amino acids 746–864) and GST-dyn1xB-PRR (short tail, rat, amino acids 746–851) were cloned into pGEX-6P-1 by subcloning the coding region from GFP-tagged full-length Dyn1aA and mCerulean tagged full-length Dyn1aB (Anggono et al, 2006). Point mutations on GST-dyn1xA-PRR were generated using the QuickChange site-directed mutagenesis kit (Stratagene) and were cloned into pGEX-6P-1 vector (GE Healthcare). GST-Endophilin A1-SH3 domain (mouse, amino acids 292–352) was from Peter McPherson (McGill University, Canada). Human full-length Amphiphysin I was PCR amplified from GST-Amphiphysin I plasmid in pGEX-6P-1 (provided by Dr Pietro De Camilli, Yale, USA), and subcloned into the pEGFP-N2 vector with GATEWAY system (Invitrogen, USA). GST constructs were transformed into *Escherichia coli* JM109 by heat shock and expressed recombinant proteins bound to glutathione (GSH)-

sepharose beads were prepared according to the manufacturer's instructions. Production of isotopically labeled $^{15}$N-labeled, or $^{15}$N, $^{13}$C labeled Dyn1xA-PRR and 1xB-PRR, and Endophilin-A1-SH3 for NMR spectroscopy was achieved by expression in shaker flasks with $^{15}$NH$_4$Cl and $^{13}$C-glucose as sole nitrogen and carbon sources, respectively (Marley et al, 2001). The isotopically labeled proteins were purified in the same manner as the non-labeled proteins. For all NMR experiments, recombinant proteins (GST-Dyn1xA, Dyn1xA 751-798, Dyn1xA 806-864, and GST-Endophilin A1-SH3) were proteolytically cleaved to remove the GST tag and were purified by gel filtration.

## Pull-down experiments and immunoblotting

Crude (P2) synaptosomes were prepared from rat brain (Robinson et al, 1993) and lysed in ice-cold buffer containing 20 mM Tris (pH 7.4), 150 mM NaCl, 1% Triton X-100, 1 mM EDTA, 1 mM EGTA, 20 μg/ml leupeptin, 1 mM phenylmethylsulfonyl fluoride (PMSF) and EDTA-free Complete-Protease inhibitor tablets (Roche, Mannheim Germany). The homogenate was centrifuged at 75,600 × *g* for 30 min at 4 °C. The extract was pre-cleared with GSH-sepharose beads for 1 h, and then incubated with the GSH-sepharose beads coated with GST recombinant proteins at 4 °C for 1 h. The beads were isolated and washed extensively with ice-cold lysis buffer and eluted in 2x concentrated SDS sample buffer. Bound proteins were separated on 10% polyacrylamide mini-gels. The proteins were transferred onto a nitrocellulose transfer membrane (PerkinElmer, MA USA) (Towbin et al, 1979). The membranes were blocked in phosphate-buffered saline (PBS), pH 7.4 with 5% skimmed milk for 1 h and washed in a PBS (pH 7.4) containing 0.1% Tween 20 (PBST), then incubated with the first antibody for 2 h and with the second antibody for 1 h in PBST, and washed with PBST buffer. Immunoblotting was performed by chemiluminescent detection (Pierce SuperSignal West Dura).

## Trypsin digestion

Gel bands from the GST-Dyn1xA-PRR or 1xB-PRR pull-downs from rat synaptosomes migrating at 40 kDa (Fig. 1B) were excised from each lane and diced from colloidal Coomassie Blue G-250-stained dried SDS polyacrylamide gels and were de-stained in six changes of 50% acetonitrile (ACN), 50 mM triethylammonium bicarbonate (TEAB, pH 8) for 15 min. Gel bands were dehydrated with 100% ACN, air dried and incubated with 20 μl of Promega Rapid Lys-C/Trypsin mixture (10 ng/μl, ~200 ng) in 50 mM TEAB (pH 8) for 1 h on ice. Excess Lys-C/Trypsin mixture was removed and 20 μl of 50 mM TEAB (pH 8) added for 8 h at 42 °C. The digested peptide solution was removed, and tryptic peptides remaining in the gel plug were extracted using 50 mM TEAB (pH

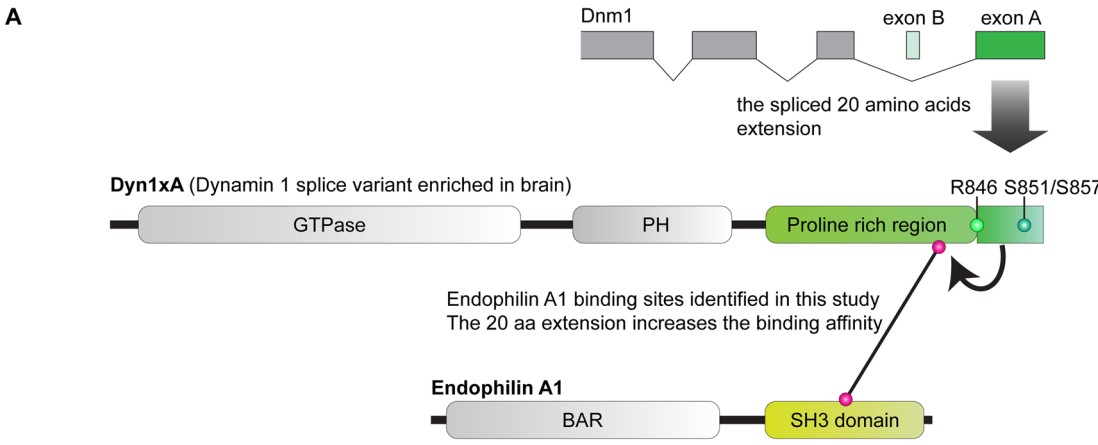

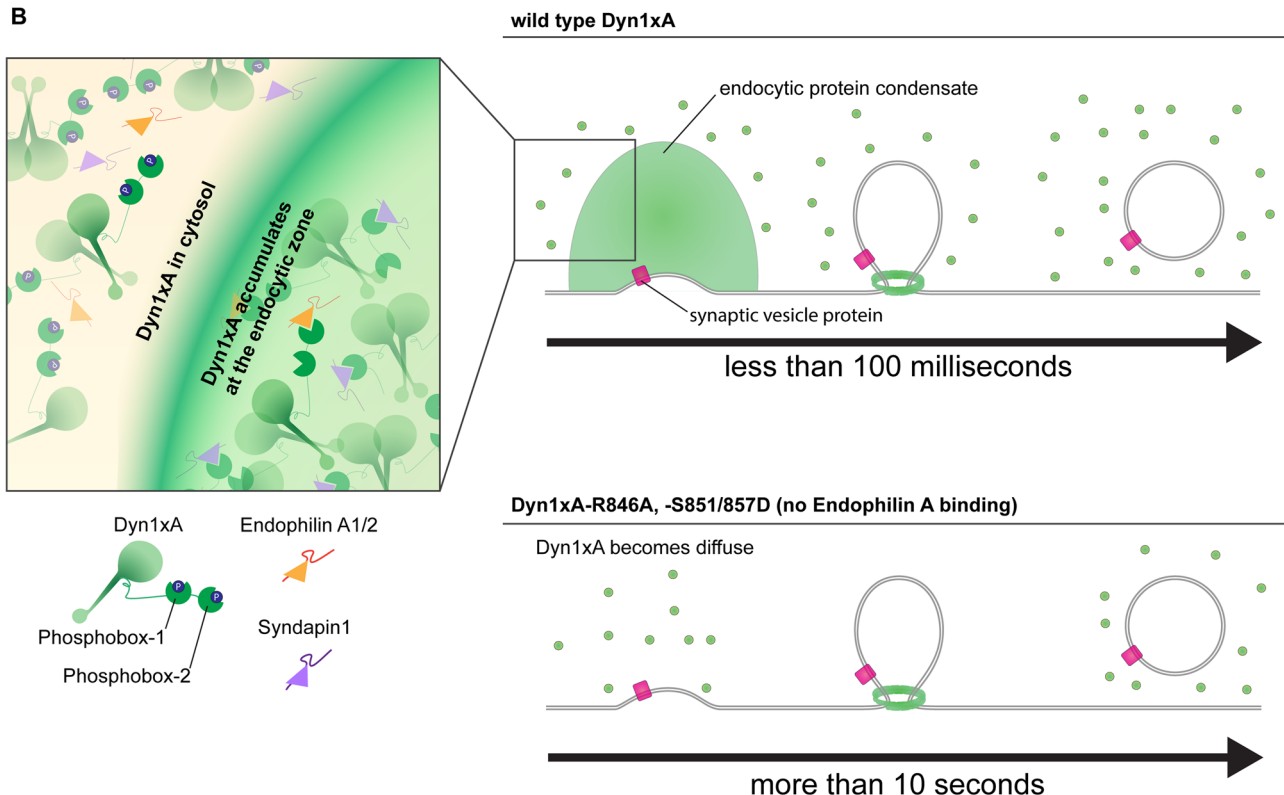

**Figure 7.    Schematics depicting how specific isoforms of Dyn1xA and Endophilin A mediate ultrafast endocytosis.**

A splice variant of Dynamin 1, Dyn1xA, but not other isoforms/variants, can mediate ultrafast endocytosis. (A) Dyn1xA has 20 amino acid extension, containing a newly identified high affinity Endophilin A1 binding site. Three amino acids, R846 at the splice site boundary, S851 and S857, act as long-distance elements to enhance affinity of proline rich motifs (PRM) of Dyn1xA to EndophilinA1-SH3 domain. (B) At a resting state, Dyn1xA accumulates at the endocytic zone with SH3 containing BAR protein Syndapin 1 (Imoto et al, 2022) and Endophilin A1/2. When phosphobox-1 (Syndapin 1 binding) and phosphobox-2 (Endophilin A1/2 binding, around S851/S857) are phosphorylated, Dyn1xA molecules are diffuse within the cytoplasm. Only the dephosphorylated fraction of Dyn1xA molecules can interact with these BAR domain proteins and localize to the endocytic zone. Loss of interactions (i.e., by Dyn1xA-R846A or -S851/857D mutations) disrupts their pre-accumulation at the endocytic zone. Consequently, ultrafast endocytosis fails or slows down.

8) and combined with the digestion solution. A second extraction of the gel plugs was carried out using 100% ACN, a third extraction with 5% formic acid (FA), and the final extraction with 0.5% FA in 90% ACN. The combined solution was concentrated using the GeneVac to completeness, then peptides resuspended in 10 µl of 0.1% FA aqueous solution. Custom-made POROS R2 chromatographic microcolumns were used for desalting and concentration of the peptide mixture prior to mass spectrometry analysis. The

sample was loaded and washed in 0.5% FA. Bound peptides were eluted using 20 µl of 70% ACN, 0.5% FA, dried to completeness, and peptides were resuspended in 12 µl of 0.1% FA and quantified using the Nanodrop system at A280nm.

## Selective reaction monitoring (SRM) LC–MS/MS

To identify and quantify which of the three Endophilin A proteins were present in the excised 40 kDa band from rat brain synaptosomes a Selective Reaction Monitoring (SRM) assay was developed. A Vanquish Neo ultra-high performance liquid chromatographic system operating in nanoflow mode, coupled to a TSQ Altis Plus QQQ mass spectrometer (ThermoFisher Scientific) was used for the SRM analyses. The peptide digests (1.5 pmol) were loaded using direct injection onto an Easy-Spray PepMap Neo analytical column (ES75150PN, 15 cm, 75 mm ID PepMap C18, 100 Å, 2 µm particle size column (ThermoFisher Scientific)). The column was maintained at a temperature of 40 °C. The flow rate during the run was 0.3 µL/min but was raised to 1 µl/min during the column equilibration and loading steps. The gradient started at 3% solvent B (99.9% [v/v] acetonitrile, 0.1% [v/v] formic acid) and increased to 5% solvent over 2 min followed by an increase of solvent B to 15% over 12 min. This was followed by an increase of solvent B to 30% over 12 min, then a further increase to 95% over 6 min. The column was washed with a 5 min linear gradient to 97% solvent B and held for 1 min, followed by a 2 min column equilibration step with 97% solvent A. The LC method duration was 37 min. The TSQ Altis Plus spray voltage was set at 1.7 kV and fragmented at 1.5 mTorr in the second quadrupole. The first quadrupole was set at 0.7 Da FWHM, and the third quadrupole at 0.7 Da FWHM. All transitions were measured with a dwell time of 15 ms. Ion spray voltage was set to 275 °C. The MS method duration was 35 min. Collision energy was set per transition using Skyline software with the "Thermo Altis" CE formula (CE = 0.0339 $m/z$ + 2.3398 for doubly charged precursors and CE = 0.0295 $m/z$ + 1.4831 for precursor charges of three and higher).

The three rat Endophilin proteins (sp|O35179|SH3G2_RAT Endophilin-A1, sp|O35964|SH3G1_RAT Endophilin-A2 sp|O35180|SH3G3_RAT Endophilin-A3) were each in silico-digested with Trypsin [KR|P]. Note that Endophilin A2 has a larger predicted mass than A1 (41492.01 vs 39899.28 Da), and migrates on SDS gels at approximately 45 rather than 40 kDa (Ringstad et al, 2001). This generated a sequence library of all possible peptides (from 6 to 25 amino acids in length) including two missed cleavages, with Carbamidomethyl (C) and Oxidation (M) modifications included using Skyline software (version 22.2.0.351). Precursor charge states of +2, +3, and +4 were included. A Spectral library was generated from importing the MASCOT peptide search results for the data-independent data for the GST-Dynamin1xa/1xb gel band digests that were acquired on the SCIEX ZenoTOF 7600 instrument. A preliminary SRM-transition list for tryptic-peptides originating from Endophilin-A1, Endophilin-A2, and Endophilin-A3 was prepared selecting three to eight dominant Q1/Q3-transitions consisting of doubly, triply and quadruply charged precursor states and y- and b-fragment ions with charge states of 1+ and 2+. The preliminary SRM assay was run using a tryptic digest of endophilin tryptic digest to obtain peptide retention time and to refine the number of transitions monitored. The final SRM transition list included 7 unique peptides from A1 and 6 each from A2 and A3 (Fig. EV1B). The final SRM assay

was re-analyzed using a 5 min window and a dwell time of 15 ms per transition. Both samples were run in technical duplicate.

The LC–MS/MS SRM data was imported into Skyline using the following transition settings: Precursor mass: Monoisotopic, Product ion mass: Monoisotopic, Collision energy: Thermo Altis. Filter-Peptides: Precursor charges: 2, 3, 4; Ion charges: 1, 2; Ion types: y, b. Ion match tolerance of 0.05 $m/z$, Instrument: 50–1500 $m/z$; Precursor $m/z$ exclusion window: 5 $m/z$. Library: Ion match tolerance: 0.5 $m/z$. Instrument: Min $m/z$: 300, max $m/z$: 1500. Method match tolerance: 0.055 $m/z$. Full-Scan: MS/MS filtering: Acquisition method: Targeted (Product mass analyzer: QIT, Resolution: 0.7 $m/z$. The peak integration boundaries were manually revised and synchronized peak integration was applied. To determine the relative quantification of Endophilin A1-3 the Peak Area intensity for each Q1/Q3 transition was extracted from Skyline and the Total Peak Area for each peptide was summed to produce the Total Peak Area intensity for each protein. The mass spectrometry proteomics data have been deposited to the ProteomeXchange Consortium via the PRIDE (Perez-Riverol et al, 2022) partner repository with the dataset identifier PXD045183. Size-exclusion chromatography - multi-angle laser light scattering (SEC-MALS).

Size-exclusion chromatography (SEC) combined with multi-angle laser light scattering (MALS) was carried out using a Superdex 75 10/300 GL column (GE Healthcare) equilibrated in a buffer containing 20 mM HEPES, 100 mM NaCl, 1 mM DTT, 1 mM PMSF, pH 7.2. The column was run at a flow rate of 0.5 mL min$^{-1}$ on an AKTAbasic liquid chromatography system (GE Healthcare) coupled to a miniDawn MALS detector and an Optilab refractive index detector (Wyatt Technology). The SEC-MALS technique provides a molecular mass estimate from the size-exclusion chromatography that is independent of shape (Folta-Stogniew and Williams, 1999). The molecular masses of bovine serum albumin (used as a control) and dyn1xA-PRR/Endophilin-A1-SH3 were calculated using a differential index of refraction (dn/dc) value of 0.185.

## NMR spectroscopy

All NMR spectra were acquired at 298 K on a Bruker Advance 600 spectrometer, equipped with a triple-resonance cryogenically cooled probe and z-axis pulsed-field gradients. $^{15}$N-HSQC and triple-resonance experiments were recorded using standard pulse sequences from the Bruker library. All spectra were processed with TopSpin software (Bruker) and analyzed using Sparky (T. D. Goddard and D. G. Kneller, SPARKY 3, University of California, San Francisco, CA). CSPs (Δδ) in $^{15}$N-HSQC experiments were calculated as $\delta = \sqrt{\{(\delta H)^2 + (\delta N/5)^2\}}$.

## Plasmid construction and lentivirus preparation of Amphiphysin 1 shRNA

For knocking down Amphiphysin 1 in hippocampal neurons, three Amphiphysin shRNA sequence cloned in pLKO.1 puro backbone were purchased from Sigma (shRNA#1, TRCN0000093273: CCGGGCTAGTCCCAACCACACATTACTCGAGTAATGTGTGGTTGGGACTAGCTTTTTG), shRNA#2, TRCN0000093270: CCG

GGCCTCAATGAAGCTCACTGAACTCGAGTTCAGTGAGCTT-CATTGAGGCTTTTTG and shRNA#3, TRCN0000380846: GTAC CGGAGCGAACTCTGATGAACTTAACTCGAGTTAAGTTCAT AGAGTTCGCTTTTTTTG) (also see Reagents and tools table). For non-targeting scramble control, oligo's containing sequence (GATCCCTTCGCACCCTACTTCGTGGttcaagagaCCACGAAGT AGGGTGCGAATTTTTFGGAAATTAAT) was cloned under U6 promoter. Annealed oligo's were inserted into BamH1 and PacI sites of modified pFUGW vector using TAKARA solution I. Lentivirus containing either Amphiphysin 1 shRNA or non-targeting scramble shRNA was prepared as described earlier (Watanabe et al, 2018). Briefly, shRNA construct along with two helper DNA constructs (pHR-CMV8.2 deltaR (Addgene 8454) and pCMV-VSVG (Addgene 8455)) at a 4:3:2 molar ratio was transfected into HEK293T cells using polyethylene amine. Cell supernatant containing the virus was collected 3 days after transfection and concentrated 20-fold using Amicon Ultra 15 10 K (Millipore) centrifugal filter. Aliquots were flash-frozen in liquid nitrogen and stored at −80 °C until use.

## Western blot analysis for Amphiphysin 1 knock down level

To test the knock-down efficiency of amphiphysin 1 shRNA, lentivirus containing scramble or shRNA was added to cultured hippocampal neurons on DIV8. shRNA#1 and #2 caused neuronal cell death. shRNA#3 did not affect the neuronal cell health when the virus is infected after DIV8. Thus, shRNA#3 was used for western blotting and EM experiments. Two different viral doses were tested in Western blotting, 10 µl and 25 µl (low and high doses, respectively, in Appendix Fig. S4) for 300 K hippocampal neurons. Neurons were harvested on DIV14 and lysed by addition of lysis buffer (50 mM Tris pH 8.0 and 1% SDS containing cOmplete Mini Protease Inhibitor (Roche)) and boiling at 95 °C for 5 min. Lysates were centrifuged at $15,000 \times g$ for 15 min at 4 °C and the supernatants were separated in SDS-PAGE and transferred onto Immobilon-FL membranes (Millipore). Following blocking with 5% skim milk in PBS containing 0.05% Tween-20 (PBST) for 30 min, membranes were incubated with Amphiphysin 1 (SYSY, 120 002, RRID: AB_887690)) and control b-actin (SYSY, 251 011) antibodies diluted in 5% BSA/PBST overnight at 4 °C, followed by secondary antibodies tagged with IR dyes (LiCor) diluted in 1:10,000 in 5% BSA/PBST for 45 min at room temperature. Signal was detected using LiCor Odyssey Clx and quantification was done by Image Studio Lite from LiCor. For EM experiments 7 µl of shRNA#3 was used per 75 K neurons to knock down Amphiphysin 1.

## Neuronal cell culture

To prepare primary neuronal cultures, the following procedures (Itoh et al, 2019) were carried out. Newborn or embryonic day 18 (E18) mice of both genders were decapitated. The brain is dissected from these animals and placed on ice cold dissection medium (1 x HBSS, 1 mM sodium pyruvate, 10 mM HEPES, 30 mM glucose, and 1% penicillin-streptomycin). For the knockout experiments, $Dnm1^{+/-}$ were bred to obtain $Dnm1^{+/+}$ and $Dnm1^{-/-}$ animals. Genotyping was performed on clipped tail meanwhile brains were kept in 1 x HBBS on ice. DNA was extracted in 50 mM NaCl at

100 °C for 30 min incubation followed by adjusting pH and PCR reaction using custom designed primers (Imoto et al, 2022) and KOD Hot Start DNA Polymerase (Takara).

For high pressure freezing, hippocampal neurons were cultured on a feeder layer of astrocytes. Astrocytes were harvested from cortices with treatment of trypsin (0.05%) for 20 min at 37 °C, followed by trituration and seeding on T-75 flasks containing DMEM supplemented with 10% FBS and 0.2% penicillin-streptomycin. After 2 weeks, astrocytes were plated onto 6-mm sapphire disks (Technotrade Inc) coated with poly-D-lysine (0.1 mg/ml), collagen (0.6 mg/ml), and 17 mM acetic acid at a density of $13 \times 10^3$ cells/cm². After 1 week, astrocytes were incubated with 5-Fluoro-2′-deoxyuridine (81 µM) and uridine (204 µM) for at least 2 h to stop the cell growth, and then medium was switched to Neurobasal-A (Gibco) supplemented with 2 mM GlutaMax, 2% B27 and 0.2% penicillin-streptomycin prior to addition of hippocampal neurons. Hippocampi were dissected under a binocular microscope and digested with papain (0.5 mg/ml) and DNase (0.01%) in the dissection medium for 25 min at 37 °C. After trituration, neurons were seeded onto astrocyte feeder layers at density of $20 \times 10^3$ cells/cm². Cultures were incubated at 37 °C in humidified 5% $CO_2$/95% air atmosphere. At DIV14-15, neurons were used for high pressure freezing experiments.

For fluorescence imaging, dissociated hippocampal neurons were seeded on 18-mm or 25-mm coverslips coated with poly-L-lysine (1 mg/ml) in 0.1 M Tris-HCl (pH 8.5) at a density of $25–40 \times 10^3$ cells/cm². Neurons were cultured in Neurobasal media (Gibco) supplemented with 2 mM GlutaMax, 2% B27, 5% horse serum and 1% penicillin-streptomycin at 37 °C in 5% $CO_2$. Next day, medium was switched to Neurobasal with 2 mM GlutaMax and 2% B27 (NM0), and neurons maintained thereafter in this medium. For biochemical experiments, dissociated cortical neurons were seeded on poly-L-lysine coated plates with Neurobasal media supplemented with 2 mM GlutaMax, 2% B27, 5% horse serum and 1% penicillin-streptomycin, at a density of $1 \times 10^5$ cells/cm². Next day, the medium was switched to Neurobasal medium with 2 mM GlutaMax and 2% B27, and neurons maintained in this medium thereafter. A half of the medium was refreshed every week.

## Transient transfection of neurons

For transient protein expression, neurons were transfected at DIV13-14 using Lipofectamine 2000 (Invitrogen) in accordance with manufacture's manual with minor modifications (Araki et al, 2015). Before transfections, a half of medium in each well was transferred to 15 mL tubes and mixed with the same volume of fresh NM0 and warmed to 37 °C in $CO_2$ incubator. This solution later served as a conditioned medium. Briefly, plasmids were mixed well with 2 µl Lipofectamine in 100 µl Neurobasal media and incubated for 20 min. For Dyn1xA and Endophilin A expressions, 0.5 µg of constructs were used to reduce the overexpression artifacts (Imoto et al, 2022). The plasmid mixture was added to each well with 1 ml of fresh Neurobasal media supplemented with 2 mM GlutaMax and 2% B27. After 4 h, medium was replaced with the pre-warmed conditioned media. To prevent way higher expression of proteins, neurons were incubated for less than 20 h and fixed for imaging.

## Immunofluorescence staining

For immunofluorescence, 125k neurons were seeded on 18 mm poly-L-lysins (1 mg/mL, overnight) coated coverslips (Thickness 0.09–0.12 mm, Caroline Biological) in 12-well plate (Corning). Neurons were fixed at DIV14-15 with pre-warmed (37 °C) 4% paraformaldehyde and 4% sucrose in PBS for 20 min and wash three times with PBS to remove remaining PFA. Next, fixed neurons were permeabilized with 0.2% Triton X-100 in PBS for 8 min at room temperature followed by three times wash with PBS. After blocking with 1% BSA in PBS for 30 min, cells were incubated with primary antibodies diluted in 60 µl of 1% BSA/PBS overnight at 4 °C in a humidity chamber. Unbound antibodies were removed by washing with PBS three times followed by appropriate secondary antibodies in 60 µl of 1% BSA/PBS for 1 h at room temperature in a humidity chamber. Excess antibodies were removed by washing with PBS three times. Coverslips were rinsed with MilliQ water three times to remove salts and were mounted in ProLong Gold Antifade Mountant (Invitrogen) and stored at 4 °C until imaging. Dyn1xA-GFP was stained with 1:500 dilution of anti-GFP antibodies, Rabbit polyclonal (MBL International) (RRID:AB_591819) or 1:500 dilution of anti-GFP antibodies, Chicken polyclonal (Chemicon®, Abcam) (RRID:AB_90890). Endophilin A1 or A2-mCherry was stained with 1:500 dilution of anti-DsRed antibodies, Rabbit polyclonal (Clontech) (RRID:AB_10013483) and endogenous Bassoon was stained with 1:500 dilution of anti-Bassoon antibody, mouse monoclonal (Enzo Life Sciences) (RRID:AB_2038857). 50 µM of anti-rabbit ATTO647 (Rockland) (RRID:AB_10895682) and 1:1000 dilution of anti-mouse Alexa594 (Invitrogen) (RRID:AB_141372) was used for the secondary antibodies. Coverslips were rinsed with MilliQ water three times to remove salts and mounted in ProLong Diamond Antifade Mountant (Thermo Fisher) and stored at 4 °C until imaging.

## Flash-and-freeze experiments

For flash-and-freeze experiments (Watanabe et al, 2013a, 2014), mouse primary cultured hippocampal neurons were cultured on the sapphire disks. Lentivirus was added on DIV3 for the dynamin rescue experiments, on DIV8 for the amphiphysin KD experiments. Sapphire disks with cultured neurons (DIV14-15) were mounted in the freezing chamber of the high-pressure freezer (HPM100 or EM ICE, Leica), which was set at 37 °C. The physiological saline solution contained 140 mM NaCl, 2.4 mM KCl, 10 mM HEPES, 10 mM Glucose (pH adjusted to 7.3 with NaOH, 300 mOsm, 4 mM $CaCl_2$, and 1 mM $MgCl_2$. Additionally, NBQX (3 µM) and Bicuculline (30 µM) were added to suppress recurrent network activity following optogenetic stimulation of neurons. To minimize the exposure to room temperature, solutions were kept at 37 °C water bath prior to use. The table attached to the high-pressure freezer was heated to 37 °C while mounting specimens on the high-pressure freezer. The transparent polycarbonate sample cartridges were also warmed to 37 °C. Immediately after the sapphire disk was mounted on the sample holder, recording solution kept at 37 °C was applied to the specimen and the cartridge was inserted into the freezing chamber. The specimens were left in the chamber for 30 s to recover from the exposure to ambient light. We applied a single light pulse (10 ms) to the specimens (20 mW/mm²). This stimulation protocol was chosen based on the results from previous experiments showing approximately 90% of cells fire at least one action potential (Watanabe et al, 2013b). Unstimulated controls for each experiment were always frozen on the same day from the same culture. We set the device such that the samples were frozen at 0.1, 1, or 10 s after the initiation of the first stimulus. For ferritin-loading experiments, cationized ferritin (Sigma-Aldrich) was added in the saline solution at 0.25 mg/ml. The calcium concentration was reduced to 1 mM to suppress spontaneous activity during the loading. The cells were incubated in the solution for 5 min at 37 °C. After ferritin incubation, the cells were immersed in the saline solution containing 4 mM $Ca^{2+}$. For dynamin experiments, all samples were incubated with 1 µM TTX for overnight to block spontaneous network activity and reduce the number of pits arrested on the plasma membrane prior to flash-and-freeze experiments (Raimondi et al, 2011; Wu et al, 2014).

Following high-pressure freezing, samples were transferred into flow-through rings (Leica) containing 0.1% tanic acid (EMS),1% glutaraldehyde (EMS) in anhydrous acetone (EMS) chilled at −90 °C in an automated freeze-substitution device (AFS2, Leica). The freeze-substitution was performed with the following program: −90 °C for 36 h, five times wash with anhydrous acetone chilled at −90 °C, 2% osmium tetroxide (EMS) at at −90 °C for 11 h, 5 °C per hour to −20 °C, 12 h at −20 °C, and 10 °C per hour to 4 °C. Following en bloc staining with 0.1% uranyl acetate and infiltration with Epon-Araldite (30% for 3 h, 70% for 4 h, and 90% for overnight), the samples were embedded into 100% Epon-Araldite resin (Araldite 4.4 g, Epon 6.2 g, DDSA 12.2 g, and BDMA 0.8 ml) and cured for 48 h in a 60 °C oven. Serial 40-nm sections were cut using a microtome (Leica) and collected onto pioloform-coated single-slot grids. Sections were stained with 2.5% uranyl acetate before imaging.

## Electron microscopy

Ultrathin sections of samples were imaged at 80 kV on the Hitachi 7600 at 150,000x. At these magnifications, one synapse essentially occupies the whole screen, and thus, with the bidirectional raster scanning of a section, it is difficult to pick certain synapses, reducing bias while collecting the data. In all cases, microscopists were additionally blinded to specimens and conditions of experiments. Both microscopes were equipped with an AMT XR80 camera, which is operated by AMT capture software v6. About 120 images per sample were acquired. If synapses do not contain a prominent postsynaptic density, they were excluded from the analysis.

Electron micrographs were analyzed using SynpasEM. Briefly, images were pooled into a single folder from one set of experiments, randomized, and annotated blind. Using custom macros (Watanabe et al, 2020), the x- and y-coordinates of the following features were recorded in Fiji and exported as text files: plasma membrane, postsynaptic density, synaptic vesicles, large vesicles, endosomes, and pits. By visual inspection, large vesicles were defined as any vesicle with a diameter of 60–100 nm. Endosomes were distinguished by any circular structures larger than large vesicles or irregular membrane-bound structures that were not mitochondria or endoplasmic reticulum. Late endosomes and multivesicular bodies were not annotated in this study. Pits were defined as smooth membrane invaginations within or next to the active zone, which were not mirrored by the postsynaptic

membranes. After annotations, the text files were imported into Matlab (MathWorks). The number and locations of vesicles, pits, endosomes were calculated using custom scripts (Watanabe et al, 2020). The example micrographs shown were adjusted in brightness and contrast to different degrees, rotated and cropped in Adobe Photoshop (v21.2.1) or Illustrator (v24.2.3). Raw images and additional examples are provided in Figshare or upon request. The macros and Matlab scripts are available at https://github.com/shigekiwatanabe/SynapsEM.

## Statistical analysis

All electron microscopy data are pooled from multiple experiments after examined on a per-experiment basis (with all freezing on the same day); none of the pooled data show significant deviation from each replicate (Source data Fig. 5). Sample sizes were based on our prior flash-and-freeze experiments (~2–3 independent cultures, over 200 images), not power analysis. An alpha was set at 0.05 for statistical hypothesis testing. Normality was determined by D'Agostino-Pearson omnibus test. Comparisons between two groups were performed using a two-tailed Welch two-sample t-test if parametric or Wilcoxon rank-sum test and Mann–Whitney Test if nonparametric. For groups, full pairwise comparisons were performed using one-way analysis of variance (ANOVA) followed by Holm-Šídák's multiple comparisons test if parametric or Kruskal–Wallis test followed by Dunn's multiple comparisons test if nonparametric. All statistical analyses were performed and all graphs created in Graphpad Prism (v8).

All fluorescence microscopy data were first examined on a per-experiment basis. For Fig. 4, the data were pooled; none of the pooled data show significant deviation from each replicate (Appendix Fig. S7 and Source data Fig. 4). Sample sizes were 2 independent cultures, 50-100 synapses from 4 different neurons in each condition. An alpha was set at 0.05 for statistical hypothesis testing. The skewness was determined by Pearson's skewness test in GraphPad Prism (v8). Since data were all nonparametric, Mann–Whitney test or Kruskal–Wallis test were used. p-values in multiple comparison were adjusted with Bonferroni correction. Confidence levels were shown in each graph. All statistical analyses were performed, and all graphs created in Graphpad Prism (v8).

## Confocal and stimulated emission depletion (STED) microscopy

All confocal and STED images were obtained using a home-built two-color STED microscope (Han and Ha, 2015; Ma and Ha, 2019). Quality of the STED images are examined by comparing the confocal and STED images and measuring the size of signals at synapses and PSF (non-specific signals from antibodies). If transfected neurons are dying or expression levels is too high and showing abnormal morphology, they are excluded from the imaging. Non-neuronal cells (e.g., glia cells) are also excluded from the imaging (Imoto et al, 2022). Basically, a femtosecond laser beam with repetition rate of 80 MHz from a Ti:Sapphire laser head (Mai Tai HP, Spectra-Physics) is split into two parts: one part is used for producing the excitation beam, which is coupled into a photonic crystal fiber (Newport) for wide-spectrum light generation and is further filtered by a frequency-tunable acoustic optical tunable filter (AA Opto-Electronic) for multi-color excitation. The other part of the laser pulse is temporally

stretched to around 300 ps (with two 15-cm-long glass rods and a 100-m long polarization-maintaining single-mode fiber, OZ optics), collimated and expanded, and wave-front modulated with a vortex phase plate (VPP-1, RPC photonics) for hollow STED spot generation to de-excite the fluorophores at the periphery of the excitation focus, thus improving the lateral resolution. The STED beam is set at 765 nm with power of 120 mW at back focal plane of the objective (NA = 1.4 HCX PL APO 1003, Leica), and the excitation wavelengths are set as 594 nm and 650 nm for imaging Alexa-594 and Atto-647N labeled targets, respectively. The fluorescent photons are detected by two avalanche photodiodes (SPCM-AQR-14-FC, Perkin Elmer). The images are obtained by scanning a piezo-controlled stage (Max311D, Thorlabs) controlled with the Imspector data acquisition program.

## STED image segmentation analysis

All the cluster distance measurements are performed using STED images. For the measurements, a custom MATLAB code package (Imoto et al, 2022) was modified using GPT-4 (OpenAI) to perform semi-automated image segmentation and analysis of the endocytic protein distribution relative to the active zone marked by Bassoon or relative to Dyn1xA cluster in STED images. First, the STED images were blurred with a Gaussian filter with radius of 1.2 pixels to reduce the Poisson noise, and then deconvoluted twice using the built-in deconvblind function: the initial point spread function (PSF) input is measured from the unspecific antibodies in the STED images. And the second PSF (enhanced PSF) input is chosen as the returned PSF from the initial run of blind deconvolution (Sapoznik et al, 2020). The enhanced PSF was used to deconvolute the STED images to be analyzed. Each time 10 iterations are performed. Series of the deconvoluted STED images are loaded to the segmentation MATLAB script utilizing MIJ: Running ImageJ and Fiji within Matlab (Sage 2017, MATLAB Central File Exchange). All presynaptic boutons in each deconvoluted images were selected within $30 \times 30$-pixel ($0.81 \ mm^2$) ROIs based on the varicosity shape and bassoon or Dyn1xA signals. The boundary of active zone or Dyn1xA puncta was identified as the contour that represents half of the intensity of each local maxima in the Bassoon channel. The Dyn1xA clusters and Endophilin A clusters are picked by calculating pixels of local maxima. The distances between the Dyn1xA cluster and active zone boundary or Endophilin A clusters are automatically calculated correspondingly. For this distance measurement, first, MATLAB distance2curve function (John D'Errico 2024, MATLAB Central File Exchange) calculated the distance between the local maxima pixel and all the points on the contour of the active zone or Dyn1xA cluster boundary. Next, the minimum distance for each local maxima pixel was selected. Signals over crossing the ROIs and the Bassoon signals outside of the transfected neurons were excluded from the analysis. The MATLAB scripts are available by request.

## pHluorin experiments

Experiments using vesicular glutamate transporter 1-pHluorin (vGlut1-pHluorin) imaging were performed on primary cultured hippocampal neurons from littermates. Lentivirus: vGlut1-pHluorin, and Dyn1xA, Dyn1xA-S851/857D or Dyn1xA-R846A were infected at DIV3 and were imaged between DIV14 or 15. Extracellular field stimulation was performed using a perfusion chamber with field stimulation (Warner Instruments). Stimulation protocols were 10 stimuli at 20 Hz, with each pulse lasting 1 ms. The stimulation

chamber was set to deliver 150 mA current for each action potential. Before each train of action potentials, baseline images were obtained for 5 s in the absence of stimulation. Images were acquired at a rate of 2 Hz using 63× objective lens (NA = 1.45) and LED light source at 2048 × 2048 pixel resolution by ECLIPSE Ti2 (Nikon) equipped with ORCA-Fusion BT Digital CMOS camera (HAMAMATSU). NIS-Elements AR software (Nikon) was used for the image acquisition. During imaging experiments, neurons were bathed in extracellular solution containing: 4 mM $Ca^{2+}$ and 1 mM $Mg^{2+}$, 30 mM bicuculline, 3 mM NBQX. Solutions were delivered through a custom heated flow-pipe, adjusted for a set point of 37 ℃. Temperatures were also verified by intermittent checks with a temperature probe. Fields of view were selected randomly by eye to image. The selection of puncta that were used to analyze pHluorin kinetics is described below.

### pHluorin image analysis

pHluorin images were analyzed in ImageJ, all images for a given field of view were background subtracted using "Rolling Ball" background subtraction with radius 50. To identify synapses, circular regions of 2 μm² (ROIs) were manually selected from the images between the average baseline (a frame immediately prior to stimulation) and the peak fluorescence. Synapses were chosen on the basis of their response to activity and irrespective of decay kinetics. Intensity was measured for each ROI. To correct signal loss due to the bleaching during the imaging, the fluorescence intensity over time, $F(t)$, was then fitted to an exponential function:

$$F_{corr}(t) = F(t) \times e^{tK}$$

where the corrected normalized fluorescence over time is $F_{corr}(t)$, time after the photo bleaching is $t$, and $K$ is the rate constant acquired from the no stimulation control and measured exponential curve fitting tool in Fiji. Intensity was then normalized to one frame before stimulus and the peak fluorescence following the stimulus. Time constants were obtained by fitting a single exponential in GraphPad Prism to the decay of each normalized field of view. Time constant and percentage of fluorescence recovered at 40 s wer analyzed using the ROI averages at each time-point after the peak for each field of view. If data point does not fit the curve or contain outlier value, those were excluded from the analysis. We used Kruskall–Wallis nonparametric ANOVA for dataset collected from pHluorin imaging. Dunn's post-test was performed to compare all pairs of columns.

## Data availability

All original data: Mendeley (https://doi.org/10.17632/r63yvjpkvj.1—available on the date of publication). Morphometry analysis codes: Github (https://github.com/shigekiwatanabe/SynapsEM). STED image analysis codes: Github (https://github.com/imotolab-neuroem/STED_image_analysis_package_public_v1.4/tree/main). Any additional information required for data re-analysis is available from the lead contact upon request.

The source data of this paper are collected in the following database record: biostudies:S-SCDT-10_1038-S44318-024-00145-x.

## Peer review information

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

## Acknowledgements

We thank Pietro De Camilli for sharing mice and Amphiphysin I plasmids, and Peter McPherson for Endophilin A1-SH3 domain plasmids. We are also indebted to M Delanoy, B Smith, and Hoku West-Foyle at the Johns Hopkins Microscopy Facility for technical assistance in electron and optical microscopy, Sydney Brown, and Christian Pearson for DNA cloning, animal husbandry and preparation for the neuron cultures, Quan Gan and Grant Kusick for microscopy setup for the pHluorin experiment. The Ha group would like to thank Kyu Young Han for the initial building of the STED microscope. We are grateful for financial support from the National Health & Medical Research Council Australia (GNT1069493, GNT1052494, and GNT1047070) and the Children's Medical Research Institute (CMRI), and for equipment from the Australian Cancer Research Foundation, the Ramaciotti Foundation and the Cancer Institute NSW. SK and MC gratefully acknowledge scholarship support from CMRI. SW and this work were supported by start-up funds from the Johns Hopkins University School of Medicine, Johns Hopkins Discovery funds, Johns Hopkins Catalyst award, Marine Biological Laboratory Whitman Fellowship, Chan-Zuckerberg Initiative Collaborative Pair Grant, Chan-Zuckerberg Initiative Supplement Award, Brain Research Foundation Scientific Innovation Award, Helis Foundation award, the National Science Foundation (1727260), and the National Institutes of Health (1DP2 NS111133-01, 1R01 NS105810-01A1, and R35 NS132153) awarded to SWW is an Alfred P Sloan fellow, a McKnight Foundation Scholar, a Klingenstein and Simons Foundation scholar, and Vallee Foundation Scholar. YI was supported by JSPS, Kazato Research Foundation and American Lebanese Syrian Associated Charities (ALSAC). TH is an investigator of the Howard Hughes Medical Institute. The EM ICE high-pressure freezer was purchased partly with funds from an equipment grant from the National Institutes of Health (S10RR026445) awarded to Scot C Kuo.

## Author contributions

**Yuuta Imoto**: Conceptualization; Resources; Data curation; Software; Formal analysis; Funding acquisition; Validation; Investigation; Visualization; Methodology; Writing—original draft; Writing—review and editing. **Jing Xue**: Conceptualization; Resources; Data curation; Software; Formal analysis; Validation; Investigation; Visualization; Methodology; Writing—original draft; Writing—review and editing. **Lin Luo**: Resources; Data curation; Validation; Investigation. **Sumana Raychaudhuri**: Resources; Data curation; Validation; Investigation; Visualization. **Kie Itoh**: Resources; Data curation; Validation; Investigation; Methodology. **Ye Ma**: Resources; Data curation; Software; Investigation; Visualization; Methodology. **George E Craft**: Resources; Data curation; Investigation; Methodology. **Ann H Kwan**: Resources; Data curation; Formal analysis; Validation; Investigation. **Tyler H Ogunmowo**: Resources; Validation. **Annie Ho**: Resources; Validation; Investigation. **Joel P Mackay**: Resources; Data curation; Validation; Investigation; Methodology. **Taekjip Ha**: Funding acquisition; Investigation. **Shigeki Watanabe**: Data curation; Supervision; Funding acquisition; Investigation; Visualization; Writing—original draft; Project administration; Writing—review and editing. **Phillip J Robinson**: Supervision; Funding acquisition; Writing—original draft; Project administration; Writing—review and editing.

Source data underlying figure panels in this paper may have individual authorship assigned. Where available, figure panel/source data authorship is listed in the following database record: biostudies:S-SCDT-10_1038-S44318-024-00145-x.

## Disclosure and competing interests statement

The authors declare no competing interests.

# Expanded View Figures

**A**

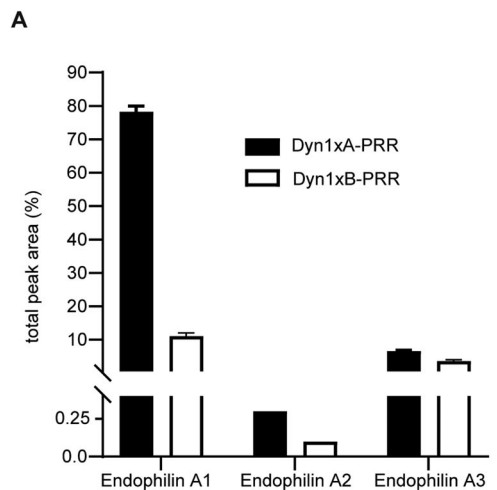

**B**

Endophilin A1
1.     K.VGGAEGTKLDDDFKEMER.K [21, 38]
2.     R.AVMEIMTK.T [46, 53]
3.     K.LSMINTMSK.I [68, 76]
4.     K.QNFIDPLQNLHDK.D [137, 149]
5.     K.QAVQILQQVTVR.L [228, 239]
6.     R.ALYDFEPENEGELGFK.E [297, 312]

Endophilin A2
1.     K.VGGAEGTKLDDDFREMEK.K [21, 38]
2.     K.QNFIDPLQNLCDK.D [137, 149]
3.     R.QAVQILEELADK.L [228, 239]
4.     K.ITASSSFR.S [283, 290]
5.     K.ALYDFEPENDGELGFR.E [313, 328]

Endophilin A3
1.     K.ASQLFSEK.I [13, 20]
2.     K.ATEYLQPNPAYR.A [54, 65]
3.     K.DSLDINVK.Q [129, 136]
4.     R.QSTEILQELQNK.L [228, 239]
5.     R.IALASQVPR.R [244, 252]
6.     R.GLYDFEPENEGELGFK.E [292, 307]

**Figure EV1.  Mass spectrometry evidence for identification of Endophilin A1 bound to Dyn1xA-PRR.**

(**A**) Rat brain synaptosome lysates were incubated with GST-Dyn1-PRR (either xA or xB) coupled to GSH-sepharose beads. Bound proteins were separated by SDS-PAGE and stained with Coomassie blue. The protein band at 40 kDa from both GST-Dyn1-PRR (either xA or xB) were individually cut and digested using Trypsin/LysC and analyzed using a targeted LC–MS/MS SRM analyses. The total amount of each Endophilin protein bound to each Dyn1-PRR was calculated by summing the total peak area for each Q1/Q3 transition to provide the total peak area. The percentage of the total peak area for each protein was then calculated. Endophilin A1 was the predominant protein with a level of Endophilin A3 at 11-fold lower levels, while A2 levels were more than 250-fold lower than A1. (**B**) List of unique Endophilin isoform-specific peptides used for SRM assay (rat sequences: Endophilin-A1_sp|O35179|SH3G2, Endophilin-A2-sp|O35964|SH3G1, Endophilin-A3-sp|O35180|SH3G3).

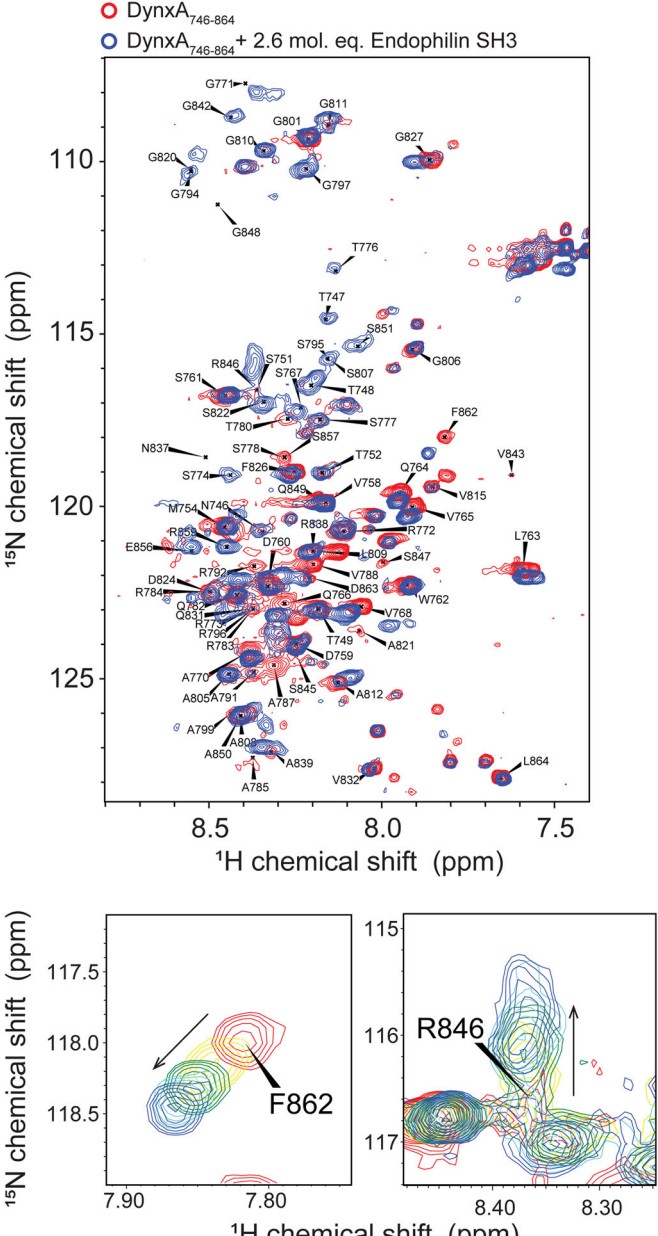

**Figure EV2.** **15N-HSQC titration data for the dynamin–endophilin interaction.**

Partial 15N-HSQC spectra are shown for 0.27 mM 15N-DynxA$_{746-864}$ alone (red) and for the same protein following the addition of 2.6 molar equivalents of unlabeled Endophilin A1 SH3 domain. Assignments are shown for DynxA alone. Note that some signals for which assignment locations are marked are not visible in this spectrum but were visible in other spectra recorded for longer times. The bottom panels show cropped portions of R846 and F862 from the top panel. Arrows indicate peak shifting during the titration processes.

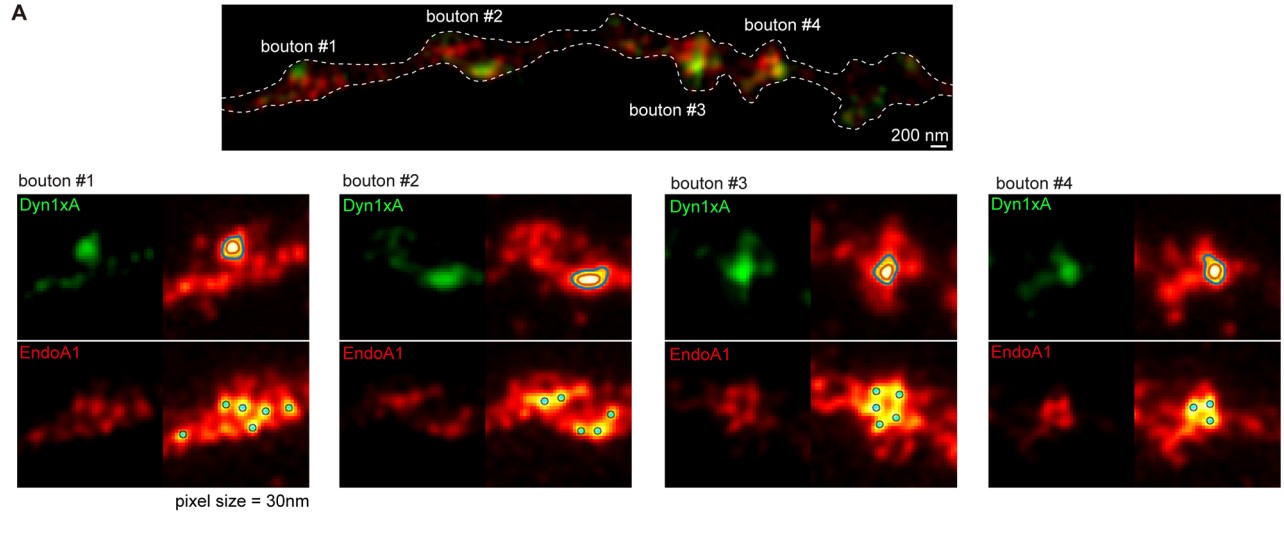

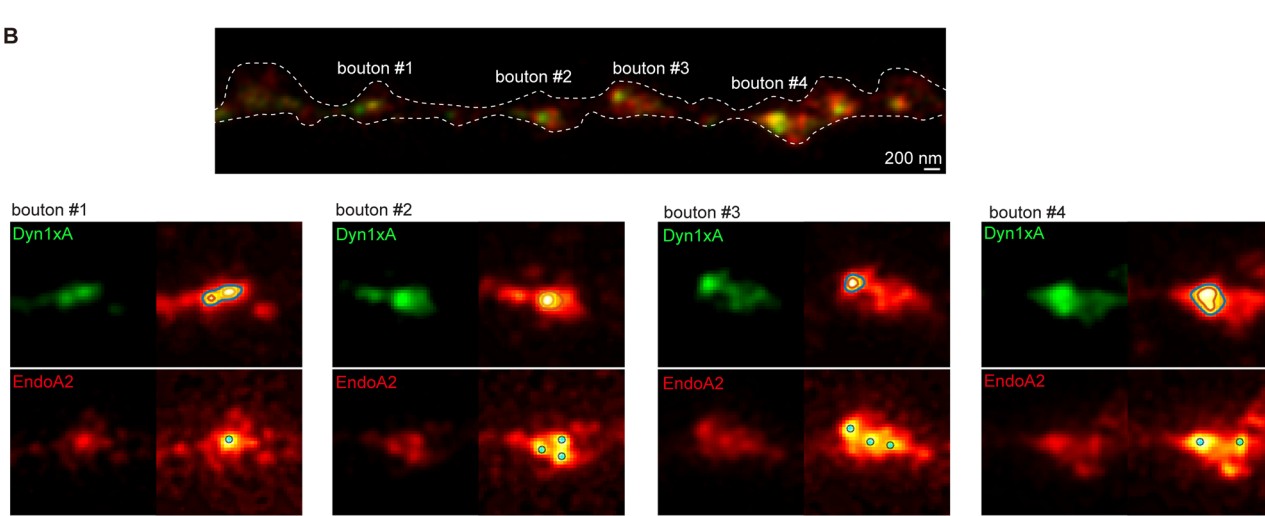

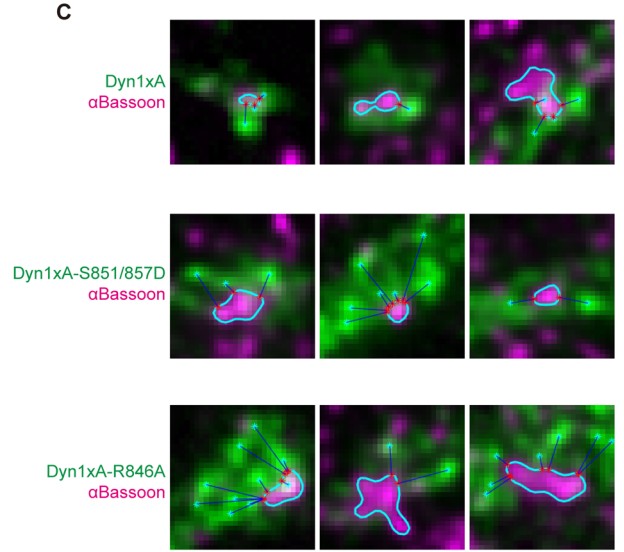

**Figure EV3.   Additional STED images for Fig. 4.**

(A) The top image shows an axon containing multiple boutons. Signals show overexpression of GFP-tagged Dyn1xA (Dyn1xA) and mCherry-tagged Endophilin A1 (EndoA1). The bottom images show magnifications of four boutons in the top image. Red hot LUT images on the right side of Dyn1xA and EndoA1 images are enhanced contrast images. Outer and inner contour 50% and 70% of local maxima of the Dyn1xA, respectively. Black circles represent local maxima of Endophilin A1. In these boutons, there are multiple EndophilinA1 puncta in each bouton. (B) The top image shows an axon containing multiple boutons. Signals show overexpression of mCherry-tagged Dyn1xA (Dyn1xA) and GFP-tagged Endophilin A2 (EndoA2). The bottom images show magnifications of four boutons in the top image. Red hot LUT images on the right side of Dyn1xA and EndoA2 images are enhanced contrast images. Outer and inner contour 50% and 70% of local maxima of the Dyn1xA, respectively. Black circles represent local maxima of Endophilin A2. In these boutons, there are multiple EndophilinA2 puncta in each bouton. (C) STED micrographs of the same synapses as in Fig. 4E with an active zone marker Bassoon (magenta) visualized by antibody staining of GFP-tagged Dyn1xA, Dyn1xA S851D/857D or Dyn1xA R846A (green). Local maxima of Dyn1xA, Dyn1xA S851D/857D or Dyn1xA R846A signals and minimum distance to the active zone boundary are overlayed.

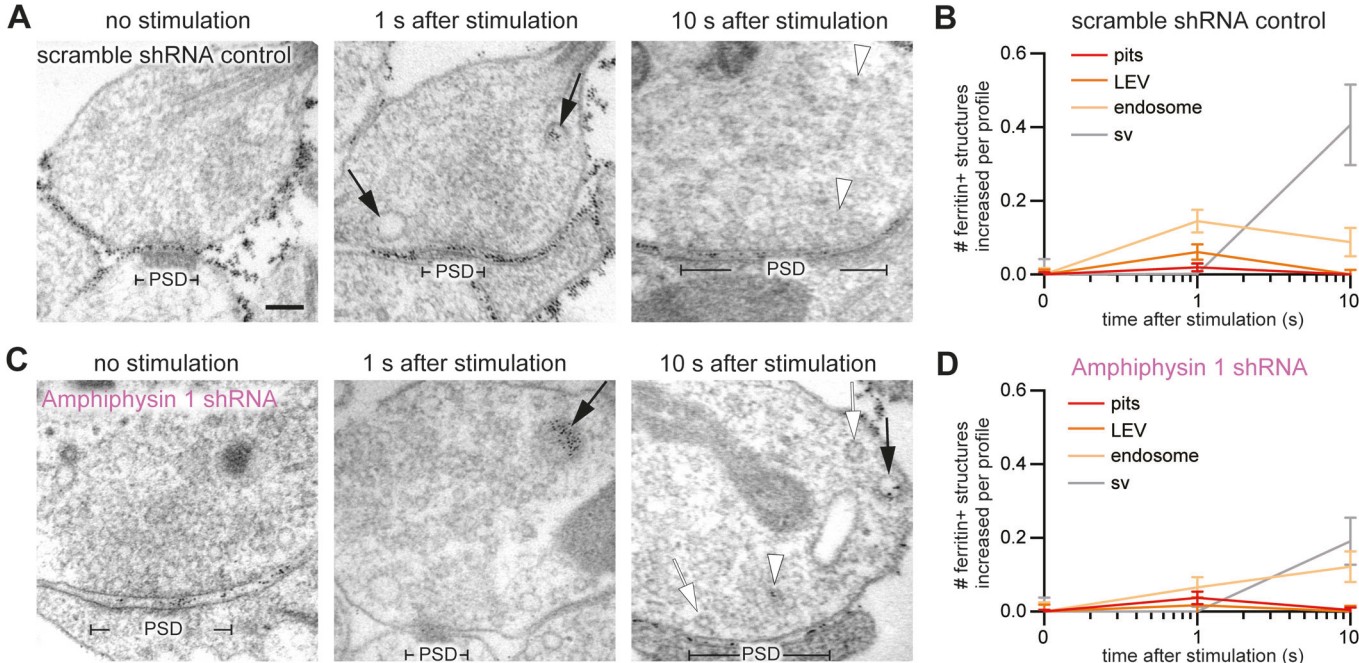

**Figure EV4. Amphiphysin 1 is not essential for ultrafast endocytosis.**

(A, C) Example micrographs showing endocytic pits and ferritin-containing endocytic structures at the indicated time points in neurons expressing scramble RNA (A) and Amphiphysin 1 shRNA (C). Black arrows, ferritin-positive large endocytic vesicles (LEVs) or endosomes; white arrowheads, ferritin-positive synaptic vesicles. Scale bar: 100 nm. PSD, post-synaptic density. (B, D) Plots showing the increase in the number of each endocytic structure per synaptic profile after a single stimulus in neurons expressing scramble RNA (B) and Amphiphysin 1 shRNA (D). The mean and SEM are shown in each graph. All data are from two independent experiments from $N = 2$ cultures prepared and frozen on different days. $n =$ scramble RNA, 436; Amphiphysin 1 shRNA, 609.

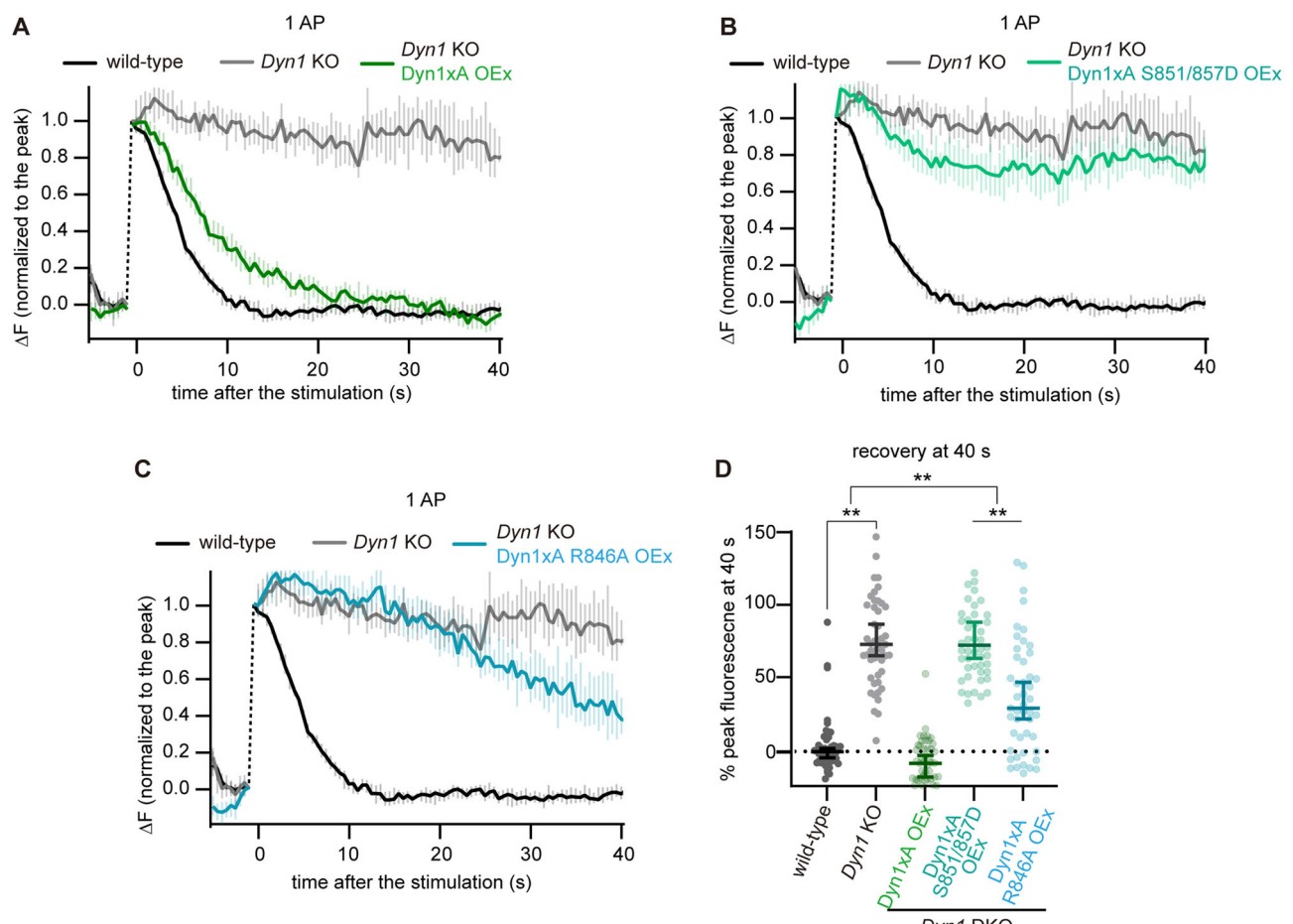

**Figure EV5. Dyn1xA and its long tail is important for endocytosis of synaptic vesicle protein at single action potential.**

(A–C) Plots showing average responses of vesicular glutamate transporter 1 (VGLUT1)-pHluorin in DNM1$^{+/+}$ (wild-type), DNM1$^{-/-}$ (*Dyn1* KO), *Dyn1* KO neurons, overexpressing Dyn1xA (*Dyn1KO* Dyn1xA OEx) (A), *Dyn1* KO neurons, overexpressing Dyn1xA S851/857D (*Dyn1KO* Dyn1xA S851/857D OEx) (B) or *Dyn1* KO neurons, overexpressing Dyn1xA R846A (*Dyn1KO* Dyn1xA R846A OEx). Mouse primary cultured hippocampal neurons were stimulated at single action potentials (AP). The fluorescence signals are normalized to the peak for each bouton. Before stimulation, fluorescence images are acquired for 5 s followed by the stimulation and continued acquisition. (D) The percentage of peak fluorescence remaining at 40 s after the beginning of the imaging. *n* > 60 presynaptic boutons from five different coverslips in each condition. *N* = 2 culture born from three different mothers at DIV14. **$p$ < 0.0001. Knock out neurons are from the littermates in all cases. Kruskal–Wallis Test with full comparisons by post hoc Dunn's multiple comparisons tests.

