## [Peer Review File · The EMBO Journal]

Dynamin 1xA interacts with Endophilin A1 via its spliced long C-terminus for ultrafast endocytosis

Yuuta Imoto, Jing Xue, Lin Luo, Sumana Raychaudhuri, Kie Itoh, Ye Ma, George Craft, Ann Kwan, Tyler Ogunmowo, Annie Ho, Joel Mackay, Taekjip Ha, Shigeki Watanabe, and Phillip Robinson

Corresponding authors: Shigeki Watanabe (shigeki.watanabe@jhmi.edu) , Phillip Robinson (probinson@cmri.org.au)

Review Timeline:

Transferred from Review Commons:	8th Mar 24
Editorial Decision:	13th Apr 24
Revision Received:	26th Apr 24
Accepted:	24th May 24

Editor: Kelly Anderson

Transaction Report:

Review
COMMONS

(Note: This article was transferred to The EMBO Journal following peer review at Review Commons. With the exception of the correction of typographical or spelling errors that could be a source of ambiguity, letters and reports are not edited. Depending on transfer agreements, referee reports obtained elsewhere may or may not be included in this compilation. Referee reports are anonymous unless the Referee chooses to sign their reports.)

Review #1

1. Evidence, reproducibility and clarity:

Evidence, reproducibility and clarity (Required)

In this manuscript, Imoto et al. analyze the specific role of the Dynamin1 splice variant Dyn1xA in so-called ultrafast endocytosis, an important mechanism of synaptic vesicle recycling at synapses. In a previous publication (Imoto et al. Neuron 2022), some of the authors had shown that Dyn1xA, and not the other splice variant Dyn1xB, is essential for ultrafast endocytosis. Moreover, Dyn1xA forms clusters around the active zone for exocytosis and interacts with Syndapin 1 in a phosphorylation dependent manner. However, it was unclear which molecular interactions underlie the specific role of Dyn1xA. Here, the authors provide convincing evidence with pull down assays and CSP that Dyn1xA PRR interacts with EndophilinA1/2 with two binding sites. The first binding site lies in the part common to xA and xB, was previously characterized. The second site was previously uncharacterized, is specific for Dyn1xA, and is regulated by phosphorylation (phosphobox 2). The location of these splice variants and mutated forms at presynaptic sites correlate with the prediction made by the biochemical assays. Finally, the authors perform rescue experiments ('flash and freeze' and VGLUT1-pHluorin imaging experiments) to show that Dyn1xA-EndophilinA1/2 binding is important for ultrafast endocytosis. I find the results interesting, providing an important step in the understanding of the interplay between dynamin and the endocytic proteins interacting with it (endophilin, syndapin, amphiphysin) in the context of synaptic vesicle recycling. The manuscript is clearly written and for the most part the data supports the authors' conclusions (see specific comments below). However, there are some issues which need to be clarified before this manuscript is fully suitable for publication.*

Introduction: the dynx1B Calcineurin binding motif is written PxIxIT consensus but actual sequence is PRITISDP. Is this a typo?

Figure 1: the difference between the constructs used in panels C and D is not clear. In D, is it a truncation without residues 796 and 845? If so, it should be labelled clearly in the Western blots. In Panel E, Dyn1xA 746-798 should be labeled Dyn1x 746-798 because it is common to both splice variants. For amphiphysin binding the authors write that "No difference in binding to Amphiphysin 1 was observed among these peptides (Figure1D-F)." They should write that Dyn1x 746-798 does not bind Amphiphysin1 SH3 domain, confirming the specificity of binding to the 833-838 motif.

Figure S2. The panels are way too small to see the shifts and the labelling. Please provide bigger panels

Figure 2 panel B. There is a typo in the connecting line between the sequence and the CSP peaks. It is 846 instead of 864 (after 839).

Figure 3 panel E. In the text, the authors write that "Western blotting of the bound proteins from the R838A pull-down experiment showed that R838A almost abolished both Endophilin and Amphiphysin binding in xA806-864 (Figure 3D), and reduced Endophilin binding to xA-PRR (Figure 3E)." I think they should write "only slightly reduced Endophilin binding..." it is more faithful to the result and consistent with the conclusion that Endophilin A1 has two binding sites on Dyn1xA PRR. It is unclear why the R846A mutant affects binding of Dyn1xA 806-864 but not Dyn1xA-PRR. Moreover, it affects binding to endophilin as well as

amphiphysin, and therefore it is not specific. It is thus not correct to write that "R846 is the only residue found to specifically regulate the Dyn1 interaction with Endophilin as a part of an SDE". In the Discussion (page 11), the authors refer to the R846A mutation as specifically affecting Endophilin binding. This should be toned down, as it also affects Amphiphysin binding. For this important point, the data on quantification of Endophilin binding should be presented.

Figure 3F-G: what do the star symbols represent in the graphs? I guess the abscissa represents retention time. Please write it clearly instead of a second ordinate for molecular mass, which does not make much sense if this reflects the estimate for the 3 conditions.

Figure 4: The statement that "By contrast [to Dyn1xA], Endophilin A1 or A2 formed multiple clusters (1-5 clusters)" is not at all clear on the presented pictures. The authors should provide views of portions of axons with several varicosities, for the reader to appreciate the cases where there are more EndoA clusters than Dyn1 clusters. Moreover, overexpression of EndophilinA1/2-mCherry is not sufficient to assess its localization. Please consider either immunofluorescence or genome editing (e.g. Orange or TKIT techniques). The analysis of the confocal microscopy data is not explained. How is the number of clusters determined? How far apart are they? Confocal microscopy may not have the resolution to distinguish clusters within a synapse. For the STED microscopy, a representation of the processed image (after deconvolution) and the localization of the peaks would be important to assess the measurement of distances. If Dyn1xA S851/857D is more diffuse, are there still peaks to measure for every synapse?

Figures 5 and 6: No specific comment. The data and its analysis is very nice and elegant. The comment on the lack of rescue of Dyn1xA on endosome maturation may be a bit overstated, because many "controls" (shRNA control Figure S5 or Dyn3 KO in Imoto et al. 2022) have a significant number of endosomes 10 s after stimulation. By the way, why did the authors use Dyn1 KO in this study, and not Dyn1.3 DKO as in Imoto et al. 2022?

In the Discussion, the authors present the binding sites (for endophilin and amphiphysin SH3 domains) as independent. However, these proteins form dimers or even multimers as they cluster around the neck of a forming vesicle. Even though they provide evidence in vitro (Figure 3) that in these conditions of high concentration one dyn1xA-PRR binds one SH3 domain, in cells multiple binding sites on the PRR to these proteins may involve avidity effects, as discussed for example in Rosendale et al. 2019 doi 10.1038/s41467-019-12434-9. For example, the high affinity binding of Dyn1-PRR to amphiphysin cannot be explained only by the sequence 830-838.

2. Significance:

Significance (Required)

This study provides a significant advance on the mechanisms of dynamin recruitment to endocytic zones in presynaptic terminals. The work adds a significant step by experienced labs (Robinson, Watanabe) who have provided important insight in the mechanisms by many publications in the last years.

3. How much time do you estimate the authors will need to complete the suggested revisions:

Estimated time to Complete Revisions (Required)

(Decision Recommendation)

Between 1 and 3 months

Yes

Review #2

1. Evidence, reproducibility and clarity:

Evidence, reproducibility and clarity (Required)

This is a compelling study that reports a key discovery to understand the molecular mechanism of ultrafast endocytosis. The authors demonstrate that the Dynamin splice version 1xA (Dynamin 1xA) uniquely binds Endophilin A, in contrast to Dynamin splice version 1xB (Dynamin 1xB) that does not bind Endophilin A and it is not required for ultrafast endocytosis. In addition, the Endophilin A binding occurs in a dephosphorylation-regulated manner. The study is carefully carried out and it is based on high quality data obtained by means of advanced biochemical methodologies, state-of-art flash-freezing electron microscopy analysis, superresolution microscopy and dynamic imaging of exo-and endocytosis in neuronal cultures. The results convincingly support the conclusions.

Although additional experiments are not essential to support the claims of the paper there is room, however, for improvement within the pHluorin experiments. These experiments, that are clearly informative and consistent with the rest of experimental data, do not apply the useful approach to separate endo- from exocytosis. The use of bafilomycin or folimycin to block the vesicular proton pump allows the unmasking the endocytosis that is occurring during the stimulus, that should correspond to ultrafast endocytosis. It would be very elegant to demonstrate that such a component, as expected according to the electron microscopy data, requires the binding of Endophilin A to Dynamin 1xA. If the authors have the pHluorin experiments running, the suggested experiments are very much doable because the reagents and the methodology is already in place and the new data could be generated in around six weeks.

The methods are carefully explained. Some of the experiments are only replicated in two

cultures and the authors should justify the reasons to convince the audience that the approaches used have enough low variability for not increasing the n number. The pHluorin experiments, however, are performed only in a single culture; they should replicate these experiments in at least 3 different cultures (three different mice).

****Minor comments:****

Prior studies referenced appropriately and the the text and figures are clear and accurate.

1. The authors should discuss about the mediators (enzymes) responsible for dephosphorylation of phosphor-box 2 that is key for the Dynamin 1 α -Endophilin A interaction.
3. It would be very helpful to include a final cartoon depicting the key protein-protein interactions regulated by dephosphorylation (activity) and the sequence of molecular events that leads to ultrafast endocytosis

2. Significance:

Significance (Required)

This is a remarkable and important advance in the field of endocytosis. The study reports a key discovery to understand the molecular mechanism of ultrafast endocytosis. Scientist interested in synaptic function and the general audience of cell biologist interested in membrane trafficking will very much value this study. The mechanism reported will potentially be included in textbooks in the near future.

My field of expertise includes molecular mechanisms of presynaptic function and membrane trafficking.

I have not enough experience to evaluate the quality of the NMR experiments, however, I do not have any problem at all with, in my opinion, elegant results reported.

3. How much time do you estimate the authors will need to complete the suggested revisions:

Estimated time to Complete Revisions (Required)

(Decision Recommendation)

Between 1 and 3 months

4. *Review Commons* values the work of reviewers and encourages them to get credit for their work. Select 'Yes' below to register your reviewing activity at Web of Science Reviewer Recognition Service (formerly Publons); note

that the content of your review will not be visible on Web of Science.

Yes

Full Revision

Manuscript number: #RC-2023-02248

Corresponding author(s): Shigeki Watanabe and Phillip J. Robinson

[Please use this template only if the submitted manuscript should be considered by the affiliate journal as a full revision in response to the points raised by the reviewers.]

*If you wish to submit a preliminary revision with a revision plan, please use our "Revision Plan" template. **It is important to use the appropriate template to clearly inform the editors of your intentions.**]*

1. General Statements [optional]

Reviewer #1 (Evidence, reproducibility, and clarity (Required)):

1. In this manuscript, Imoto et al. analyze the specific role of the Dynamin1 splice variant Dyn1xA in so-called ultrafast endocytosis, an important mechanism of synaptic vesicle recycling at synapses. In a previous publication (Imoto et al. Neuron 2022), some of the authors had shown that Dyn1xA, and not the other splice variant Dyn1xB, is essential for ultrafast endocytosis. Moreover, Dyn1xA forms clusters around the active zone for exocytosis and interacts with Syndapin 1 in a phosphorylation dependent manner. However, it was unclear which molecular interactions underlie the specific role of Dyn1xA. Here, the authors provide convincing evidence with pull down assays and CSP that Dyn1xA PRR interacts with EndophilinA1/2 with two binding sites. The first binding site lies in the part common to xA and xB, was previously characterized. The second site was previously uncharacterized, is specific for Dyn1xA, and is regulated by phosphorylation (phosphobox 2). The location of these splice variants and mutated forms at presynaptic sites correlate with the prediction made by the biochemical assays. Finally, the authors perform rescue experiments ('flash and freeze' and VGLUT1-pHluorin imaging experiments) to show that Dyn1xA-EndophilinA1/2 binding is important for ultrafast endocytosis. I find the results interesting, providing an important step in the understanding of the interplay between dynamin and the endocytic proteins interacting with it (endophilin, syndapin, amphiphysin) in the context of synaptic vesicle recycling. The manuscript is clearly written and for the most part the data supports the authors' conclusions (see specific comments below). However, there are some issues which need to be clarified before this manuscript is fully suitable for publication.

>> We thank the reviewer for noting the importance of our study. Indeed, our previous study has raised the question as to why only the Dyn1xA splice variant mediates ultrafast endocytosis, and our current manuscript now resolves this issue.

2. Introduction: the dynx1B Calcineurin binding motif is written PxlxIT consensus but actual sequence is PRITISDP. Is this a typo?

>>The sequence is correct. One thing we failed to mention is that the last amino acid in this motif can be either threonine or serine for calcineurin binding, as we demonstrated previously [Jing, et al., 2011 JBC; PMC3162388]. We have amended the text as follows.

p. 3. calcineurin-binding motif (PxlxI[T/S])¹⁹.

3. Figure 1: the difference between the constructs used in panels C and D is not clear. In D, is it a truncation without residues 796 and 845? If so, it should be labelled clearly in the Western blots. In Panel E, Dyn1xA 746-798 should be labeled Dyn1x 746-798 because it is common to both splice variants.

>> We thank the reviewer for pointing this out. Both C and D used the full-length PRRs of Dyn1xA_{746 to 864} and xB_{746 to 851}. To make the labeling clear, we changed Dyn1xA PRR to "Dyn1xA PRR (746-864)" and Dyn1xB PRR to "Dyn1xB PRR 746-851" in Figure 1. In the main text, we made the following changes.

p. 4: "To identify the potential isoform-selective binding partners, the full-length PRRs of Dyn1xA₇₄₆₋₈₆₄ and xB₇₄₆₋₈₅₁ (hereafter, Dyn1xA-PRR and Dyn1xB-PRR, respectively)."

4. Figure 1: For amphiphysin binding the authors write that "No difference in binding to Amphiphysin 1 was observed among these peptides (Figure1D-F)." They should write that Dyn1x 746-798 does not bind Amphiphysin1 SH3 domain, confirming the specificity of binding to the 833-838 motif.

>> We edited the sentence as suggested.

p. 5. "Dyn1x 746-798 does not bind Amphiphysin1 SH3 domain (Figure 1G), confirming the specificity of binding to the 833-838 motif as reported in previous studies^{29,30}. (Figure 1D-F)."

6. Figure S2. The panels are way too small to see the shifts and the labelling. Please provide bigger panels

>> As suggested, we have now provided bigger panels in Figure S2, and amended the text and Figure legend accordingly.

We also removed Figure S2B as it was not referred to in the text in any way. (It was the reverse experiment – HSQCs of 15N-labelled SH3 titrated with unlabelled dynamin).I

7. Figure 2 panel B. There is a typo in the connecting line between the sequence and the CSP peaks. It is 846 instead of 864 (after 839).

>> Corrected.

8. Figure 3 panel E. In the text, the authors write that "Western blotting of the bound proteins from the R838A pull-down experiment showed that R838A almost abolished both Endophilin and Amphiphysin binding in xA806-864 (Figure 3D), and reduced Endophilin binding to xA-PRR (Figure 3E)." I think they should write "only slightly reduced Endophilin binding..." it is more

Full Revision

faithful to the result and consistent with the conclusion that Endophilin A1 has two binding sites on Dyn1xA PRR.

>> We have now provided quantitative data for R838A and R846A (Fig. 3F and G). Endophilin binding is significantly reduced with R846A.

9. It is unclear why the R846A mutant affects binding of Dyn1xA 806-864 but not Dyn1xA-PRR-.

>> The reviewer asks why the R846A mutant affects binding of Dyn1xA 806-864, but not so much of Dyn1xA-PRR. The explanation is simply that there are two endophilin binding sites in Dyn1xA-PRR. The first is not present in the xA806-864 peptide, while both are present in Dyn1xA-PRR (the full length tail). When doing pull-down experiments, the binding tends to saturate – even when the second site is blocked by R846A. The first site is still able to bind, and the binding appears as normal. The same applies to the R838A mutant.

10. Moreover, it affects binding to endophilin as well as amphiphysin, and therefore it is not specific. It is thus not correct to write that "R846 is the only residue found to specifically regulate the Dyn1 interaction with Endophilin as a part of an SDE". In the Discussion (page 11), the authors refer to the R846A mutation as specifically affecting Endophilin binding. This should be toned down, as it also affects Amphiphysin binding. For this important point, the data on quantification of Endophilin binding should be presented.

>>The reviewer's concern is about our claims of specificity of Endophilin A binding in Dyn1xA R846 mutation experiments. The reviewer is correct, and we have now defined specific parameters for those claims. Specifically, we have added new quantitative data from the Western blots in Fig 3E (full-length Dyn1aX-PRR) as Fig 3F-G. We used full-length Dyn1aX-PRR rather than the xA806-864 peptide because the subsequent transfection experiments use full length Dyn1xA. In the new figures 3F and 3G, we quantified Endophilin A, Amphiphysin and Syndapin1 amounts from the multiple Western blots such as Figure 3E (now n=14, 6 experiments, each in with 2-4 replicates for Dyn1xA PRR). R846A mutated in Dyn1xA-PRR significantly reduces the binding to Endophilin A, but it does not significantly affect the binding to Amphiphysin 1 and Syndapin1 (Fig 3G). Therefore, this particular Dyn1xA-PRR mutation specifically affects Endophilin A binding, in the context of the full-length tail Dyn1aX-PRR. To make these results clear, we modified the text as below.

P7. "R838A and R846A caused smaller reductions in Endophilin binding compared to wild-type Dyn1xA-PRR, (Figure 3E, 3F, R838A, median 68.5 ; Figure 3G, R846A, median 59.3 % : R838A reduced the Dyn1/Amphiphysin interaction (Figure 3E, 3F, median 14.2 % binding compared to wild-type Dyn1xA-PRR). By contrast, R846A did not affect Amphiphysin and Syndapin binding to Dyn1xA-PRR (Figure 3E, 3G). Therefore, R846, being part of an SDE, is the only residue we found to specifically regulate the Dyn1 interaction with Endophilin in the context of the full length tail (DynxA-PRR)".

Full Revision

Additionally, the reviewer notes that “the authors refer to the R846A mutation as specifically affecting Endophilin binding. This should be toned down, as it also affects Amphiphysin binding.” In the light of the above data and new quantitative analysis (Fig 3F-G), we have clarified the conclusion. However, to be clear that this statement is only correct in the context of the full-length DynxA-PRR, we amended texts as follows:

P7. “By contrast, R846A did not affect Amphiphysin and Syndapin binding to Dyn1xA-PRR (Figure 3E, 3G). Therefore, R846, being part of an SDE, is the only residue we found to specifically regulate the Dyn1 interaction with Endophilin in the context of the full length tail (DynxA-PRR)”.

New legends for Figure 3F and G have now been added as follows.

“(F) The binding of Endophilin A, and Amphiphysin 1 and Syndapin1 to Dyn1xA-PRR (wild type) or R838A mutant quantified from Western blots in (E). n=14 (6 experiments with 2-4 replicates in each). Median and 95% confidential intervals are shown. Kruskal-Wallis with Dunn’s multiple comparisons test (** $p < 0.0001$).

(G) The binding of Endophilin A, and Amphiphysin 1 and Syndapin1 to Dyn1xA-PRR (wild type) or R846A mutant quantified from Western blots in (E). n=14 (6 experiments with 2-4 replicates in each). Median and 95% confidential intervals are shown. Kruskal-Wallis with Dunn’s multiple comparisons test was applied ($p < 0.005$.”

11. Figure 3F-G (which are now 3H and 3I in the revised text): what do the star symbols represent in the graphs? I guess the abscissa represents retention time. Please write it clearly instead of a second ordinate for molecular mass, which does not make much sense if this reflects the estimate for the 3 conditions.

>> The “stars” are crosses (x) and represent individual data points. The figure legends have been updated for clarity. The reviewer is correct that the X-axis is retention time (min). The second Y-axis is needed to define the points in the curve marked with crosses (x’s). The legends for Figure 3H and I are now changed as follows.

“(H) SEC-MALS profiles for Dyn1xA alone (in green), Endophilin A SH3 alone (in red) and the complex of the two (in black) are plotted. The x-axis shows retention time. The left axis is the corresponding UV absorbance (280 nm) signals in solid lines, and the right axis shows the molar mass of each peak in crosses. The molecular weight of the complex was determined and tabulated in comparison with the predicted molecular weight. x represent individual data points.

(I) SEC-MALS profiles for a high concentration of Dyn1xA-PRR/Endophilin A SH3 complex (0.5 mg) (in dark blue) and a low concentration of Dyn1xA-PRR/endophilin A SH3 complex (0.167 mg) (in blue). The x-axis shows retention time. The left axis is the corresponding UV absorbance (280 nm) signals in solid lines, and the right axis shows the molar mass of each peak in crosses. The molecular weight of the complex was determined and tabulated in the table. x represent individual data points.”

12. Figure 4: The statement that "By contrast [to Dyn1xA], Endophilin A1 or A2 formed multiple clusters (1-5 clusters)" is not at all clear on the presented pictures. The authors should provide views of portions of axons with several varicosities, for the reader to appreciate the cases where there are more EndoA clusters than Dyn1 clusters.

>>In the revised Figure S4, we added additional STED images for a region of axons with more EndoA1/2 clusters than Dyn1xA clusters. The locations of Dyn1xA and EndoA1/2 clusters are annotated in each image based on the local maximum of intensity, which is determined using our custom Matlab analysis scripts (Imoto, et al., Neuron 2022; for the description of the methods, please refer to the Point #14 below). We also added Figure S3 to describe our analysis pipelines. In the Dyn1xA channel, outer contour indicates 50% of local maxima (boundary of Dyn1xA cluster) while inner contour indicates 70% of local maxima of the clusters. In the EndoA1/2 channel, local maxima of the clusters are indicated as points. To reflect these changes, we modified text as below.

P 9. "By contrast, Endophilin A1 or A2 formed multiple clusters (1-5 clusters) (Figure S4)"

The legends for Figure S4 are now as follows.

"Figure S4. Additional STED images for Figure 4.

(A) The top image shows an axon containing multiple boutons. Signals show overexpression of GFP-tagged Dyn1xA (Dyn1xA) and mCherry-tagged Endophilin A1 (EndoA1). The bottom images show magnifications of four boutons in the top image. Red hot look-up table (LUT) images on the right side of Dyn1xA and EndoA1 images are enhanced contrast images. Outer and inner contours represent 50% and 70% of local maxima of the Dyn1xA, respectively. Black circles represent local maxima of Endophilin A1. In these boutons, multiple EndophilinA1 puncta are present.

(B) The top image shows an axon containing multiple boutons. Signals show overexpression of mCherry-tagged Dyn1xA (Dyn1xA) and GFP-tagged Endophilin A1 (EndoA1). The bottom images show magnifications of four boutons in the top image. Red hot LUT images on the right side of Dyn1xA and EndoA2 images are enhanced contrast images. Outer and inner contours represent 50% and 70% of local maxima of the Dyn1xA, respectively. Black circles represent local maxima of Endophilin A2. In these boutons, multiple EndophilinA2 puncta are present.

(C) STED micrographs of the same synapses as in Figure 4E with an active zone marker Bassoon (magenta) visualized by antibody staining. GFP-tagged Dyn1xA, Dyn1xA S851D/857D or Dyn1xA R846A (green) are additionally stained with GFP-antibodies. Local maxima of Dyn1xA, Dyn1xA S851D/857D or Dyn1xA R846A signals and minimum distance to the active zone boundary are indicated by dark blue lines."

13. Moreover, overexpression of EndophilinA1/2-mCherry is not sufficient to assess its localization. Please consider either immunofluorescence or genome editing (e.g. Orange or TKIT techniques).

>> We agree with the reviewer that overexpression obscures the endogenous localization of proteins. To address this point in our previous publication, we titrated the amount of plasmids for Dyn1xA-GFP and transfected neurons just for 20 hours – this protocol allowed us to uncover the endogenous localization of Dyn1xA despite the fact that it was overexpressed in wild-type

neurons (Imoto, et al., 2022). We also confirmed this localization by ORANGE-based CRISPR knock-in of GFP-tag in the endogenous locus of Dyn1 just after the exon 23 and confirm the true endogenous localization of Dyn1xA (Imoto, et al., 2022). Similar approaches were taken by the Chapman lab to localize Synaptotagmin-1 and Synaptobrevin 2 in axons (Watson et al, 2023, eLife, PMID: 36729040). We did not emphasize this in the first submission, but we took the same approach for the EndoA1/2 localization. This does not mean that they also unmask the endogenous localization, and the reviewer is correct that additional evidence would strengthen the data here. Thus, as suggested, we have looked at the endogenous EndophilinA1 localization by antibody staining. As the reviewer is likely aware, EndophilinA1 also localizes to other places including dendrites and postsynaptic terminals, making it difficult to analyze the data. However, we observe colocalization of Dyn1xA with endogenous EndoA1. Thus, we believe that our major conclusion here drawn based on EndoA1/2-mCherry overexpression is valid (Reviewer's Figure 1). Since the Endophilin signals in neighboring processes obscures its localization in synapses-of-interest, repeating this localization experiments with ORANGE-based knock-in would be ideal.

Reviewers figure 1

However, with the lead author starting his own group and many validations needed to confirm the knock-in results, this experiment would require us at least 4-6 months, and thus, it is beyond the scope of our current study. We will follow up on this localization in the near future, but given that endophilin is required for ultrafast endocytosis (Watanabe, et al., Neuron 2018, PMID: 29953872) and these proteins need to be in condensates at the endocytic sites for accelerating the kinetics of endocytosis (Imoto, et al., Neuron 2022, PMID: 35809574), we are confident that endogenous EndoA1/2 are localized with Dyn1xA.

14. The analysis of the confocal microscopy data is not explained. How is the number of clusters determined? How far apart are they? Confocal microscopy may not have the resolution to distinguish clusters within a synapse.

>>We apologize for the insufficient description of the method. We had provided a more thorough description of the methods in our previous publication (Imoto, et al., Neuron 2022, PMID:

35809574). To make this more automated, we improved our custom Matlab scripts. Please note that all the analysis for the cluster location is performed on STED images, not on normal confocal images. To determine the cluster, first, presynaptic regions (based on Bassoon signals or Dyn1xA signals within boutons) in each STED image are cropped with 900 by 900 nm (regions-of-interest) ROIs. Then, our Matlab scripts calculate the local maxima of fluorescence intensity within the ROIs. To determine the distance between the active zone and the Dyn1xA or EndoA1/2 clusters, the Matlab scripts perform the same local maxima calculations in both channels and make contours at 50% intensity of the local maxima. The minimum distance reflects the shortest distance between the active zone and Dyn1xA/EndoA1/2 contours. To make these points clearer, we modified the main text and the Methods section. In addition, we have added workflow of these analysis as Figure S3.

P9. Main. "Signals of these proteins are acquired by STED microscopy and analyzed by custom MATLAB scripts, similarly to our previous work²³."

P20. Methods. "All the cluster distance measurements are performed on STED images. For the measurements, a custom MATLAB code package²³ was modified using GPT-4 (OpenAI) to perform semi-automated image segmentation and analysis of the endocytic protein distribution relative to the active zone marked by Bassoon or relative to Dyn1xA cluster in STED images. First, the STED images were blurred with a Gaussian filter with radius of 1.2 pixels to reduce the Poisson noise and then deconvoluted twice using the built-in deconvblind function: the initial point spread function (PSF) input is measured from the unspecific antibodies in the STED images. The second PSF (enhanced PSF) input is chosen as the returned PSF from the initial run of blind deconvolution⁶². The enhanced PSF was used to deconvolute the STED images to be analyzed. Each time, 10 iterations were performed. All presynaptic boutons in each deconvoluted image were selected within 30×30-pixel (0.81 μm^2) ROIs based on the varicosity shape and bassoon or Dyn1xA signals. The boundary of active zone or Dyn1xA puncta was identified as the contour that represents half of the intensity of each local maxima in the Bassoon channel. The Dyn1xA clusters and Endophilin A clusters were picked by calculating pixels of local maxima. The distances between the Dyn1xA cluster and active zone boundary or Endophilin A clusters were automatically calculated correspondingly. For the distance measurement, MATLAB distance2curve function (John D'Errico 2024, MATLAB Central File Exchange) first calculated the distance between the local maxima pixel and all the points on the contour of the active zone or Dyn1xA cluster boundary. Next, the shortest distance was selected as the minimum distance. Signals over crossing the ROIs and the Bassoon signals outside of the transfected neurons were excluded from the analysis. The MATLAB scripts are available by request."

In the legend of Figure S3,

"Protein localization in presynapses is determined by semi-automated MATLAB scripts (see Methods).

(A) Series of deconvoluted STED images are segmented to obtain 50-100 presynapse ROIs in each condition.

(B) Two representations of the MATLAB analysis interface are shown. The first channel (ch1, green) is processed to identify the pixels of local maxima within this channel. The second channel (ch2, magenta) is normally an active zone protein, Bassoon. Active zone boundary is determined by the contour generated at 50% intensity of the local maxima of ch2. The contours outside of the transfected neurons are manually selected on the interface and excluded from the analysis. Minimum distances from each pixel of the local maxima in ch1 to the contour in ch2 are calculated and shown in the composite image. The plot “Distance distribution” shows all the minimum distance identified in this presynapses ROI (unit of the y axis is nanometer). The plot “Accumulated distance distribution” shows the accumulated distance distribution from the initial to the current presynapses ROI. The plot “Histogram of total intensity” shows the intensity counts around individual local maxima pixels in ch1.”

15. For the STED microscopy, a representation of the processed image (after deconvolution) and the localization of the peaks would be important to assess the measurement of distances. If Dyn1xA S851/857D is more diffuse, are there still peaks to measure for every synapse?

>>We thank the reviewer for bringing up this important question. In Figure S4C, we have added the position of the local maxima of wild-type and mutant Dyn1xA shown in the main Figure 4E. As the reviewer pointed out, when a protein is more diffuse, it is difficult to find the peak intensity by STED. However, since these proteins are still found at a higher density within a very confined space of a presynapse and synapses are packed with organelles like synaptic vesicles and macromolecules, signals from even diffuse proteins can be detected as clusters, and local maxima can be detected in these images.

To illustrate this point better, we added Reviewer’s Figure 2 below. In this experiment, we transfected neurons with a typical amount of plasmids (2.0 µg/well) or ~10x lower amount (0.25 µg/well). When the density of cytosolic proteins is high (Reviewer’s Figure 2A), the depletion laser has to be strong enough to induce sufficient stimulated emission and resolve protein localization. Insufficient power would produce low resolution images, leading to inappropriate detection of the local maxima (Reviewer’s Figure 1A). Thus, we set our excitation and depletion laser powers to resolve the protein localization to ~40-80 nm at presynapses. Furthermore, to avoid mislocalization of proteins due to the overexpression, we use 0.25-0.5 µg/well (in 12-well plate) of plasmid DNA for transfection, which is around 10 times lower than the amount used in the typical lipofectamine neuronal transfection protocol (Imoto, et al., Neuron 2022). We also change the medium around 20 hours after the transfection instead of the typical 48 hours (Imoto, et al., Neuron 2022). With these modifications and settings, we can obtain the location of the local maxima of the diffuse signals (Reviewer’s Figure 1B and Figure 4E and Figure S4). We modified the Method section to make these points clearer.

Reviewers figure 2

P 17, “Briefly, plasmids were mixed well with 2 μ l Lipofectamine in 100 μ l Neurobasal media and incubated for 20 min. For Dyn1xA and Endophilin A expressions, 0.5 μ g of constructs were used to reduce the overexpression artifacts²³. The plasmid mixture was added to each well with 1 ml of fresh Neurobasal media supplemented with 2 mM GlutaMax and 2% B27. After 4 hours, the medium was replaced with the pre-warmed conditioned media. To prevent too much expression of proteins, neurons were transfected for less than 20 hours and fixed for imaging.”

P 20, “Quality of the STED images are examined by comparing the confocal and STED images and measuring the size of signals at synapses and PSF (non-specific signals from antibodies).”

Legends for Figure S4C,

“(C) STED micrographs of the synapses shown in Figure 4F with an active zone marker Bassoon (magenta). GFP-tagged Dyn1xA, Dyn1xA S851D/857D or Dyn1xA R846A are visualized by antibody staining of GFP (green). Local maxima of Dyn1xA, Dyn1xA S851D/857D or Dyn1xA R846A signals and minimum distance to the active zone boundary are overlaid.”

16. Figures 5 and 6: No specific comment. The data and its analysis are very nice and elegant.

The comment on the lack of rescue of Dyn1xA on endosome maturation may be a bit overstated, because many "controls" (shRNA control Figure S5 or Dyn3 KO in Imoto et al. 2022) have a significant number of endosomes 10 s after stimulation.

>>We thank the reviewer for noting the strength of our data and pointing out this issue on endosomal resolution. In particular, the reviewer is concerned about our interpretation of the ferritin positive endosomes present at 10 s in time-resolved electron microscopy experiments. Indeed, the number of ferritin positive endosomes in *Dyn1* KO, Dyn1xA OEx neurons (0.1/profile) is similar to the control conditions: scramble shRNA control (0.1/profile, Figure S5) and Dyn3KO neurons (0.2/profile) in our previous study (Imoto et al. 2022). Although we do not consider Dyn3 KO as a control, given the presence of abnormal endosomal structures, we agree with the reviewer that scramble shRNA control in Figure S5 does indicate that some ferritin-positive endosomes even at 10 s after stimulation. We would like to note that this result is in stark contrast to our previous studies where we observed the number of ferritin positive endosomes returning to the basal level in both wild-type neurons and many scramble shRNA controls (Watanabe et al. 2014, 2018, Imoto et al 2022). Thus, the majority of the data we have indicate that the number of ferritin positive endosomes returns to basal level by 10 s, suggesting that endosomes are typically resolved into synaptic vesicles by this time. However, given that we do not know the nature of the inconsistency here and we cannot exclude the possibility of overexpression artifact of Dyn1xA as an alternative, we changed the following lines.

p. 10, "Interestingly, the number of ferritin-positive endosomes did not return to the baseline (Figure 5E, F) as in previous studies^{3,35,36}, suggesting that Dyn1xA may not fully rescue the knockout phenotypes or that overexpression of Dyn1xA causes abnormal endosomal morphology."

17. By the way, why did the authors use Dyn1 KO in this study, and not Dyn1,3 DKO as in Imoto et al. 2022?

>>This is simply because Dyn3KO displayed an endosomal defect in our previous study (Imoto et al 2022), and we wanted to focus on endocytic phenotypes of Dyn1 KO and mutant rescues in this study.

18. In the Discussion, the authors present the binding sites (for endophilin and amphiphysin SH3 domains) as independent. However, these proteins form dimers or even multimers as they cluster around the neck of a forming vesicle. Even though they provide evidence in vitro (Figure 3) that in these conditions of high concentration one dyn1xA-PRR binds one SH3 domain, in cells multiple binding sites on the PRR to these proteins may involve avidity effects, as discussed for example in Rosendale et al. 2019 doi 10.1038/s41467-019-12434-9. For example, the high affinity binding of Dyn1-PRR to amphiphysin cannot be explained only by the sequence 830-838.

>> The reviewer suggests "*In the Discussion, the authors present the binding sites (for endophilin and amphiphysin SH3 domains) as independent.*" However, we do not claim these interactions are functionally independent, except in the context of in vitro experiments where they are sequence-independent.

Full Revision

They also suggest “*However, these proteins form dimers or even multimers as they cluster around the neck of a forming vesicle*”. However we do not agree with this in the context of our Discussion, because the evidence of multimers and clustering is convincing but is entirely in vitro data.

Thirdly they comment that “*For example, the high affinity binding of Dyn1-PRR to amphiphysin cannot be explained only by the sequence 830-838.*” We fully agree with the statement and felt we had addressed this in the manuscript. To explain, it’s important to point out our relatively new concept here and previously reported by us (Lin Luo et al 2016, PMID: 26893375) of the existence and importance of SDE and LDE for SH3 domains (Endophilin here, syndapin in our previous report). These elements act at a distance from the so-called core PxxP motifs and they provide much higher affinity and specificity than the core region alone. We had further mentioned this in the p11 discussion “*Although this is a previously characterized binding site for Amphiphysin and is also present in Dyn1xB-PRR, the extended C-terminal tail of Dyn1xA contains short and long distance elements (SDE and LDE) essential for Endophilin binding, making it higher affinity for Endophilin.*” Because the NMR identified F862 as a chemical shift for dynamin, we performed a pulldown with this mutant in the xA746-798 construct (which only contains the higher affinity site) and found that indeed “*F862A reduced Endophilin binding 29% (p<0.01, n=4) (plus see Figure 3D)*” which confirms that it also contributes to endophilin binding at the new site but acting from 17 amino acids distally. Therefore, the core PxxP is just the core, not the whole story. These points support the reviewer, but also extends their concept of avidity and illustrates that avidity is not the only further explanation.

Overall, the reviewer correctly points out that “*multiple binding sites on the PRR to these proteins may involve avidity effects*” could play a role in vivo. We agree that avidity is an additional possibility, not examined in our study. Therefore, as suggested, we added the following sentence to the discussion on the SDE and LDE impacts.

p. 11. “Our pull-down results showed that R846A abolished endophilin binding to xA806-864 (which contains only the second and higher affinity binding site and the associated SDE (A839) and LDE (F862)) and reduced about 40% of endophilin binding to the Dyn1xA-PRR (which contains both binding sites) without affecting its interaction with Amphiphysin, providing important partner specificity, although we cannot exclude the possibility that avidity effects may additionally come in play in vivo ⁴²

Reviewer #1 (Significance (Required)):

This study provides a significant advance on the mechanisms of dynamin recruitment to endocytic zones in presynaptic terminals. The work adds a significant step by experienced labs (Robinson, Watanabe) who have provided important insight in the mechanisms by many publications in the last years.

>> We thank the reviewer for the careful read of our manuscript and positive outlook of our work.

Reviewer #2 (Evidence, reproducibility and clarity (Required)):

1. This is a compelling study that reports a key discovery to understand the molecular mechanism of ultrafast endocytosis. The authors demonstrate that the Dynamin splice version 1xA (Dynamin 1xA) uniquely binds Endophilin A, in contrast to Dynamin splice version 1xB (Dynamin 1xB) that does not bind Endophilin A and it is not required for ultrafast endocytosis. In addition, the Endophilin A binding occurs in a dephosphorylation-regulated manner. The study is carefully carried out and it is based on high quality data obtained by means of advanced biochemical methodologies, state-of-art flash-freezing electron microscopy analysis, superresolution microscopy and dynamic imaging of exo-and endocytosis in neuronal cultures. The results convincingly support the conclusions.

>>We thank the reviewer for supporting the conclusions of our study.

2. Although additional experiments are not essential to support the claims of the paper there is room, however, for improvement within the pHluorin experiments. These experiments, that are clearly informative and consistent with the rest of experimental data, do not apply the useful approach to separate endo- from exocytosis. The use of bafilomycin or folimycin to block the vesicular proton pump allows the unmasking the endocytosis that is occurring during the stimulus, that should correspond to ultrafast endocytosis. It would be very elegant to demonstrate that such a component, as expected according to the electron microscopy data, requires the binding of Endophilin A to Dynamin 1xA. If the authors have the pHluorin experiments running, the suggested experiments are very much doable because the reagents and the methodology is already in place and the new data could be generated in around six weeks.

>> We thank the reviewer for the suggestion. The reviewer is concerned that vGlut1 pHluorin experiment in Figure 6 may not correspond to ultrafast endocytosis. We agree that bafilomycin/folimycin treatment will reveal the amount of endocytosis that takes place while neurons are stimulated. However, we are not certain that endocytosis during this phase would fully correspond to ultrafast endocytosis because reacidification of endocytosed vesicles typically takes 3-4 s (Atluri and Ryan, 2006, PMID: 16495458; although see <https://elifesciences.org/articles/36097>) and thus, the nature of endocytosis cannot be fully determined by this assay. To claim that endocytosis measured by pHluorin assay during stimulation all correspond to ultrafast endocytosis, we would need to perform very careful work to track single pHluorin molecules at the ultrastructural level and correlate their internalization to pHluorin signals. Perhaps, a rapid acid quench technique used by the Haucke group would also be appropriate to estimate the amount of ultrafast endocytosis (Soykan et al. 2017 PMID: 28231467), but we are not set up to perform such experiments here. Also, our lead author, Yuuta Imoto, is leaving the lab to start up his own group, and it will take us months rather than weeks to get the requested experiments done. Since the point of this experiment was to test whether the interaction of Dyn1xA and EndoA is essential for protein retrieval regardless of the actual mechanisms and the reviewer acknowledges that this point is sufficiently supported by the experiments, we will set this experiment as the priority for the next paper.

Instead of the bafilomycin or rapid acid quenching experiments, we have now added data from vglut1-pHluorin experiment with a single action potential. With a single action potential, all synaptic vesicle recycling is mediated by ultrafast endocytosis in these neurons (Watanabe et al, 2013 PMID: 24305055; Watanabe et al. 2014, PMID: 25296249). Our electron microscopy experiments in Figure 5 is also performed with a single action potential. As with 10 action potentials, 20 Hz experiments, re-acidification of vglut1-pHluorin is blocked when Dyn1 and EndophilinA1 interaction is disrupted (Figure 6 F-I). We added a description of this result as below.

P 11. “Similar defects were observed when the experiments were repeated with a single action potential – synaptic vesicle recycling is mediated by ultrafast endocytosis with this stimulation paradigm²⁵ (S851/857 recovery is 73.3% above the baseline; R846A, recovery is 30.0% above the baseline) (Figure S9 A-D). Together, these results suggest that the 20 amino acid extension of Dyn1xA is important for recycling of synaptic vesicle proteins mediated by specific phosphorylation and Endophilin binding sites within the extension.”

3. The methods are carefully explained. Some of the experiments are only replicated in two cultures and the authors should justify the reasons to convince the audience that the approaches used have enough low variability for not increasing the n number. The pHluorin experiments, however, are performed only in a single culture; they should replicate these experiments in at least 3 different cultures (three different mice).

>>The reviewer is correct. The variability is very low in our ultrastructural studies and STED imaging, and thus, in all our previous publications, two independent cultures are used. We do agree that in the ideal case, we would like to have three independent cultures, but given the nature of ultrastructural studies (control, mutants, and multiple time points), triplicating the data would add another year to our work. We are currently developing AI-based segmentation analysis, and once this pipeline is established, we will be able to increase N. However, please note that for these experiments, we examine around 200 synapses from each condition in electron microscopy studies (Table S2)– these numbers are far more than the gold standard in the field. Likewise, 50-100 synapses are examined for STED experiments (Table S2). To examine variability of our analysis results, we compared a significance between the dataset using cumulative curves and Kolmogorov–Smirnov test (Figure S11). As shown in the summarized data and p value in each condition, there are no significant difference between the datasets.

For pHluorin analysis, the reviewer is correct. We repeated the experiments twice to increase the N after the initial submission. The data are consistent, and the conclusions are not changed by the additional experiments (Figure 6 and Figure S9). We also changed the Statistical analysis section in Methods as below.

p. 19. “All electron microscopy data are pooled from multiple experiments after examined on a per-experiment basis (with all freezing on the same day); none of the pooled data show significant deviation from each replicate (Table S2).”

p 19, “All fluorescence microscopy data were first examined on a per-experiment basis. For Figure 4, the data were pooled; none of the pooled data show significant deviation from each replicate

(Figure S11 and Table S2). Sample sizes were 2 independent cultures, at least 50-100 synapses from 4 different neurons in each condition..”

Legends for Figure S11

Figure S11. Data variability in Figure 4.

Cumulative curves are made from each dataset of (A) distance of Endophilin A1 puncta from the edge of Dyn1xA puncta, (B) distance of Endophilin A2 puncta from the edge of Dyn1xA puncta, distance distribution of Dyn1xA from active zone edge in (C) neurons expressing wild-type Dyn1xA-GFP, (D) Dyn1xA-S851/857-GFP and (E) Dyn1xA-R846-GFP. $n > 4$ coverslips from 2 independent cultures. Kolmogorov–Smirnov (KS) test, p values are indicated in each plot.

Minor comments:

4. Prior studies referenced appropriately and the text and figures are clear and accurate.

>> We thank the reviewer for the careful read of our manuscript.

5. The authors should discuss about the mediators (enzymes) responsible for dephosphorylation of phospho-box 2 that is key for the Dynamin 1xa-Endophilin A interaction.

>>We thank the reviewer for the suggestion. We added a discussion on a potential mediator, Dyrk1, as below.

p. 12. "What are the kinases that regulate Dyn1? The phosphorylation of phosphobox-1 is mediated by Glycogen synthase kinase-3 beta (GSK3 β) and Cyclin-dependent kinase 5 (CDK5)¹⁷, while phosphobox-2 is likely phosphorylated by Trisomy 21-linked dual-specificity tyrosine phosphorylation-regulated kinase 1A (Mnb/Dyrk1)^{44,45} since Ser851 in phosphobox-2 is shown to be phosphorylated by Mnb/Dyrk1 *in vitro*³². Furthermore, overexpression of Mnb/Dyrk1 in cultured hippocampal neurons causes slowing down the retrieval of a synaptic vesicle protein vGlut1⁴⁶. Consistently, our data showed that phosphomimetic mutations in phosphobox-2 results disruption of Dyn1xA localization, perturbation of ultrafast endocytosis, and slower kinetics of vGlut1 retrieval. However, how these kinases interplay to regulate the interaction of Dyn1xA, Syndapin1 and Endophilin A1 for ultrafast endocytosis is unknown."

6. It would be very helpful to include a final cartoon depicting the key protein-protein interactions regulated by dephosphorylation (activity) and the sequence of molecular events that leads to ultrafast endocytosis

>>As suggested, we made a model figure, (new Figure 7) showing how Dyn1xA and its interaction with EndoA and Syndapin1 increases the kinetics of endocytosis at synapses.

Regarding the sequence of molecular events, we think that there are already dephosphorylated fraction of Dyn1xA molecules sitting on the endocytic zone at the resting state and they mediate

ultrafast endocytosis. However, it is equally possible that activity-dependent dephosphorylation of Dyn1xA also may play a role (Jing et al. 2011, PMID: 21730063). However, we have no evidence about the sequence of activity dependent modulation of Dyn1xA and its binding partners during ultrafast endocytosis yet. This is much beyond what we have reported in this work and therefore, excluded from the model figure. We added the following to the end of the discussion:

p13, “Nonetheless, these results suggest that Dyn1xA long C-terminal extension allows multivalent interaction with endocytic proteins and that the high affinity interaction with Endophilin A1 permits phospho-regulation of their interaction and defines its function at synapses (Figure S7)”.

Figure legend Figure 7,

“Figure 7. Schematics depicting how specific isoforms Dyn1xA and Endophilin A mediate ultrafast endocytosis.

A splice variant of dynamin 1, Dyn1xA, but not other isoforms/variants can mediate ultrafast endocytosis. (A) Dyn1xA has 20 amino acid extension which introduces a new high affinity Endophilin A1 binding site. Three amino acids, R846 at the splice site boundary, S851 and S857, act as long-distance element which can enhance affinity of proline rich motifs (PRM) to SH3 motif from outside of the PRM core sequence PxxP. (B) At a resting state, Dyn1xA accumulates at endocytic zone with SH3 containing BAR protein Syndapin 1²³ and Endophilin A1/2. When phosphobox-1 (Syndapin1 binding) and phosphobox-2 (Endophilin A1/2 binding, around S851/S857) within Dyn1xA PRD are phosphorylated, these proteins are diffuse within the cytoplasm. A dephosphorylated fraction of Dyn1xA molecules can interact with these BAR domain proteins. Loss of interactions including Dyn1xA-R846A or -S851/857D mutations, disrupts endocytic zone pre-accumulations. Consequently, ultrafast endocytosis fails.”

Reviewer #2 (Significance (Required)):

This is a remarkable and important advance in the field of endocytosis. The study reports a key discovery to understand the molecular mechanism of ultrafast endocytosis. Scientist interested in synaptic function and the general audience of cell biologist interested in membrane trafficking will very much value this study. The mechanism reported will potentially be included in textbooks in the near future.

My field of expertise includes molecular mechanisms of presynaptic function and membrane trafficking.

I have not enough experience to evaluate the quality of the NMR experiments, however, I do not have any problem at all with, in my opinion, elegant results reported.

>>We thank the reviewer for the positive outlook of our manuscript.

Dear Shigeki,

Congratulations on a great revision, overall the referees have been positive. However, referee 1 has a few minor concerns that we ask you to non-experimentally address in a revised version. When you submit your revision, please also take care of the following editorial items and add this also to your point-by-point response:

1. Please provide an author checklist, see link below.
2. The figures should be in separate files. Please upload the main figures as high res figure files in TIFF, EPS or PDF format. Legends should be in the manuscript, after References.
3. Please reduce the number of keywords to 5.
4. Please provide a data availability section as outlined in our guide to authors online.
5. Please provide the NSF 1727260 funding information online in eJP.
6. Please remove the author contribution section from the main manuscript.
7. Please review our new policy on conflict of interests on the EMBO author guide website and update the title of this section to: Disclosure and competing interests statement.
8. Please update the reference format so that it is listed in alphabetical order and please remove DOIs of published work.
9. We require the publication of source data, particularly for electrophoretic gels and blots and graphs, with the aim of making primary data more accessible and transparent to the reader. It would be great if you could provide me with a PDF file per figure that contains the original, uncropped and unprocessed scans of all or key gels used in the figure or for graphs, an Excel spreadsheet with the original data used to generate the graphs. The PDF files should be labeled with the appropriate figure/panel number, and should have molecular weight marker; further annotation could be useful but is not essential. The PDF files will be published online with the article as supplementary "Source Data" files.
10. We include a synopsis of the paper (see <http://emboj.embopress.org/>). Please provide me with a general summary statement and 3-5 bullet points that capture the key findings of the paper.
11. We also need a summary figure for the synopsis. The size should be 550 wide by 200-440 high (pixels). You can also use something from the figures if that is easier.
12. Table S1 could be considered source data files for Fig4 and Fig5 rather than being Table S1.
13. Please rename the Summary to Abstract
14. Up to 5 supplementary figures can be made Expanded View Figures. Nomenclature is "Figure EV"1 etc; the files should be uploaded as high res figure files in TIFF, EPS or PDF format, the legends should stay in the manuscript under the heading "Expanded View Figure Legends", after the main figure legends. The remaining figures should be compiled with their legends in an appendix file: a PDF, with page numbers and a table of contents. Nomenclature is "Appendix Figure S1" etc.
15. Please provide a 'data information' section in the legend of figures 4d, f, g; 5b, d, f, h, j-l; 6a-e.
16. Please note that the legends for figures 5b-i is not provided in the sequential manner (legends for figures 5c, e, g, i, are provided before legends of figures 5b, d, f, h, respectively). This needs to be rectified.
17. Please note that the p values are not represented in the figure 5b, d, f, h, j-l, however statistical test related in provided in the legends of the corresponding figures. This needs to be rectified.
18. We do not allow statistics on data where N=2, please correct this for figure 3l-m.

We appreciate the opportunity to consider your work for publication and look forward to your revision.

Warm wishes,
Kelly

Kelly M Anderson, PhD

Editor, The EMBO Journal
k.anderson@embojournal.org

- a complete author checklist, which you can download from our author guidelines
(<https://www.embopress.org/page/journal/14602075/authorguide>).

- Expanded View files (replacing Supplementary Information)

Referee #1:

I have no major comment, other than my congratulations to the very thorough and rigorous answers to the points raised previously after my review for ReviewCommons. Just a few comments:

Figure 4 legend: B "The top image shows an axon congaing (!) multiple boutons. Signals show overexpression of mCherry-tagged Dyn1xA (Dyn1xA) and GFP-tagged Endophilin A1 (EndoA1)." In panel A, it is GFP-Dyn1xA and mCherry-EndoA1. Is this a typo or are they truly different transfections?

I also agree that localization of endogenous axonal EndophilinA1 by genome editing would be nice, but is not mandatory for this publication, and I am satisfied with the new analysis provided in Figure 4. Likewise, I appreciate the analysis of STED images of neurons expressing low amount of GFP matching mCh-EndoA1 to determine the location of multiple clusters. I think this Figure could be added as a supplementary figure (if allowed in the Journal format).

Finally, I find the discussion now very well detailed and balanced. The addition on the possible kinases phosphorylating Dyn1xA (suggested by Reviewer 2), as well as the cartoon, is very informative.

Referee #2:

I have previously reviewed an earlier version of this manuscript. The authors have provided convincing responses to my concerns and they have improved the manuscript.

Rev_Com_number: RC-2023-02248
New_manu_number: EMBOJ-2024-117221-T
Corr_author: Watanabe
Title: Dynamin 1xA interacts with Endophilin A1 via its spliced long C-terminus for ultrafast endocytosis

Response to Referee #1:

I have no major comment, other than my congratulations to the very thorough and rigorous answers to the points raised previously after my review for ReviewCommons. Just a few comments:

Figure 4 legend: B "The top image shows an axon congaing (!) multiple boutons. Signals show overexpression of mCherry-tagged Dyn1xA (Dyn1xA) and GFP-tagged Endophilin A1 (EndoA1)." In panel A, it is GFP-Dyn1xA and mCherry-EndoA1. Is this a typo or are they truly different transfections?

>We appreciate the reviewer #1 for pointing out the error. We fixed it as GFP-tagged Endophilin A2 (EndoA2). Please note that for Figure S4A and B (now Figure EV4), different transfections are used.

I also agree that localization of endogenous axonal EndophilinA1 by genome editing would be nice, but is not mandatory for this publication, and I am satisfied with the new analysis provided in Figure 4.

>We are grateful that the reviewer agreed.

Likewise, I appreciate the analysis of STED images of neurons expressing low amount of GFP matching mCh-EndoA1 to determine the location of multiple clusters. I think this Figure could be added as a supplementary figure (if allowed in the Journal format).

>We agree with the reviewer #1. The reviewer's figure #2 is now incorporated into manuscript as Appendix Figure S2. Legend is also added as below,

“Appendix Figure S2. Different plasmid expression levels and the depletion laser power affect cluster analysis in STED images.

(A) STED images show high expression of cytosolic GFP scanned by low depletion laser power. Anti-Bassoon antibody is used as presynaptic marker. Bottom panel shows distance of misdirected GFP clusters from active zone edge.

(B) STED images show low expression of cytosolic GFP scanned by sufficient depletion laser power. Anti-Bassoon antibody is used as presynaptic marker. Bottom panel shows distance of GFP clusters from active zone edge. By using appropriate amount of plasmid transfection and the depletion laser power, diffuse cytosolic GFP clusters are detected.”

Finally, I find the discussion now very well detailed and balanced. The addition on the possible kinases phosphorylating Dyn1xA (suggested by Reviewer 2), as well as the cartoon, is very informative.

>We appreciate positive comments on our revised manuscript.

Response to Referee #2:

I have previously reviewed an earlier version of this manuscript. The authors have provided convincing responses to my concerns and they have improved the manuscript.

>We appreciate reviewer#2's positive comment on our revised manuscript.

Dear Shigeki,

Congratulations on an excellent manuscript, I am pleased to inform you that your manuscript has been accepted for publication in The EMBO Journal. Thank you for your comprehensive response to the referee concerns and for providing detailed source data. It has been a pleasure to work with you to get this to the accepted stage.

I will begin the final checks on your manuscript before submitting to the publisher next week. Once at the publisher, it will take about 3 weeks for your manuscript to be published online. As a reminder, the entire review process including referee concerns and your point-by-point responses, will be available to readers.

I will be in touch throughout the final editorial process until publication. In the meantime, I hope you find time to celebrate!

Kind regards,

Kelly

Kelly M Anderson, PhD
Editor, The EMBO Journal
k.anderson@embojournal.org
